# Sparse MoEs meet Efficient Ensembles

**James Urquhart Allingham**[1,*]                         *jua23@cam.ac.uk*

**Florian Wenzel**[2,†]                                   *fln.wenzel@gmail.com*

**Zelda E Mariet, Basil Mustafa**                         *{zmariet,basilm}@google.com*

**Joan Puigcerver, Neil Houlsby**                         *{jpuigcerver,neilhoulsby}@google.com*

**Ghassen Jerfel**[3,†]                                   *ghassen@google.com*

**Vincent Fortuin**[1,4,*]                                *vbf21@cam.ac.uk*

**Balaji Lakshminarayanan, Jasper Snoek**                 *{balajiln,jsnoek}@google.com*

**Dustin Tran, Carlos Riquelme, Rodolphe Jenatton**       *{trandustin,rikel,rjenatton}@google.com*

*Google Research, Brain Team;* [1] *University of Cambridge;* [2] *no affiliation;* [3] *Waymo;* [4] *ETH Zürich*

**Reviewed on OpenReview:** *https://openreview.net/forum?id=i0ZM36d2qU*

## Abstract

Machine learning models based on the aggregated outputs of submodels, either at the activation or prediction levels, often exhibit strong performance compared to individual models. We study the interplay of two popular classes of such models: ensembles of neural networks and sparse mixture of experts (sparse MoEs). First, we show that the two approaches have complementary features whose combination is beneficial. This includes a comprehensive evaluation of sparse MoEs in uncertainty related benchmarks. Then, we present *efficient ensemble of experts* ($\mathrm{E}^3$), a scalable and simple ensemble of sparse MoEs that takes the best of both classes of models, while using up to 45% fewer FLOPs than a deep ensemble. Extensive experiments demonstrate the accuracy, log-likelihood, few-shot learning, robustness, and uncertainty improvements of $\mathrm{E}^3$ over several challenging vision Transformer-based baselines. $\mathrm{E}^3$ not only preserves its efficiency while scaling to models with up to 2.7B parameters, but also provides better predictive performance and uncertainty estimates for larger models.

## 1 Introduction

Neural networks (NNs) typically use all their parameters to process an input. Sustaining the growth of such models—reaching today up to 100B+ parameters (Brown et al., 2020)—is challenging, e.g., due to their high computational and environmental costs (Strubell et al., 2019; Patterson et al., 2021). In this context, sparse mixtures of experts (sparse MoEs) employ *conditional computation* (Bengio et al., 2013) to combine multiple submodels and route examples to specific "expert" submodels (Shazeer et al., 2017; Lepikhin et al., 2021; Fedus et al., 2021; Riquelme et al., 2021). Conditional computation can decouple the growth of the number of parameters from the training and inference costs, by only activating a subset of the overall model in an input-dependent fashion.

Paralleling this trend, the deployment of ML systems in safety-critical fields, e.g., medical diagnosis (Dusenberry et al., 2020b) and self-driving cars (Levinson et al., 2011), has motivated the development of reliable

---

*Work done as a Google Research intern.
†Work done while at Google Research.

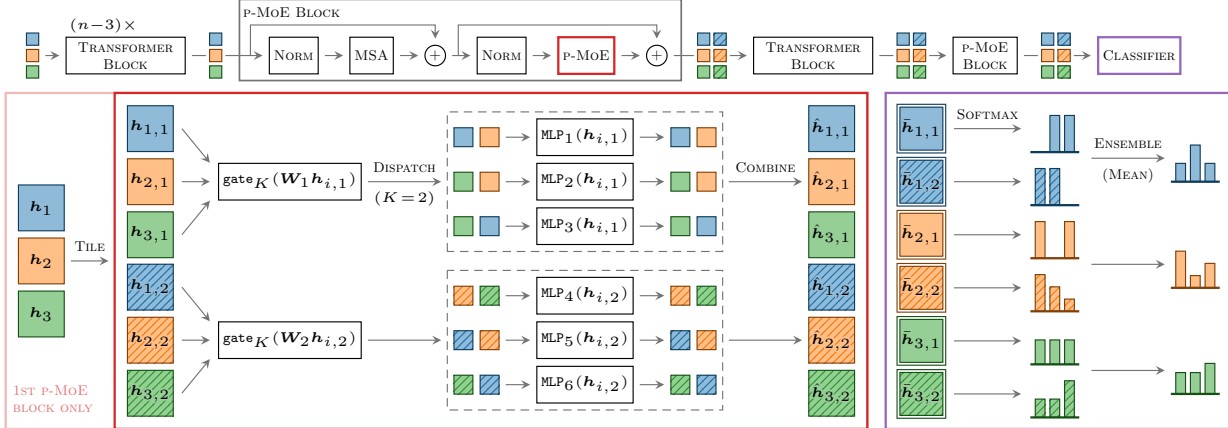

Figure 1: End-to-end overview of $E^3$ with $E=6$ experts, partitioned into $M=2$ groups, with sparsity of $K=2$, and a "last-2" configuration. **Top**: $E^3$ contains a sequence of transformer blocks, followed by alternating transformer and p(artitioned)-MoE blocks. As in ViT, images are split into patches whose embeddings are processed by each block. Here, we show 1 embedding for each of three images (🟦, 🟧, 🟩). **Bottom left**: In a p-MoE block, we replace the transformer block's MLP with parallel partitioned expert MLPs, see (2). The effect of the routing weights is not depicted. Embeddings are tiled (▨) in the first p-MoE block only. **Bottom right**: The classifier averages predictions from the final tiled representations (▣).

deep learning, e.g., for *calibrated and robust predictions* (Ovadia et al., 2019). Among the existing approaches, ensembles of NNs have remarkable performance for calibration and accuracy under dataset shifts (Ovadia et al., 2019). These methods improve reliability by aggregating the predictions of individual submodels, referred to as ensemble members. However, this improvement comes at a significant computational cost. Hence, naively ensembling NNs that continue to grow in size becomes less and less feasible. In this work, we try to overcome this limitation. Our core motivation is to improve the robustness and uncertainty estimates of large-scale fine-tuned models through ensembling, but to do so in a tractable—and thus practically useful—manner, by carefully developing a hybrid approach using advances in sparse MoEs.

While sharing conceptual similarities, these two classes of models—MoEs and ensembles—have different properties. Sparse MoEs adaptively combine their experts depending on the inputs, and the combination generally happens at internal activation levels. Ensembles typically combine several models in a static way and at the prediction level. Moreover, these two classes of models tend to be benchmarked on different tasks: few-shot classification for MoEs (Riquelme et al., 2021) and uncertainty-related evaluation for ensembles (Ovadia et al., 2019; Gustafsson et al., 2020). For example, sparse MoEs are seldom, if ever, applied to the problems of calibration.

Here, we study the interplay between sparse MoEs and ensembles. This results in two sets of contributions:

**Contribution 1: Complementarity of sparse MoEs and ensembles.** We show that sparse MoEs and ensembles have complementary features and benefit from each other. Specifically:

- The adaptive computation in sparse MoEs and the static combination in ensembles are orthogonal, with additive benefits when associated together. At the intersection of these two model families is an exciting trade-off between performance and compute (FLOPs). That is, the frontier can be mapped out by varying the ensemble size and the sparsity of MoEs.

- Over tasks where either sparse MoEs or ensembles are known to perform well, naive—and computationally expensive—ensembles of MoEs provide the best predictive performance. Our benchmarking effort includes the first evaluation of sparse MoEs on uncertainty-related vision tasks, which builds upon the work of Riquelme et al. (2021).

**Contribution 2: Efficient ensemble of experts.** We propose Efficient Ensemble of Experts ($E^3$), see Figure 1, an efficient ensemble approach tailored to sparse MoEs:

- $E^3$ improves over sparse MoEs across few-shot error, likelihood and calibration error. $E^3$ matches the performance of deep ensembles while using from 30% to 45% fewer FLOPs.

- $E^3$ gracefully scales up to 2.7B parameter models.

- $E^3$ is both simple—requiring only minor implementation changes—and convenient—$E^3$ models can be fine-tuned directly from standard sparse-MoE checkpoints. Code can be found at https://github.com/google-research/vmoe.

## 2 Preliminaries

We focus on classification tasks where we learn classifiers of the form $f(\boldsymbol{x};\boldsymbol{\theta})$ based on some training data $\mathcal{D} = \{(\boldsymbol{x}_n, y_n)\}_{n=1}^N$. A pair $(\boldsymbol{x}_n, y_n)$ corresponds to an input $\boldsymbol{x}_n \in \mathbb{R}^P$ together with its label $y_n \in \{1, \ldots, C\}$ belonging to one of the $C$ classes. The model $f(\cdot;\boldsymbol{\theta})$ is parametrized by $\boldsymbol{\theta}$ and outputs a $C$-dimensional probability vector. We use $\circ$ to refer to the matrix element-wise product.

### 2.1 Vision Transformers and Sparse MoEs

**Vision Transformers.** Throughout the paper, we choose the model $f$ to be a vision Transformer (ViT) (Dosovitskiy et al., 2021). ViT is growing in popularity for vision, especially in transfer-learning settings where it was shown to outperform convolutional networks while requiring fewer pre-training resources. ViT operates at the level of patches. An input image is split into equal-sized patches (e.g., $32 \times 32$, $16 \times 16$, or $14 \times 14$ pixels) whose resulting sequence is (linearly) embedded and processed by a Transformer (Vaswani et al., 2017). The operations in the Transformer then mostly consist of a succession of multiheaded self-attention (MSA) and MLP layers. ViT is defined at different scales (Dosovitskiy et al., 2021): S(mall), B(ase), L(arge) and H(uge); see specifications in Appendix A.1. For example, ViT-L/16 stands for a large ViT with patch size $16 \times 16$.

**Sparse MoEs and V-MoEs.** The main feature of sparsely-gated mixture-of-experts models (sparse MoEs) lies in the joint use of sparsity and *conditional computation* (Bengio et al., 2013). In those models, we only activate a small subset of the network parameters *for a given input*, which allows the total number of parameters $\boldsymbol{\theta}$ to grow while keeping the overall computational cost constant. The *experts* are the subparts of the network activated on a per-input fashion.

Central to our study, Riquelme et al. (2021) recently extended ViT to sparse MoEs. Their extension, referred to as V-MoE, follows the successful applications of sparse models in NLP (Shazeer et al., 2017). Riquelme et al. (2021) show that V-MoEs dominate their "dense" ViT counterparts on a variety of tasks for the same computational cost. In the specific case of V-MoEs, the experts are placed in the MLP layers of the Transformer, a design choice reminiscent of Lepikhin et al. (2021) in NLP. Given the input $\boldsymbol{h} \in \mathbb{R}^D$ of such a layer, the output of a single $\texttt{MLP}(\boldsymbol{h})$ is replaced by

$$\texttt{MoE}(\boldsymbol{h}) = \sum_{e=1}^E g_e(\boldsymbol{h}) \cdot \texttt{MLP}_e(\boldsymbol{h}) \quad \text{with} \quad \{g_e(\boldsymbol{h})\}_{e=1}^E = \texttt{top}_K(\texttt{softmax}(\boldsymbol{W}\boldsymbol{h})), \tag{1}$$

where the *routing* weights $\{g_e(\boldsymbol{h})\}_{e=1}^E$ combine the outputs of the $E$ different experts $\{\texttt{MLP}_e\}_{e=1}^E$. To sparsely select the experts, $\texttt{top}_K$ sets all but the $K$ largest weights to zero. The router parameters $\boldsymbol{W} \in \mathbb{R}^{E \times D}$ are trained together with the rest of the network parameters. We call the layer defined by (1) an MoE layer. In practice, the weights $\{g_e(\boldsymbol{h})\}_{e=1}^E$ are obtained by a noisy version of the routing function $\texttt{top}_K(\texttt{softmax}(\boldsymbol{W}\boldsymbol{h} + \sigma\varepsilon))$ with $\varepsilon \sim \mathcal{N}(\boldsymbol{0}, \boldsymbol{I})$, which mitigates the non-differentiability of $\texttt{top}_K$ when combined with auxiliary losses (see Appendix A in Shazeer et al. (2017)). Making non-differentiable operators smooth with some noise injection is an active area of research (Berthet et al., 2020; Abernethy, 2016; Duchi et al., 2012). We use the shorthand $\texttt{gate}_K(\boldsymbol{z}) = \texttt{top}_K(\texttt{softmax}(\boldsymbol{z} + \sigma\varepsilon))$ and take $\sigma = 1/E$ as in Riquelme et al. (2021).

In this paper, we consider the "last-$n$" setting of Riquelme et al. (2021) wherein only a few MoE layers are placed at the end of the Transformer ($n = 2$ for the {S, B, L} scale and $n = 5$ for H). This setting retains most of the performance gains of V-MoEs while greatly reducing the training cost.

Table 1: Overview of key properties of sparse MoEs, ensembles, and $\text{E}^3$. $\text{E}^3$ acheives the best of both worlds. `dense` is a base model upon which we add the sparse MoE or ensemble logic, e.g., a ViT model in this paper.

|  | PREDICTIONS | COMBINATIONS | CONDITIONAL COMPUTATION | COST |
|---|---|---|---|---|
| **Sparse MoEs** | Single | Activation level | Yes, adaptively per-input | $\approx$ `dense` |
| **Ensembles** | Multiple | Prediction level | No, static | $>$ `dense` |
| $\text{E}^3$ | Multiple | Activation & prediction level | Yes, adaptively per-input | $\approx$ `dense` |

### 2.2 Ensembles of Neural Networks

**Ensembles.** We build on the idea of ensembles, which is a known scheme to improve the performance of individual models (Hansen & Salamon, 1990; Geman et al., 1992; Krogh & Vedelsby, 1995; Opitz & Maclin, 1999; Dietterich, 2000; Lakshminarayanan et al., 2017). Formally, we assume a set of $M$ model parameters $\Theta = \{\boldsymbol{\theta}_m\}_{m=1}^M$. We refer to $M$ as the *ensemble size*. Prediction proceeds by computing $\frac{1}{M}\sum_{\boldsymbol{\theta}\in\Theta} f(\boldsymbol{x};\boldsymbol{\theta})$, i.e., the average probability vector over the $M$ models. To assess the diversity of the predictions in the ensemble, we will use the KL divergence $D_{\text{KL}}(f(\boldsymbol{x}_t;\boldsymbol{\theta}_m)\|f(\boldsymbol{x}_t;\boldsymbol{\theta}_{m'}))$ between the predictive distributions $f(\boldsymbol{x}_t;\boldsymbol{\theta}_m)$ and $f(\boldsymbol{x}_t;\boldsymbol{\theta}_{m'})$, averaged over the test input $\boldsymbol{x}_t$ and all pairs $(m,m')$ of ensemble members.

**Batch ensembles.** Wen et al. (2019) construct a *batch ensemble* (BE) as a collection of submodels, with the parameters $\boldsymbol{\theta}_m \in \Theta$ sharing components. This mitigates the computational and memory cost of ensembling, while still improving performance. We focus on the example of a single dense layer in $f$ with parameters $\boldsymbol{U} \in \mathbb{R}^{D\times L}$, assuming no bias. BE defines $M$ copies of parameters $\{\boldsymbol{U}_m\}_{m=1}^M$ so that $\boldsymbol{U}_m = \boldsymbol{U} \circ (\boldsymbol{r}_m\boldsymbol{s}_m^\top)$, where $\boldsymbol{U}$ are parameters shared across ensemble members, and $\boldsymbol{r}_m$ and $\boldsymbol{s}_m$ are separate $D$- and $L$-dimensional vectors for ensemble member $m$. Given an input, BE produces $M$ outputs, which are averaged after applying all layers. Despite the simple rank-1 parametrization, BE leads to remarkable predictive performance and robustness (Wen et al., 2019). Notably, the efficiency of BE relies on both the parameter sharing and the tiling of the inputs to predict with the $M$ ensemble members, two insights that we exploit in our paper.

### 2.3 Pre-training and Fine-tuning

Large-scale Transformers pre-trained on *upstream* tasks were shown to have strong performance when fine-tuned on smaller *downstream* tasks, across a variety of domains (Devlin et al., 2018; Dosovitskiy et al., 2021; Radford et al., 2021). We follow this paradigm and focus on the fine-tuning of models pre-trained on JFT-300M (Sun et al., 2017), similar to Riquelme et al. (2021). We will thus assume the availability of already pre-trained ViT and V-MoE model checkpoints. Our assumption relies on the growing popularity of transfer learning, e.g. Kolesnikov et al. (2020), and the increasing accessibility of pre-trained models in repositories such as `www.tensorflow.org/hub` or `www.pytorch.org/hub`. The fine-tuning of all the approaches we study here, including extensions of ViT and V-MoE, will be either directly compatible with those checkpoints or require only mild adjustments, e.g., reshaping or introducing new downstream-specific parameters (see Appendix B). Also, unless otherwise mentioned, the performance we report will always be downstream, e.g., for ImageNet (Deng et al., 2009) or CIFAR10/100 (Krizhevsky, 2009). In all our comparisons, we will use the downstream training floating point operations per second (FLOPs), or GFLOPs (i.e., $10^9\times$FLOPs), to quantify the computational cost of the different methods.

## 3 Sparse MoEs meet Ensembles

As illustrated in Table 1, sparse MoEs and ensembles have different properties. For instance, ensembles typically do not use conditional computation and just statically combine members at the prediction level. This contrasts with sparse MoEs where the different experts are combined at internal activation levels while enjoying per-input adaptivity through the routing logic, see (1). In terms of cost, sparse MoEs are usually designed to match the inference time of their dense counterparts whereas ensembles, in their simplest forms, will typically lead to a substantial overhead. In this section, we study the extent to which these properties are complementary and may benefit from each other. In Section 5, we further evaluate this complementarity on tasks where either sparse MoEs or ensembles are known to perform well, e.g., few-shot and out-of-distribution (OOD) evaluations, respectively.

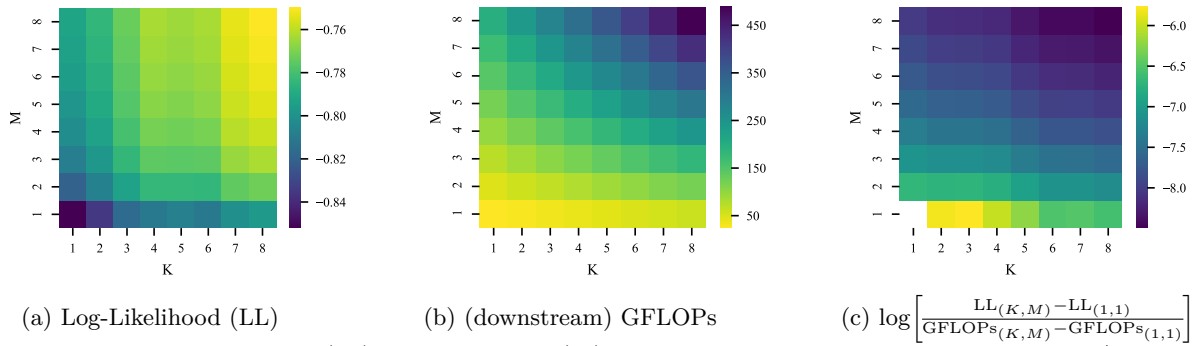

(a) Log-Likelihood (LL)        (b) (downstream) GFLOPs        (c) $\log\left[\frac{\text{LL}_{(K,M)} - \text{LL}_{(1,1)}}{\text{GFLOPs}_{(K,M)} - \text{GFLOPs}_{(1,1)}}\right]$

Figure 2: Increasing static ($M$) and adaptive ($K$) ensembling on ImageNet, for V-MoE-S/32. Yellow/**purple** indicates better/worse performance. Increasing *both* static and adaptive ensembling is beneficial, the latter being more efficient.

To investigate the interactions of the properties in Table 1, we study the performance of *downstream* deep ensembles (i.e., with all ensemble members having the same upstream checkpoint) formed by $M$ independent V-MoEs with $E$ experts per MoE layer and a sparsity level $K$ (the larger $K$, the more selected experts). $M$ controls the static combination, while $K$ and $E$ impact the adaptive combination of experts in each sparse MoE model. We report in Figure 2 the ImageNet performance and compute cost for ensembles with varying choices of $K$ and $M$, while keeping $E = 32$ fixed. We focus on $K$ rather than $E$ to explore adaptive computation, as we found the performance quickly plateaus with $E$ (see Figure 12 in the Appendix). Also, by fixing $E = 32$, we match more closely the setup of Riquelme et al. (2021). The architecture of the V-MoE is ViT-S/32, see details in Appendix A.7. We make the following observations:

**Investigating the cumulative effects of adaptive and static ensembling.** In the absence of ensembles (i.e., when we consider $M = 1$), and given a fixed number of experts, the authors of Riquelme et al. (2021) already reported an increase in performance as $K$ gets larger. Interestingly, we observe that for each value of $K$, it is also beneficial to increase the ensemble size $M$.

In other words, the static combination of ensembles is beneficial when applied to sparse MoEs. This observation is perhaps surprising since adaptive combination may already encapsulate the effect of static combination. Figure 3, and Appendix L.1, show that the combination of static ensembling and adaptivity is beneficial to NLL for a range of ViT families. We also see that the benefits of static ensembling are similar for V-MoE and ViT (which does not have any adaptivity).

**Investigating ensembles of sparse MoEs with fewer experts.** In Appendix L.2, we compare the performance of a V-MoE with $E = 32$ experts and ensembles of V-MoEs with fewer experts, namely ($M = 2$, $E = 16$) and ($M = 4$, $E = 8$). We see that the performance—e.g., as measured by NLL—is better for ($M = 4$, $E = 8$) than ($M = 2$, $E = 16$) which is in turn better than ($M = 1$, $E = 32$). Thus, we conclude that *reducing the number of experts only mildly affects the combination of adaptive and static ensembling.*

**Investigating the trade-off between FLOPs and performance.** Without any computational constraints, the previous observation would favor approaches with the largest values of $K$ and $M$. However, different values of $(K, M)$ lead to different computational costs, as measured here by FLOPs, with $(K, M) = (1, 1)$ being the cheapest. Figure 2b shows, as expected, that the number of FLOPs grows more

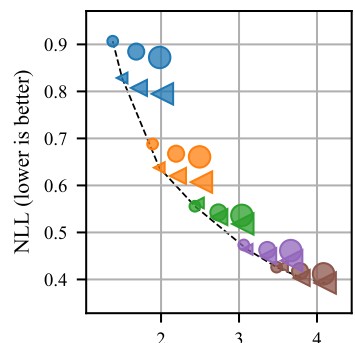

Figure 3: ImageNet evaluation for ViT (●) and V-MoE ($K = 1$) (◄) ensembles of size $\{1, 2, 4\}$ (◆,◆,◆). Model sizes: S/32 (◆), B/32 (◆), L/32 (◆), L/16 (◆), and H/14 (◆). ViT and V-MoE benefit from ensembling equally, at all scales.

quickly along the $M$ axis than along the $K$ axis. To capture the various trade-offs at play, in Figure 2c we report the logarithm of the normalized gains in log likelihood $\frac{\text{LL}_{(K,M)} - \text{LL}_{(1,1)}}{\text{GFLOPs}_{(K,M)} - \text{GFLOPs}_{(1,1)}}$ when going from $(K, M) = (1, 1)$ to other choices of $(K, M)$. Interestingly, it appears more advantageous to first grow $K$, i.e., the adaptive combination, before growing $M$.

**Summary.** While simple ensembling of sparse MoEs results in strong predictive performance, we lose their computational efficiency. We next show how to efficiently combine ensembling and sparse MoEs, exploiting the fact that statically combining sparse MoEs with fewer experts remains effective.

## 4 Efficient Ensemble of Experts

Equipped with the insights from Section 3, we describe efficient ensemble of experts ($\text{E}^3$), with the goal of keeping the strengths of both sparse MoEs and ensembles. Conceptually, $\text{E}^3$ jointly learns an ensemble of smaller sparse MoEs, where all layers without experts (e.g., attention layers) are shared across the members.

### 4.1 The Architecture

There are two main components in $\text{E}^3$:

**Disjoint subsets of experts.** We change the structure of (1) by *partitioning* the set of $E$ experts into $M$ subsets of $E/M$ experts (assuming that $E$ is a multiple of $M$). We denote the subsets by $\mathcal{E}_m$. For example, $\mathcal{E}_1 = \{1, 2, 3\}$ and $\mathcal{E}_2 = \{4, 5, 6\}$ for $E = 6$ and $M = 2$. Intuitively, the ensemble members have separate parameters for independent predictions, while efficiently sharing parameters among all non-expert layers. Instead of having a single routing function $\text{gate}_K(\boldsymbol{W}\cdot)$ as in (1), we apply separate routing functions $\text{gate}_K(\boldsymbol{W}_m\cdot)$ to each subset $\mathcal{E}_m$. Note that this does not affect the total number of parameters since $\boldsymbol{W}$ has $E$ rows while each $\boldsymbol{W}_m$ has $E/M$ rows. A similar partitioning of the experts was done in Yang et al. (2021) but not exploited to create different ensemble members, in particular not in conjunction with tiled representations, which we show to be required to get performance gains (see Section 4.2.1).

**Tiled representation.** To jointly handle the predictions of the $M$ ensemble members, we tile the inputs by a factor $M$, inspired by Wen et al. (2019). This enables a simple implementation of $\text{E}^3$ on top of an existing MoE. In Appendix E, we connect sparse MoEs and BE, illustrating that tiling naturally fits into the formalism of sparse MoEs. Because of the tiling, a given image patch has $M$ different representations that, when entering an MoE layer, are each routed to their respective expert subsets $\mathcal{E}_m$. Formally, consider some tiled inputs $\boldsymbol{H} \in \mathbb{R}^{B \times M \times D}$ where $B$ refers to the batch size (a batch contains image patches) and $\boldsymbol{h}_{i,m} \in \mathbb{R}^D$ is the representation of the $i$-th input for the $m$-th member. The routing logic in $\text{E}^3$ can be written as

$$\text{p-MoE}(\boldsymbol{h}_{i,m}) = \sum_{e \in \mathcal{E}_m} g_e(\boldsymbol{h}_{i,m}) \cdot \text{MLP}_e(\boldsymbol{h}_{i,m}) \quad \text{with} \quad \{g_e(\boldsymbol{h}_{i,m})\}_{e \in \mathcal{E}_m} = \text{gate}_K(\boldsymbol{W}_m \boldsymbol{h}_{i,m}), \quad (2)$$

where the routing weights are now $\boldsymbol{W}_m \in \mathbb{R}^{(E/M) \times D}$; see Figure 1.

To echo the observations from Section 3, we can first see that $\text{E}^3$ brings together the static and adaptive combination of ensembles and sparse MoEs, which we found to be complementary. However, we have seen that static ensembling comes at the cost of a large increase in FLOPs, thus we opt for an efficient ensembling approach. Second, we "split" the MoE layers along the axis of the experts, i.e., from $E$ experts to $M$ times $E/M$ experts. We do so since we observed that the performance of sparse MoEs tends to plateau quickly for more experts. We note that $\text{E}^3$ retains the property of ensembles to output multiple predictions per input.

In a generic implementation, we tile a batch of $B$ inputs $\boldsymbol{X} \in \mathbb{R}^{B \times P}$ by a factor $M$ to obtain the tiled inputs $\boldsymbol{X}_{\text{tiled}} = [\boldsymbol{X}; \ldots; \boldsymbol{X}] \in \mathbb{R}^{(M \cdot B) \times P}$ and the model processes $f(\boldsymbol{X}_{\text{tiled}}; \boldsymbol{\theta})$. Since tiling in $\text{E}^3$ has an effect only from the first MoE layer onwards, we postpone the tiling operation to that stage, thus saving otherwise redundant prior computations in non-MoE-layers. For example, for L/16 and $K = M = 2$, we can save about 47% of the FLOPs. Further implementation details of $\text{E}^3$, and a discussion of the increased memory consumption due to tiling, are in Appendix C. Code can be found at https://github.com/google-research/vmoe. Finally, we note that although $\text{E}^3$ and BE share conceptual design similarities—tiled representation and sharing of parameters—they differ in fundamental structural ways, see Appendix F.

### 4.2 Ablation Studies: Partitioning and Tiling

Our method introduces two changes to V-MoEs: (a) the partitioning of the experts and (b) the tiling of the representations. In this section, we assess the separate impact of each of those changes and show that

Table 2: ImageNet performance (means $\pm$ standard errors over 8 seeds) of $\mathrm{E}^3$-B/32 ($K = M = 2$), V-MoE ($K = 4$), and two ablations: *only* tiling and *only* partitioning. The noise in $\mathtt{gate}_K$ is denoted by $\sigma$.

| | NLL ↓ | ERROR ↓ | ECE ↓ | KL ↑ |
|---|---|---|---|---|
| V-MoE | 0.636 $_{\pm\ 0.001}$ | 16.70 $_{\pm\ 0.04}$ | 0.034 $_{\pm\ 0.001}$ | — |
| $\mathrm{E}^3$ | **0.612** $_{\pm\ 0.001}$ | **16.49** $_{\pm\ 0.02}$ | **0.013** $_{\pm\ 0.000}$ | **0.198** $_{\pm\ 0.003}$ |
| Tiling | 0.637 $_{\pm\ 0.002}$ | 16.74 $_{\pm\ 0.06}$ | 0.028 $_{\pm\ 0.001}$ | 0.000 $_{\pm\ 0.000}$ |
| Tiling ($\sigma \times 2$) | 0.638 $_{\pm\ 0.001}$ | 16.72 $_{\pm\ 0.03}$ | 0.033 $_{\pm\ 0.001}$ | 0.001 $_{\pm\ 0.000}$ |
| Tiling ($\sigma \times 4$) | 0.638 $_{\pm\ 0.001}$ | 16.74 $_{\pm\ 0.03}$ | 0.033 $_{\pm\ 0.001}$ | 0.002 $_{\pm\ 0.000}$ |
| Partitioning | 0.640 $_{\pm\ 0.001}$ | 16.72 $_{\pm\ 0.05}$ | 0.034 $_{\pm\ 0.001}$ | — |

it is indeed their combination that explains the performance gains. We summarize the results of this study in Table 2 which shows ImageNet performance—negative log likelihood (NLL), classification error, expected calibration error (ECE) (Guo et al., 2017), and Kullback–Leibler divergence—for different ablations of $\mathrm{E}^3$. See Appendix N, for end-to-end overview diagrams in the style of Figure 1, for the these ablations. We use $E = 32$ experts. We provide FLOP measurements for these ablations in Appendix M.1.

### 4.2.1 Partitioning without Tiling

We first compare $\mathrm{E}^3$ with a variant of V-MoE where we only partition the set of experts (*Partitioning*). In this variant, each input $\boldsymbol{h}_i \in \mathbb{R}^D$ (note the dropping of the index $m$ due to the absence of tiling) can select $K$ experts in subset $\mathcal{E}_m$, resulting in a total of $K \times M$ selected experts per input. Formally, (2) becomes

$$\mathtt{partitioning\text{-}only\text{-}MoE}(\boldsymbol{h}_i) = \sum_{m=1}^{M} \sum_{e \in \mathcal{E}_m} g_e(\boldsymbol{h}_i) \cdot \mathtt{MLP}_e(\boldsymbol{h}_i) \quad \text{with} \quad \{g_e(\boldsymbol{h}_i)\}_{e \in \mathcal{E}_m} = \mathtt{gate}_K(\boldsymbol{W}_m \boldsymbol{h}_i).$$

The *expert prototyping* of Yang et al. (2021) leads to a similar formulation. As shown in Table 2, across all metrics, *Partitioning* is not competitive with $\mathrm{E}^3$. We do not report KL since without tiling, *Partitioning* does not output multiple predictions per input.

### 4.2.2 Tiling without Partitioning

We now compare $\mathrm{E}^3$ with the variant where only the tiling is enabled (*Tiling*). In this case, we have tiled inputs $\boldsymbol{H} \in \mathbb{R}^{B \times M \times D}$ applied to the standard formulation of (1). Compared with (2), there is no mechanism to enforce the $M$ representations of the $i$-th input across the ensemble members, i.e., $\{\mathtt{MoE}(\boldsymbol{h}_{i,m})\}_{m=1}^{M}$, to be different. Indeed, without partitioning, each $\boldsymbol{h}_{i,m}$ could select $K$ identical experts. As a result, we expect *Tiling* to output $M$ similar predictions across ensemble members. This is confirmed in Table 2 where we observe that the KL for *Tiling* is orders of magnitude smaller than for $\mathrm{E}^3$. To mitigate this effect, we also tried to increase the level of noise $\sigma$ in $\mathtt{gate}_K$ (by a factor $\{2, 4\}$), to cause the expert assignments to differ across $\{\boldsymbol{h}_{i,m}\}_{m=1}^{M}$. While we do see an increase in KL, *Tiling* still performs worse than $\mathrm{E}^3$ across all metrics.

Interestingly, we can interpret *Tiling* as an approximation, via $M$ samples, of the marginalization $\mathbb{E}_{\varepsilon_1,\ldots,\varepsilon_\ell}[f(\boldsymbol{x}; \boldsymbol{\theta})]$ with respect to the noise $\{\varepsilon_l\}_{l=1}^{\ell}$ in the $\ell$ MoE layers of $f(\cdot; \boldsymbol{\theta})$ (further assuming the capacity constraints of the experts, as described in Riquelme et al. (2021), do not bias the $M$ samples).

### 4.2.3 Tiling with Increasing Parameter Sharing

The results in Table 2 (as well as Appendix I) suggest that the strong performance of $\mathrm{E}^3$ is related to its high-diversity predictions. We hypothesise that this diversity is a result of the large number of non-shared parameters in each ensemble member, i.e., the partitioning of the experts. To test this hypothesis, we allow the subsets $\mathcal{E}_m$ in $\mathrm{E}^3$ to have *some degree of overlap* (i.e., ensemble members share some experts), thus interpolating between $\mathrm{E}^3$ and *Tiling*. For example, with total experts $E = 32$ and an ensemble size $M = 2$, an overlap of 8 shared experts means that each ensemble member has $(32 - 8)/2 = 12$ unique experts, and $12 + 8 = 20$ in total. Table 3 shows that increasing the number of shared experts directly decreases diversity and thus NLL, Error, and ECE. We see the same trends for $K = 1$ (rather than $K = 2$; see Table 18).

Table 3: ImageNet performance (means ± standard errors over 8 seeds) of $\text{E}^3$-B/32 ($K = M = 2$), $E = 32$ total experts, and varying expert overlap between subsets $\mathcal{E}_m$.

| OVERLAP | NLL ↓ | ERROR ↓ | ECE ↓ | KL ↑ |
|---|---|---|---|---|
| 0 (=$\text{E}^3$) | **0.612** $_{\pm\ 0.001}$ | **16.49** $_{\pm\ 0.02}$ | **0.013** $_{\pm\ 0.000}$ | **0.198** $_{\pm\ 0.003}$ |
| 2 | 0.617 $_{\pm\ 0.003}$ | 16.55 $_{\pm\ 0.09}$ | 0.016 $_{\pm\ 0.001}$ | 0.167 $_{\pm\ 0.005}$ |
| 4 | 0.622 $_{\pm\ 0.001}$ | 16.62 $_{\pm\ 0.02}$ | 0.017 $_{\pm\ 0.001}$ | 0.148 $_{\pm\ 0.003}$ |
| 8 | 0.627 $_{\pm\ 0.002}$ | 16.67 $_{\pm\ 0.07}$ | 0.021 $_{\pm\ 0.001}$ | 0.122 $_{\pm\ 0.010}$ |
| 16 | 0.639 $_{\pm\ 0.004}$ | 16.82 $_{\pm\ 0.07}$ | 0.030 $_{\pm\ 0.003}$ | 0.077 $_{\pm\ 0.011}$ |
| 32 (=Tiling) | 0.637 $_{\pm\ 0.002}$ | 16.74 $_{\pm\ 0.06}$ | 0.028 $_{\pm\ 0.001}$ | 0.000 $_{\pm\ 0.000}$ |

Table 4: ImageNet performance (means ± standard errors over 8 seeds) of V-MoE-B/32 and a simple multi-prediction variant (*Multi-pred*) whose last MoE layer is changed as in (3).

| | K | NLL ↓ | ERROR ↓ | ECE ↓ | KL ↑ |
|---|---|---|---|---|---|
| $\text{E}^3$ ($M = 2$) | 1 | **0.622** $_{\pm\ 0.001}$ | **16.70** $_{\pm\ 0.03}$ | **0.018** $_{\pm\ 0.000}$ | **0.217** $_{\pm\ 0.003}$ |
| Multi-pred | 2 | 0.636 $_{\pm\ 0.001}$ | 17.16 $_{\pm\ 0.02}$ | 0.024 $_{\pm\ 0.000}$ | 0.032 $_{\pm\ 0.001}$ |
| V-MoE | 2 | 0.638 $_{\pm\ 0.001}$ | **16.76** $_{\pm\ 0.05}$ | 0.033 $_{\pm\ 0.001}$ | — |
| $\text{E}^3$ ($M = 2$) | 2 | **0.612** $_{\pm\ 0.001}$ | **16.49** $_{\pm\ 0.02}$ | **0.013** $_{\pm\ 0.000}$ | **0.198** $_{\pm\ 0.003}$ |
| Multi-pred | 4 | 0.645 $_{\pm\ 0.001}$ | 17.39 $_{\pm\ 0.04}$ | 0.021 $_{\pm\ 0.000}$ | 0.011 $_{\pm\ 0.001}$ |
| V-MoE | 4 | 0.636 $_{\pm\ 0.001}$ | 16.70 $_{\pm\ 0.04}$ | 0.034 $_{\pm\ 0.001}$ | — |
| $\text{E}^3$ ($M = 2$) | 4 | **0.611** $_{\pm\ 0.001}$ | **16.45** $_{\pm\ 0.03}$ | **0.014** $_{\pm\ 0.000}$ | **0.193** $_{\pm\ 0.003}$ |
| Multi-pred | 8 | 0.650 $_{\pm\ 0.001}$ | 17.50 $_{\pm\ 0.03}$ | 0.021 $_{\pm\ 0.000}$ | 0.005 $_{\pm\ 0.000}$ |
| V-MoE | 8 | 0.635 $_{\pm\ 0.002}$ | 16.72 $_{\pm\ 0.06}$ | 0.028 $_{\pm\ 0.001}$ | — |

#### 4.2.4 Multiple Predictions without Tiling or Partitioning

As highlighted in Table 1, an ensemble of size $M$ outputs $M$ predictions for a given input (thereafter, averaged) while sparse MoEs only produce a single prediction. Thus, a natural question is how much the gains of $\text{E}^3$ are simply due to its ability to produce multiple predictions, rather than its specific tiling and partitioning mechanisms? To answer this, we investigate a simple multi-prediction variant of sparse MoEs (*Multi-pred*) wherein the last MoE layer of the form (1) is replaced by

$$\texttt{Multi-pred-MoE}(\boldsymbol{h}) = \{g_e(\boldsymbol{h}) \cdot \texttt{MLP}_e(\boldsymbol{h})\}_{g_e(\boldsymbol{h})>0} \in \mathbb{R}^{K \times Q} \quad \text{with} \quad \{g_e(\boldsymbol{h})\}_{e=1}^E = \texttt{gate}_K(\boldsymbol{Wh}), \qquad (3)$$

where we have assumed $\texttt{MLP}_e(\boldsymbol{h}) \in \mathbb{R}^Q$. Instead of *summing* the expert outputs like in (1), we *stack* the $K$ selected expert contributions (as a reminder, $\texttt{gate}_K$ zeroes out the $E - K$ smallest weights). Keeping track of those $K$ contributions makes it possible to generate $K$ different predictions per input as in the classifier of Figure 1, thus aiming at capturing model uncertainty around the true prediction.

Table 4 compares the ImageNet performance of this simple multiple-prediction method with the standard V-MoE and $\text{E}^3$. In all cases, *Multi-pred* under performs relative to $\text{E}^3$. Indeed, despite improvements in ECE, it is only for $K = 2$, that *Multi-pred* provides small gains in NLL relative to V-MoE, while its classification error is always worse. In fact, *Multi-pred* for $K = 4$ performs *worse* in terms of NLL, classification error, and diversity than for $K = 2$. The KL diversity metric indicates that the *Multi-pred* is unable to provide diverse predictions. This indicates that it is specifically tiling and partitioning in $\text{E}^3$ that provide good performance.

### 4.3 Comparison with Alternative Efficient Ensembling Strategies

In the previous subsection we saw that a simple approach to multiple predictions in a V-MoE model is unable to achieve good diversity in predictions and thus strong predictive performance. Following Havasi et al. (2020); Soflaei et al. (2020), a possible fix to this problem would be to have a multi-prediction *and multi-input* approach. Furthermore, other efficient ensembling strategies could provide alternative solutions to this problem. Unfortunately, as we show in Table 5, while common efficient ensembling strategies such as BE (Wen et al., 2019), MC Dropout (Gal & Ghahramani, 2016), and MIMO (Havasi et al., 2020), do improve slightly on ViT/V-MoE, they are unable to match the performance of $\text{E}^3$. In Appendix G, we provide

Table 5: ImageNet performance (means $\pm$ standard errors over 8 seeds) of different efficient ensemble approaches based on a ViT-B/32 architecture.

| | K | M | NLL ↓ | ERROR ↓ | ECE ↓ | KL ↑ | GFLOPs ↓ |
|---|---|---|---|---|---|---|---|
| ViT | – | – | 0.688 $_{\pm\,0.003}$ | 18.65 $_{\pm\,0.08}$ | 0.022 $_{\pm\,0.000}$ | – | **78.0** |
| BE ViT | – | 2 | 0.682 $_{\pm\,0.003}$ | 18.47 $_{\pm\,0.05}$ | 0.021 $_{\pm\,0.000}$ | 0.040 $_{\pm\,0.001}$ | 97.1 |
| V-MoE | 2 | – | 0.638 $_{\pm\,0.001}$ | **16.76** $_{\pm\,0.05}$ | 0.033 $_{\pm\,0.001}$ | – | 94.9 |
| MC Dropout V-MoE | 1 | 2 | 0.648 $_{\pm\,0.002}$ | 17.10 $_{\pm\,0.05}$ | **0.019** $_{\pm\,0.001}$ | 0.046 $_{\pm\,0.000}$ | 97.2 |
| MIMO V-MoE | 2 | 2 | 0.636 $_{\pm\,0.002}$ | 16.97 $_{\pm\,0.04}$ | 0.028 $_{\pm\,0.001}$ | 0.000 $_{\pm\,0.000}$ | 96.3 |
| | 2 | 4 | 0.672 $_{\pm\,0.001}$ | 17.72 $_{\pm\,0.04}$ | 0.037 $_{\pm\,0.000}$ | 0.001 $_{\pm\,0.000}$ | 99.0 |
| $\mathrm{E}^3$ | 1 | 2 | **0.622** $_{\pm\,0.001}$ | **16.70** $_{\pm\,0.03}$ | **0.018** $_{\pm\,0.000}$ | **0.217** $_{\pm\,0.003}$ | 105.9 |

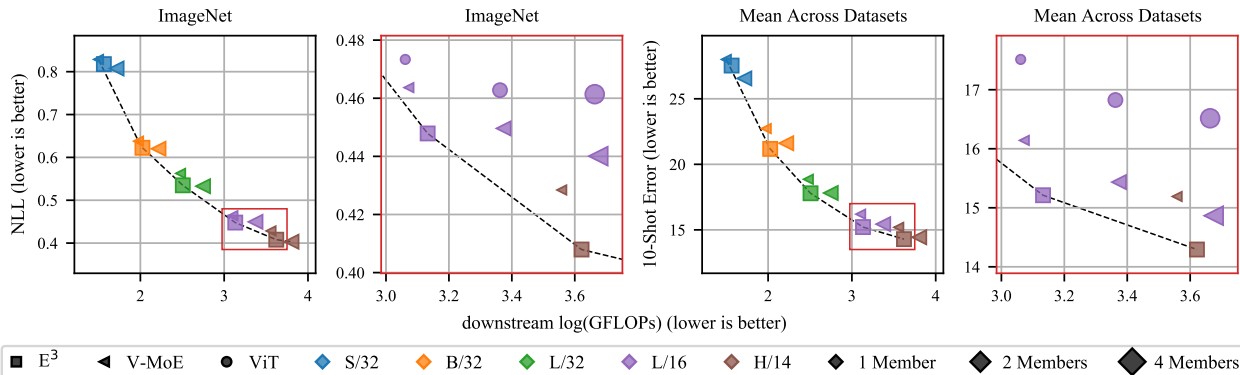

Figure 4: ImageNet NLL (left, center left) and mean 10-shot error across datasets (center right, right), with zoomed-in plots of highlighted areas. Zoomed-in plots include additional ensemble baselines for comparison. Dashed lines show Pareto frontiers which tend to be defined by $\mathrm{E}^3$.

a detailed description of how we carefully apply these methods to ViT/V-MoE to ensure a fair evaluation (sometimes even designing extensions of these methods). We also give results for additional $K$ and $M$ values.

## 5 Evaluation

We now benchmark $\mathrm{E}^3$ against V-MoE. As a baseline we also include results for *downstream* ensembles of V-MoE and ViT. These ensembles offer a natural baseline against $\mathrm{E}^3$ as they also use a single upstream checkpoint, are easy to implement, and provide consistent improvements upon V-MoE. In Appendix J, we compare with *upstream* ensembles that require multiple upstream checkpoints (Mustafa et al., 2020). All results correspond to the average over 8 (for {S, B, L} single models) or 5 (for H single models and all up/downstream ensembles) replications. In Appendices L and M we provide results for additional datasets and metrics as well as standard errors. Following Riquelme et al. (2021), we compare the predictive-performance vs. compute cost trade-offs for each method across a range of ViT families. In the results below, $\mathrm{E}^3$ uses $(K, M) = (1, 2)$, single V-MoE models use $K = 2$, V-MoE ensembles use $K = 1$, and all use $E = 32$. Experimental details, including those for our upstream training, downstream fine-tuning, hyperparameter sweeps and (linear) few-shot evaluation can be found in Appendix A. Our main findings are as follows:

### 5.1 V-MoE vs. ViT

• **Ensembles help V-MoE just as much as ViT.** Ensembling was expected to benefit ViT. However, Figures 3 to 8 suggest that ensembling provides similar gains for V-MoE in terms of few-shot performance, NLL, ECE, OOD detection, and robustness to distribution shift. We believe this has not been observed before. Moreover, a downstream ensemble with four H/14 V-MoEs leads to a 88.8% accuracy on ImageNet (even reaching an impressive 89.3% for an upstream ensemble that further benefits from multiple upstream checkpoints, see Table 15).

• **ViT consistently provides better ECE than V-MoE.** Surprisingly, despite V-MoE tending to have

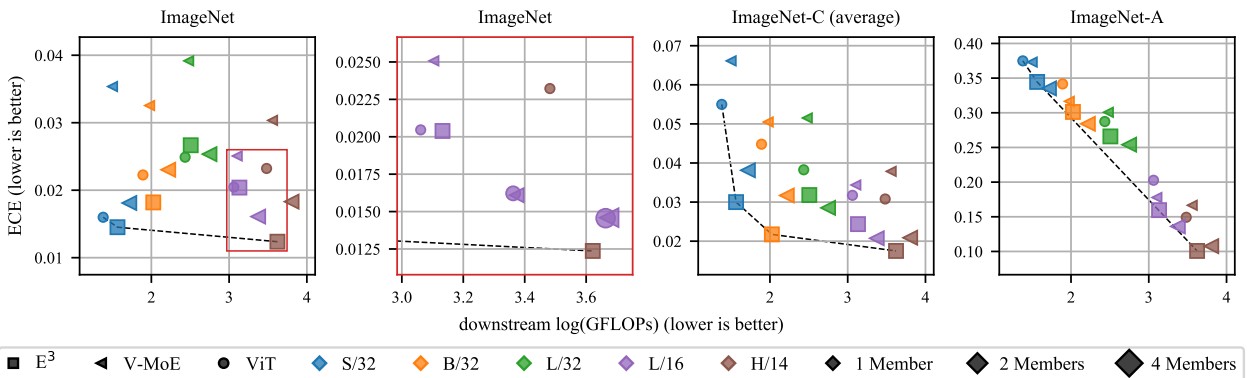

Figure 5: ECE for ImageNet (left), with a zoomed-in plot of the highlighted area (center left), and under distribution shift (right, center right). This is a metric for which ensembles are known to perform well whereas, to the best of our knowledge, the performance of V-MoE has not been evaluated. The zoomed-in plot includes additional ensemble baselines for comparison. Dashed lines show Pareto frontiers.

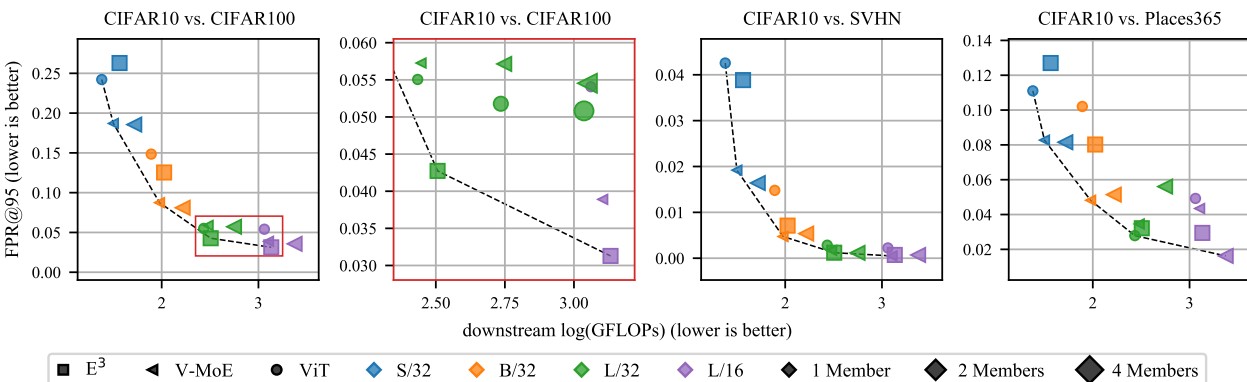

Figure 6: OOD detection, measured by false positive rate at 95% precision (Fort et al., 2021), for models fine-tuned on CIFAR10, with a zoomed-in plot of the highlighted area. This is a metric for which ensembles are known to perform well whereas, to the best of our knowledge, the performance of V-MoE has not been evaluated. The zoomed-in plot includes additional ensemble baselines for comparison. Dashed lines show Pareto frontiers.

better NLL than ViT (Figure 3), their ECE is worse (Figure 5).

• **ECE is not consistent for different ViT/V-MoE families.** We see the ECE, unlike other metrics presented in this work, tends not to provide consistent trends as we increase the ViT family size (Figure 5). This observation is consistent with Minderer et al. (2021), who noted similar behaviour for a range of models. They note that post-hoc temperature scaling can improve consistency.

• **V-MoE outperforms ViT in OOD detection.** With L/32 being the only exception, V-MoE outperforms ViT on a range of OOD detection tasks (Figure 6). While this may seem surprising, given the opposite trend for ECE, it suggests that ViT makes more accurate predictions about the scale of the uncertainty estimates while V-MoE makes better predictions about the relative ordering of the uncertainty estimates.

• **For smaller ViT families, V-MoE outperforms ViT in the presence of distribution shift.** In contrast to the OOD detection results, Figure 7 shows that for smaller ViT families V-MoE improves on the performance of ViT. However, as the ViT family becomes larger, this trend is less consistent.

## 5.2 Efficient Ensemble of Experts

• **E³ improves classification performance.** As shown in Figure 4, E³ is either on, or very near to, the Pareto frontiers for NLL and 10-shot classification error, despite the fact that these are metrics for which ensembles and V-MoE, respectively, are known to perform well. Figure 8 shows that similar conclusions hold for CIFAR10/100 NLL.

• **E³ performs best at the largest scale.** The difference in predictive performance between E³ and V-

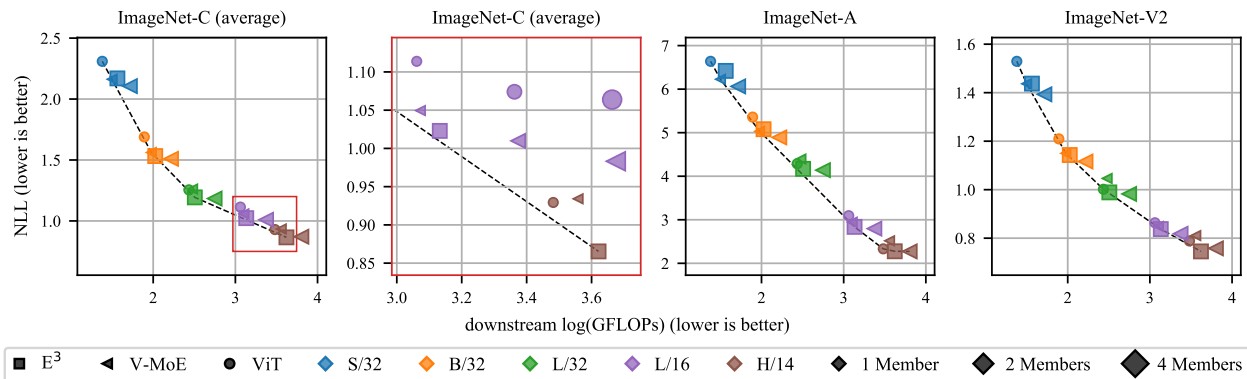

Figure 7: NLL under distribution shift for models trained on ImageNet. For ImageNet-C, we provide a zoomed-in plot of the highlighted area. The zoomed-in plot includes additional ensemble baselines for comparison. Dashed lines show Pareto frontiers which tend to be defined by E³.

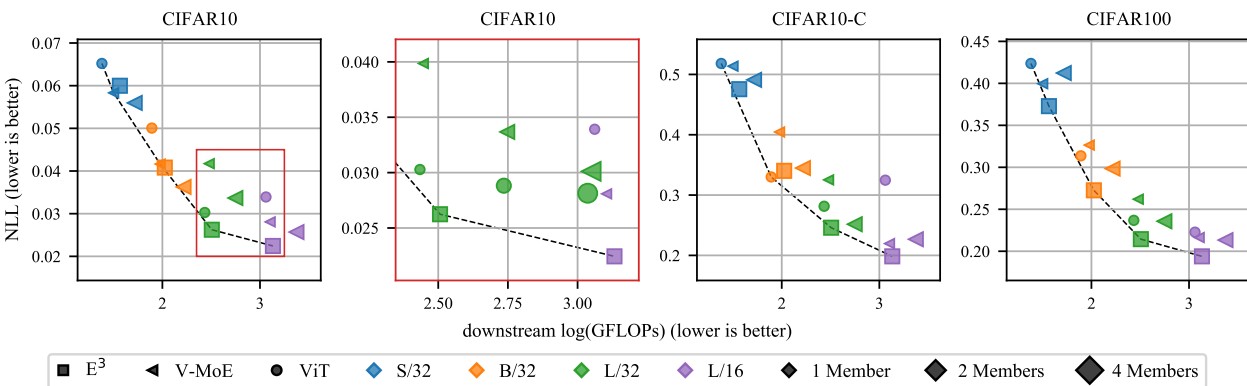

Figure 8: NLL for CIFAR10 (left), with a zoomed-in plot of the highlighted area (center left), CIFAR10-C (center right), and CIFAR100 (right). The zoomed-in plot includes additional ensemble baselines for comparison. Dashed lines show Pareto frontiers.

MoE—or ensembles thereof—increases as the ViT family becomes larger (Figures 4 to 8, and Appendix D where we propose a scheme to normalize performances across scales).

• **E³ becomes Pareto efficient for larger ViT families in the presence of distribution shift.** Figure 7 shows that E³ is more robust to distribution shift for larger ViT families, despite less consistent V-MoE performance at scale. When averaged over the shifted datasets (ImageNet-C, ImageNet-A, ImageNet-V2), E³ improves on V-MoE in terms of NLL for all ViT families other than S/32, with improvements up to 8.33% at the largest scale; see Appendix M.3.

• **E³ improves ECE over ViT and V-MoE.** Despite V-MoE providing poor ECE, E³ does not suffer from this limitation (Figure 5). For most ViT families, E³ also provides better ECE than V-MoE ensembles.

• **E³ does not provide consistent OOD detection performance.** Firstly, Figure 6 shows that for small ViT families, E³ performs worse than V-MoE and (even ViT in some cases). Nevertheless, as above, the relative performance improves for larger ViT families such that E³ becomes Pareto efficient for two dataset pairs. Secondly, E³ seems to perform better on the more difficult near OOD detection task (CIFAR10 vs. CIFAR100). These results, although sometimes subtle, are consistent across OOD detection metrics and dataset pairs, as shown in Appendix L.

**Summary.** While no single model performs best in all of our evaluation settings, we do find that E³ performs well and is Pareto efficient in *most* cases. This is particularly true for the larger ViT families. One exception was OOD detection, where E³ performance was somewhat inconsistent. On the other hand, while V-MoE clearly outperforms ViT in terms of accuracy, the uncertainty estimates can be better or worse depending on the downstream application.

## 6 Related Work

**Mixture of Experts.** MoEs (Jacobs et al., 1991; Jordan & Jacobs, 1994; Chen et al., 1999; Yuksel et al., 2012; Eigen et al., 2014) combine the outputs of different submodels, or *experts*, in an input-dependent way. Sparse MoEs only select a few experts per input, enabling to greatly scale models while keeping the prediction time constant. Sparse MoEs have been used to build large language models (Shazeer et al., 2017; Lepikhin et al., 2021; Fedus et al., 2021). Recently, sparse MoEs have been also successfully applied to vision problems (Riquelme et al., 2021; Yang et al., 2021; Lou et al., 2021; Xue et al., 2021). Our work builds on the V-MoE architecture proposed by Riquelme et al. (2021), which is based on the vision Transformer (ViT) (Dosovitskiy et al., 2021) and showed improved performance for the same computational cost as ViT. We explore the interplay between sparse MoEs and ensembles and show that V-MoEs benefit from ensembling, by improving their predictive performance and robustness. While previous work studied ViT's calibration and robustness (Minderer et al., 2021; Fort et al., 2021; Paul & Chen, 2021; Mao et al., 2021), we are the first to study the robustness of V-MoE models.

**Ensembles.** Ensemble methods combine several different models to improve generalization and uncertainty estimation. Ensembles achieve the best performance when they are composed of diverse members that make complementary errors (Hansen & Salamon, 1990; Fort et al., 2019; Wenzel et al., 2020; D'Angelo & Fortuin, 2021; Lopes et al., 2021). However, standard ensembles are inefficient since they consist of multiple models, each of which can already be expensive. To reduce test time, Xie et al. (2013) and Hinton et al. (2015); Tran et al. (2020); Nam et al. (2021) use compression and distillation mechanisms, respectively. To reduce training time, ensembles can be constructed with cyclical learning-rate schedules to snapshot models along the training trajectory (Huang et al., 2017; Zhang et al., 2019). Our work builds on batch ensemble (Wen et al., 2019) where a *single* model encapsulates an ensemble of networks, a strategy also explored by Lee et al. (2015); Havasi et al. (2020); Antorán et al. (2020); Dusenberry et al. (2020a); Rame et al. (2021). Wenzel et al. (2020) extended BE to models with different hyperparameters.

**Bayesian Neural Networks.** BNNs are an alternative approach to deep ensembles for uncertainty quantification in neural networks. In a BNN the weights are treated as random variables and the uncertainty in the weights is translated to uncertainty in predictions via marginalisation. However, because the weight posterior is intractable for NNs, approximation is required. Popular approximations include variational inference (Hinton & Van Camp, 1993; Graves, 2011; Blundell et al., 2015), the Laplace approximation (MacKay, 1992; Ritter et al., 2018), and MC Dropout (Gal & Ghahramani, 2016). However, many of these approximations make restrictive mean-field assumptions which hurt performance (Foong et al., 2020; Coker et al., 2022; Fortuin et al., 2021). Unfortunately, modeling full weight correlations for even small ResNets—with relatively few parameters compared to the ViT models considered here—is intractable (Daxberger et al., 2021).

## 7 Conclusions and Future Work

Our study of the interplay between sparse MoEs and ensembles has shown that these two classes of models are symbiotic. Efficient ensemble of experts exemplifies those mutual benefits—as illustrated by its accuracy, log-likelihood, few-shot learning, robustness, and uncertainty calibration improvements over several challenging baselines in a range of benchmarks. We have also provided the first, to the best of our knowledge, investigation into the robustness and uncertainty calibration properties of V-MoEs—showing that these models are robust to dataset shift and are able to detect OOD examples. While our study has focused on downstream fine-tuned models, we believe that an extension to the upstream case and from-scratch training, would also result in a fruitful investigation. In fact, in Appendix K we provide some promising, but preliminary, results for from-scratch training. Similarly, although we have focused on computer vision, our approach should be readily applicable to the modelling of text, where sparse MoEs have been shown to be remarkably effective. With the growing prevalence of sparse MoEs in NLP (Patterson et al., 2021), the questions of understanding and improving the robustness and reliability of such models become increasingly important. Furthermore, the computational scale at which those models operate make those questions challenging to tackle. We believe that our study, and approaches such as $E^3$, make steps in those directions.

## Acknowledgements

We would like to extend our gratitude to Josip Djolonga for having reviewed an earlier version of the draft and for helping us with `robustness_metrics` (Djolonga et al., 2020). We would also like to thank André Susano Pinto and Cedric Renggli for fruitful comments on the project, as well as Jakob Uszkoreit for insightful discussions. JUA acknowledges funding from the EPSRC, the Michael E. Fisher Studentship in Machine Learning, and the Qualcomm Innovation Fellowship. VF acknowledges funding from the Swiss Data Science Center through a PhD Fellowship, as well as from the Swiss National Science Foundation through a Postdoc.Mobility Fellowship, and from St. John's College Cambridge through a Junior Research Fellowship.

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

## Appendix Structure

This appendix is structured as follows:

- We provide experimental details, such as model specifications, upstream and downstream training hyperparameters, and few-shot evaluation settings, in Appendix A.

- We discuss the compatibility and adaption of upstream V-MoE and ViT checkpoints to the various models considered in this paper, in Appendix B.

- We provide implementation details for $E^3$, in Appendix C.

- We show that $E^3$ provides larger relative improvements versus V-MoE for larger models, in Appendix D.

- We draw connections between BE and sparse MoEs, in Appendix E.

- We highlight differences between $E^3$ and BE, in Appendix F.

- We compare empirical results for $E^3$ and various other efficient ensemble approaches applied to ViT and V-MoE, in Appendix G.

- We provide sensitivity analyses for the load balancing loss strength, expert initialisation diversity, and expert group permutation, in Appendix H.

- We investigate the roles of diversity and individual model performance in overall ensemble performance, in Appendix I.

- We compare ensembling only downstream with ensembling that takes place both up and downstream, in Appendix J.

- We provide preliminary results for training $E^3$ on ImageNet from scratch, without any pre-training, in Appendix K.

- We extend the results of Sections 3 to 5 to additional datasets, metrics, model sizes, and other settings, in Appendix L.

- We provide a range of tabular results, including flops numbers, parameter counts, and standard errors for the experiments in Section 5, in Appendix M.

- We provide additional overview diagrams, in the style of Figure 1, for the various algorithms and ablations presented in the main text, in Appendix N.

## A    Experiment Settings

### A.1    ViT Model Specifications

Following Dosovitskiy et al. (2021), we recall the specifications of the ViT models of different scales in Table 6.

Table 6: Specifications of ViT-S, ViT-B, ViT-L and ViT-H.

|  | HIDDEN DIMENSION | MLP DIMENSION | # LAYERS |
|---|---|---|---|
| Small | 512 | 2048 | 8 |
| Base | 768 | 3072 | 12 |
| Large | 1024 | 4096 | 24 |
| Huge | 1280 | 5144 | 32 |

### A.2 Upstream Setting

For all our upstream experiments, we scrupulously follow the setting described in Riquelme et al. (2021), see their Section B.2 in their appendix. For completeness, we just recall that S/32 models are trained for 5 epochs while B/{16, 32} and L/32 models are trained for 7 epochs. For L/16 models, both 7 and 14 epochs can be considered (Dosovitskiy et al., 2021; Riquelme et al., 2021); we opted for 7 epochs given the breadth of our experiments. Finally, the H/14 model are trained for 14 epochs.

In particular, the models are all trained on JFT-300M (Sun et al., 2017). This dataset contains about 305M training and 50 000 validation images. The labels have a hierarchical structure, with a total of 18 291 classes, leading on average to 1.89 labels per image.

### A.3 Downstream Setting

During fine-tuning, there is a number of *common* design choices we apply. In particular:

- Image resolution: 384.

- Clipping gradient norm at: 10.0.

- Optimizer: SGD with momentum (using half-precision, $\beta = 0.9$).

- Batch size: 512.

- For V-MoE models, we finetune with capacity ratio $C = 1.5$ and evaluate with $C = 8$.

We use the following train/validation splits depending on the dataset:

| Dataset | Train Dataset Fraction | Validation Dataset Fraction |
|---|---|---|
| ImageNet | 99% | 1% |
| CIFAR10 | 98% | 2% |
| CIFAR100 | 98% | 2% |
| Oxford-IIIT Pets | 90% | 10% |
| Oxford Flowers-102 | 90% | 10% |

All those design choices follow from Riquelme et al. (2021) and Dosovitskiy et al. (2021).

### A.4 Hyperparameter Sweep for Fine-tuning

In all our fine-tuning experiments, we use the sweep of hyperparameters described in Table 7. We use the recommendations from Dosovitskiy et al. (2021) and Riquelme et al. (2021), further considering several factors {0.5, 1.0, 1.5, 2.0} to sweep over different numbers of steps. Riquelme et al. (2021) use a half schedule (with the factor 0.5) while Dosovitskiy et al. (2021) take the factor 1.0.

We show in Table 8 the impact of this enlarged sweep of hyperparameters in the light of the results reported in Riquelme et al. (2021). We notably tend to improve the performance of ViT and V-MoE (especially for smaller models), which thus makes the baselines we compare to more competitive.

### A.5 Details about the (Linear) Few-shot Evaluation

We follow the evaluation methodology proposed by Dosovitskiy et al. (2021); Riquelme et al. (2021) which we recall for completeness. Let us rewrite our model $f$ with parameters $\boldsymbol{\theta} = \{\boldsymbol{Q}, \boldsymbol{\theta}'\}$ as

$$f(\boldsymbol{x}; \boldsymbol{\theta}) = \mathtt{softmax}(\boldsymbol{Q}\phi(\boldsymbol{x}; \boldsymbol{\theta}'))$$

where $\boldsymbol{Q} \in \mathbb{R}^{C \times S}$ corresponds to the parameters of the last layer of $f$ with the $S$-dimensional representation $\phi(\boldsymbol{x}; \boldsymbol{\theta}') \in \mathbb{R}^S$.

Table 7: Hyperparameter values for fine-tuning on different datasets. Compared with Dosovitskiy et al. (2021) and Riquelme et al. (2021), we further consider several factors {0.5, 1.0, 1.5, 2.0} to sweep over different numbers of steps.

| Dataset | Steps | Base LR | Expert Dropout |
|---|---|---|---|
| ImageNet | 20 000 × {0.5, 1.0, 1.5, 2.0} | {0.0024, 0.003, 0.01, 0.03} | 0.1 |
| CIFAR10 | 5 000 × {0.5, 1.0, 1.5, 2.0} | {0.001, 0.003, 0.01, 0.03} | 0.1 |
| CIFAR100 | 5 000 × {0.5, 1.0, 1.5, 2.0} | {0.001, 0.003, 0.01, 0.03} | 0.1 |
| Oxford-IIIT Pets | 500 × {0.5, 1.0, 1.5, 2.0} | {0.001, 0.003, 0.01, 0.03} | 0.1 |
| Oxford Flowers-102 | 500 × {0.5, 1.0, 1.5, 2.0} | {0.001, 0.003, 0.01, 0.03} | 0.1 |

Table 8: Impact of using the enlarged sweep of hyperparameters described in Table 7. We typically improve the results reported in Riquelme et al. (2021), therefore strengthening the baselines we compare to. The table displays means and standard errors over 8 replications, except for H/14 that has 4 replications. **L/16$^\star$**: For L/16, we consider the setting where the upstream models are trained with 7 epochs, as opposed to 14 epochs in Riquelme et al. (2021), hence the slightly worse accuracy reported in this paper.

| Model size | Model name | Accuracy (this paper) | Accuracy (Riquelme et al., 2021) |
|---|---|---|---|
| S/32 | ViT | $76.31_{\pm 0.05}$ | 73.73 |
| | V-MoE (K=2) | $78.91_{\pm 0.08}$ | 77.10 |
| B/32 | ViT | $81.35_{\pm 0.08}$ | 80.73 |
| | V-MoE (K=2) | $83.24_{\pm 0.05}$ | 82.60 |
| L/32 | ViT | $84.62_{\pm 0.05}$ | 84.37 |
| | V-MoE (K=2) | $84.95_{\pm 0.03}$ | 85.04 |
| B/16 | ViT | $84.30_{\pm 0.06}$ | 84.15 |
| | V-MoE (K=2) | $85.40_{\pm 0.04}$ | 85.39 |
| L/16$^\star$ | ViT | $86.63_{\pm 0.08}$ | 87.12 |
| | V-MoE (K=2) | $87.12_{\pm 0.04}$ | 87.54 |
| H/14 | ViT | $88.01_{\pm 0.05}$ | 88.08 |
| | V-MoE (K=2) | $88.11_{\pm 0.13}$ | 88.23 |

In linear few-shot evaluation, we construct a linear classifier to predict the target labels (encoded as one-hot vectors) from the $S$-dimensional feature vectors induced by $\phi(\cdot; \boldsymbol{\theta}')$; see Chapter 5 in Hastie et al. (2017) for more background about this type of linear classifiers. This evaluation protocol makes it possible to evaluate the quality of the representations $\phi$ learned by $f$.

While Dosovitskiy et al. (2021); Riquelme et al. (2021) essentially focus on the quality of the representations learned upstream on JFT by computing the (linear) few-shot accuracy on ImageNet, we are interested in the representations after fine-tuning on ImageNet. As a result, we consider a collection of 8 few-shot datasets (that does not contain ImageNet):

- Caltech-UCSD Birds 200 (Wah et al., 2011) with 200 classes,

- Caltech 101 (Bansal et al., 2021) with 101 classes,

- Cars196 (Krause et al., 2013) with 196 classes,

- CIFAR100 (Krizhevsky, 2009) with 100 classes,

- Colorectal histology (Kather et al., 2016) with 8 classes,

- Describable Textures Dataset (Cimpoi et al., 2014) with 47 classes,

- Oxford-IIIT pet (Parkhi et al., 2012) with 37 classes and

- UC Merced (Yang & Newsam, 2010) with 21 classes.

In the experiments, we compute the few-shot accuracy for each of the above datasets and we report the averaged accuracy over the datasets, for various number of shots in $\{1, 5, 10, 25\}$. As commonly defined in few-shot learning, we understand by $s$ shots a setting wherein we have access to $s$ training images per class label in each of the dataset.

To account for the different scales of accuracy across the 8 datasets, we also tested to compute a weighted average, normalizing by the accuracy of a reference model (ViT-B/32). This is reminiscent of the normalization carried out in Hendrycks & Dietterich (2019) according to the score of AlexNet. We found the conclusions with the standard average and weighted average to be similar.

### A.5.1 Specific Considerations in the Ensemble Case

For an ensemble with $M$ members, we have access to $M$ representations $\{\phi(\boldsymbol{x}; \boldsymbol{\theta}'_m)\}_{m=1}^M$ for a given input $\boldsymbol{x}$. We have explored two ways to use those representations:

- **Joint**: We concatenate the $M$ representations $\{\phi(\boldsymbol{x}; \boldsymbol{\theta}'_m)\}_{m=1}^M$ into a single "joint" feature vector in $\mathbb{R}^{M \cdot S}$, remembering that each $\phi(\boldsymbol{x}; \boldsymbol{\theta}'_m) \in \mathbb{R}^S$. We then train a *single* a linear classifier to predict the target labels from the "joint" feature vectors.

- **Disjoint**: For each of the $M$ representations $\{\phi(\boldsymbol{x}; \boldsymbol{\theta}'_m)\}_{m=1}^M$, we separately train a linear classifier to predict the target labels from the feature vectors induced by $\phi(\boldsymbol{x}; \boldsymbol{\theta}'_m)$. We then average the predictions of the $M$ linear classifiers trained in this fashion.

In Table 9, we report a comparison of those approaches. We aggregate the results over all ensemble models (namely, $\text{E}^3$ and upstream ViT/V-MoE ensembles of size 2 and 4) and over 8 replications, for the ViT families S/32, B/32 and L/32.

The results indicate that "joint" and "disjoint" perform similarly. Throughout our experiments, we use the "joint" approach because it eased some implementation considerations.

### A.6 List of Datasets

For completeness, in addition to the few-shot datasets listed in Appendix A.5, we list the datasets used for downstream training and evaluation in this work.

- ImageNet (ILSVRC2012) (Deng et al., 2009) with 1000 classes and 1281167 training examples.

- ImageNet-C (Hendrycks & Dietterich, 2019), an ImageNet test set constructed by applying 15 different corruptions at 5 levels of intensity to the original ImageNet test set. (We report the mean performance over the different corruptions and intensities.)

- ImageNet-A (Hendrycks et al., 2019), an ImageNet test set constructed by collecting new data and keeping only those images which a ResNet-50 classified incorrectly.

- ImageNet-V2 (Recht et al., 2019), an ImageNet test set independently collected using the same methodology as the original ImageNet dataset.

- CIFAR10 (Krizhevsky, 2009) with 10 classes and 50000 training examples.

- CIFAR10-C (Hendrycks & Dietterich, 2019), a CIFAR10 test set constructed by applying 15 different corruptions at 5 levels of intensity to the original CIFAR10 test set. (We report the mean performance over the different corruptions and intensities.)

- CIFAR100 (Krizhevsky, 2009) with 100 classes and training 50000 examples.

- Oxford Flowers 102 (Nilsback & Zisserman, 2008) with 102 classes and 1020 training examples.

- Oxford-IIIT pet (Parkhi et al., 2012) with 37 classes and 3680 training examples.

- SVHN (Netzer et al., 2011) with 10 classes.

- Places365 (Zhou et al., 2017) with 365 classes.

- Describable Textures Dataset (DTD) (Cimpoi et al., 2014) with 47 classes.

### A.7 Sparse MoEs meet Ensembles Experimental Details

The setup for the experiments in Figures 2 and 12 differs slightly the other experiments in this paper. Specifically, while for all other experiments we used upstream V-MoE checkpoints with $(K, E) = (2, 32)$, for these experiments we matched the upstream and downstream checkpoints. We did this to avoid a checkpoint mismatch as a potential confounder in our results.

### A.8 Multiple Predictions without Tiling or Partitioning Details

The naive multi-pred method presented in Section 4.2.4 was trained in almost the same manner as the vanilla V-MoE, the only difference being the handling of multiple predictions. This was accomplished by using the average ensemble member cross entropy as described for $\textsc{e}^3$ in Appendix C. In contrast, in order to compute the evaluation metrics presented in Table 4, we first averaged predictions of the model and then used the average prediction when calculating each metric.

Table 9: Comparison of two approaches, "joint" and "disjoint", to compute the linear few-shot evaluation in the case of ensembles. For the ViT families S/32, B/32 and L/32, the mean error across datasets is averaged over 8 replications and over all the ensemble models of size 2 and 4.

| MODEL SIZE | METHOD | MEAN ERROR ACROSS DATASETS | | | |
|---|---|---|---|---|---|
| | | 1 SHOT | 5 SHOTS | 10 SHOTS | 25 SHOTS |
| S/32 | disjoint | $51.01_{\pm 0.43}$ | $32.80_{\pm 0.34}$ | $26.33_{\pm 0.26}$ | $20.97_{\pm 0.18}$ |
| | joint | $51.12_{\pm 0.42}$ | $32.81_{\pm 0.30}$ | $26.30_{\pm 0.24}$ | $20.77_{\pm 0.17}$ |
| B/32 | disjoint | $42.43_{\pm 0.41}$ | $25.49_{\pm 0.21}$ | $20.30_{\pm 0.15}$ | $15.98_{\pm 0.11}$ |
| | joint | $42.59_{\pm 0.40}$ | $25.74_{\pm 0.18}$ | $20.54_{\pm 0.13}$ | $16.06_{\pm 0.10}$ |
| L/32 | disjoint | $36.41_{\pm 0.31}$ | $21.49_{\pm 0.15}$ | $17.13_{\pm 0.12}$ | $13.56_{\pm 0.10}$ |
| | joint | $36.48_{\pm 0.30}$ | $21.66_{\pm 0.13}$ | $17.34_{\pm 0.10}$ | $13.56_{\pm 0.08}$ |

## B Compatibility and Adaptation of the Upstream Checkpoints

Throughout the paper, we make the assumption that we can start from existing checkpoints of ViT and V-MoE models (trained on JFT-300M; see Appendix A.2). We next describe how we can use those checkpoints for the fine-tuning of the extensions of ViT and V-MoE that we consider in this paper.

In all our experiments that involve V-MoEs, we consider checkpoints with $K = 2$ and $E = 32$, which is the canonical setting advocated by Riquelme et al. (2021).

### B.1 Efficient Ensemble of Experts

In the case of $\textsc{e}^3$, the set of parameters is identical to that of a V-MoE model. In particular, neither the tiled representation nor the partitioning of the experts transforms the set of parameters.

To deal with the fact that the single routing function $\texttt{gate}_K(\boldsymbol{W}\cdot)$ of a V-MoE becomes separate routing functions $\{\texttt{gate}_K(\boldsymbol{W}_m\cdot)\}_{m=1}^M$, one for expert subset $\mathcal{E}_m$, we simply slice row-wise $\boldsymbol{W} \in \mathbb{R}^{E \times D}$ into the $M$ matrices $\boldsymbol{W}_m \in \mathbb{R}^{(E/M) \times D}$.

### B.2 Batch Ensembles (BE)

We train BE starting from ViT checkpoints, which requires to introduce downstream-specific parameters. Following the design of V-MoEs, we place the batch-ensemble layers in the MLP layers of the Transformer.

Let us consider a dense layer in one of those MLPs, with parameters $\boldsymbol{U} \in \mathbb{R}^{D \times L}$, in absence of bias term. In BE, the parametrization of each ensemble member has the following structure $\boldsymbol{U}_m = \boldsymbol{U} \circ (\boldsymbol{r}_m \boldsymbol{s}_m^\top)$ where $\{\boldsymbol{r}_m\}_{m=1}^M$ and $\{\boldsymbol{s}_m\}_{m=1}^M$ are respectively $D$- and $L$-dimensional vectors.

A standard ViT checkpoint provides pre-trained parameters for $\boldsymbol{U}$. We then introduce $\{\boldsymbol{r}_m\}_{m=1}^M$ and $\{\boldsymbol{s}_m\}_{m=1}^M$ at fine-tuning time, following the random initialization schemes proposed in Wen et al. (2019); see details in the hyperparameter sweep for BE in Appendix G.1.

### B.3 MIMO

We train MIMO models from V-MoE checkpoints. The only required modifications are to the input and output parameters of the checkpoints. The linear input embedding must be modified to be compatible with input images containing $M$ times as more channels, as required by the multiple-input structure of MIMO. Similarly, the final dense layer in the classification head must be modified to have $M$ times more output units, following the multiple-output structure in MIMO.

Concretely, the embedding weight $\boldsymbol{W}_{\text{in}} \in \mathbb{R}^{H \times W \times 3 \times D}$ is replicated in the third (channel) dimension, resulting in $\boldsymbol{W}_{\text{MIMO,in}} \in \mathbb{R}^{H \times W \times 3 \cdot M \times D}$, where $H$ and $W$ are the height and width of the convolution and $D$ is the hidden dimension of the ViT family (specified in Table 6). The output layer weight $\boldsymbol{W}_{\text{out}} \in \mathbb{R}^{D \times C}$ is replicated in the second (output) dimension, resulting in $\boldsymbol{W}_{\text{MIMO,out}} \in \mathbb{R}^{H \times C \cdot M}$, where $C$ is the number of classes. The output layer bias $\boldsymbol{b}_{\text{out}} \in \mathbb{R}^C$ is replicated resulting in $\boldsymbol{b}_{\text{MIMO,out}} \in \mathbb{R}^{C \times M}$. Finally, in order to preserve the magnitude of the activation for these layers, $\boldsymbol{W}_{\text{MIMO,in}}$ and $\boldsymbol{W}_{\text{MIMO,out}}$ are scaled by $1/M$.

## C Implementation Details of Efficient Ensemble of Experts

We provide details about the training loss and the regularizer used by $\text{E}^3$. We also discuss the memory requirements compared to V-MoE.

### C.1 Training Loss

Since $\text{E}^3$ outputs $M$ predictions $\{f(\boldsymbol{x}; \boldsymbol{\theta}_m)\}_{m=1}^M$ for a given input $\boldsymbol{x}$, we need to adapt the choice of the training loss $\mathcal{L}$ accordingly. Following the literature on efficient ensembles (Wen et al., 2019; Dusenberry et al., 2020a; Wenzel et al., 2020), we choose the average ensemble-member cross entropy

$$\mathcal{L}(y, \boldsymbol{x}; \boldsymbol{\theta}) = \frac{1}{M} \sum_{m=1}^M \text{cross-entropy}(y, f(\boldsymbol{x}; \boldsymbol{\theta}_m))$$

instead of other alternatives such as the ensemble cross-entropy

$$\text{cross-entropy}\left(y, \frac{1}{M} \sum_{m=1}^M f(\boldsymbol{x}; \boldsymbol{\theta}_m)\right)$$

that was observed to generalize worse (Dusenberry et al., 2020a).

### C.2 Auxiliary Losses

Inspired by previous applications of sparse MoEs in NLP (Shazeer et al., 2017), Riquelme et al. (2021) employ regularizers, also referred to as *auxiliary losses*, to guarantee a balanced usage of the $E$ experts. Two auxiliary losses—the importance and load losses, see Appendix A in Riquelme et al. (2021) for their formal definitions—are averaged together to form the final regularization term that we denote by $\Omega$.

As a reminder, let us recall the notation of the routing function

$$\boldsymbol{h} \in \mathbb{R}^D \mapsto \texttt{gate}_K(\boldsymbol{W}\boldsymbol{h}) = \texttt{top}_K(\texttt{softmax}(\boldsymbol{W}\boldsymbol{h} + \sigma\varepsilon)) \in \mathbb{R}^E,$$

with $\boldsymbol{W} \in \mathbb{R}^{E \times D}$ and $\varepsilon \sim \mathcal{N}(\boldsymbol{0}, \boldsymbol{I})$. Consider a batch of $B$ inputs $\{\boldsymbol{h}_i\}_{i=1}^B$ that we represent by $\boldsymbol{H} \in \mathbb{R}^{B \times D}$. Finally, let us define

$$\boldsymbol{A} = \boldsymbol{H}\boldsymbol{W}^\top + \sigma\varepsilon_{B \times E} \in \mathbb{R}^{B \times E},$$

where we emphasise that $\varepsilon_{B \times E}$ is a matrix of Gaussian noise entries in $\mathbb{R}^{B \times E}$. The regularization term $\Omega$ used by Riquelme et al. (2021) can be seen as a function that depends on $\boldsymbol{A}$ and $\boldsymbol{H}\boldsymbol{W}^\top$.

In the context of efficient ensemble of experts, the set of $E$ experts is partitioned into $M$ subsets of $E/M$ experts, denoted by $\cup_{m=1}^M \mathcal{E}_m$; see Section 4.1. With the introduction of the $M$ routing functions $\{\texttt{gate}_K(\boldsymbol{W}_m \cdot)\}_{m=1}^M$ with each $\boldsymbol{W}_m \in \mathbb{R}^{(E/M) \times D}$, the matrix $\boldsymbol{A}$ becomes accordingly partitioned into $\{\boldsymbol{A}_m\}_{m=1}^M$ where each $\boldsymbol{A}_m \in \mathbb{R}^{B \times (E/M)}$.

Since we want to enforce a balanced usage of the $E/M$ experts in each subset $\mathcal{E}_m$ of the experts, we thus redefine the regularization as the average regularization separately applied to each part of the partition

$$\Omega_{\text{partition}}(\boldsymbol{A}, \boldsymbol{H}\boldsymbol{W}^\top) = \frac{1}{M} \sum_{m=1}^M \Omega(\boldsymbol{A}_m, \boldsymbol{H}\boldsymbol{W}_m^\top).$$

We found this option to work better in practice. To guarantee a fair comparison, we also applied $\Omega_{\text{partition}}$ to the "Only partitioning" model in the ablation study of Section 4.2.1.

Following Riquelme et al. (2021), the regularization parameter controlling the strength of $\Omega_{\text{partition}}$ was set to 0.01 throughout the experiments.

## C.3 Memory Requirements versus V-MoE

Due to tiling, $\textsc{e}^3$ requires more memory than V-MoE. To be concrete, the memory complexity of V-MoE can be decomposed into two terms

$$\mathcal{O}\left(\texttt{memory}_{\text{params}} + \texttt{memory}_{\text{activations}}\right),$$

for the forward and backward passes, respectively. For $\textsc{e}^3$, the complexity becomes

$$\mathcal{O}\left(\texttt{memory}_{\text{params}} + \texttt{memory}_{\text{activations}} \times \frac{L_{\text{before}} + L_{\text{after}} \times M}{L_{\text{total}}}\right),$$

where $M$ is the ensemble size, and, $L_{\text{before}}$, $L_{\text{after}}$, and $L_{\text{total}}$ are the number of layers before tiling, after tiling, and in total, respectively. Importantly, neither $\texttt{memory}_{\text{params}}$ nor $\texttt{memory}_{\text{activations}}$ depend on $M$. Thanks to the "last-$n$" setting employed in the paper, we have $L_{\text{after}} \ll L_{\text{before}}$, and thus the increase in memory due to tiling remains mild. More concretely, for ViT-L, we have $L_{\text{total}} = 24$, $L_{\text{before}} = 21$, and $L_{\text{after}} = 3$ (with MoE layers placed at layers 22 and 24). Thus, for an ensemble of size $M = 2$, $\textsc{e}^3$ would only increase $\texttt{memory}_{\text{activations}}$ by 12.5%, while leaving $\texttt{memory}_{\text{params}}$ unchanged.

## D Efficient Ensemble of Experts and V-MoE Relative Improvements per ViT Family

In Section 5 we claim that $\textsc{e}^3$ performs best at the largest scale. In this section we motivate that claim in more detail. Specifically, we consider two metrics of improvement in performance. Firstly, we consider the percentage improvement in NLL for both $\textsc{e}^3$ and V-MoE versus vanilla ViT. Secondly, we consider a normalised version of this improvement. We consider this second metric to take into account the "difficulty" in further improving the NLL of larger ViT family models. Intuitively, the larger the ViT family, the better the corresponding NLL will be, and the more difficult it will be to improve on that NLL.

The normalisation we apply is based on the gradient of the NLL with respect to FLOPs. Indeed, this gradient captures the typical variation of NLL at a particular amount of FLOPs. The ratio of this gradient at the FLOPs values (i.e., the instantaneous change in NLL at those FLOPs values) for two ViT families

is a measure of the relative difficulty in increasing the NLL. Thus, we can use this ratio to normalise our results. To be more concrete, let us define the mapping

$$\text{NLL} = \varphi(\text{FLOPs}) \quad \text{and its derivative} \quad \varphi'(\text{FLOPs}) = \frac{d\varphi(\text{FLOPs})}{d\text{FLOPs}}.$$

We estimate $\varphi$ and its gradient by fitting a linear model to the $(\text{NLL}, \text{FLOPs})$ pairs for each ViT family, using the data of the standard ViT models we trained. We use feature expansion $[1.0, \log(\text{FLOPs}), \log(\text{FLOPs})^2, \log(\text{FLOPs})^3]$ and solve for the parameters of the linear model via ordinary least squares. We determine the gradient of this function at each FLOPs value using automatic differentiation in `JAX` (Bradbury et al., 2018). See Figure 9 for the resulting fit and an indication of the gradients.

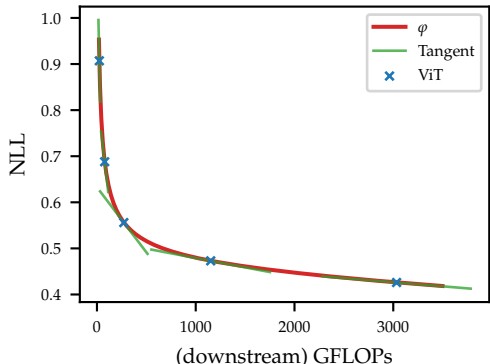

Figure 9: Estimated $\varphi$ compared to the ImageNet NLL values for our ViT models. We also include the tangent at the points corresponding to each ViT model to indicate the gradients at those points.

The normalised values are calculated as:

$$\text{Normalised improvement}(v) = \text{improvement}(v) \times \frac{\varphi'(\text{FLOPs }_{\text{H}/14})}{\varphi'(\text{FLOPs}_v)}, \tag{4}$$

where $v$ is one of the ViT families, i.e., S/32, B/32, L/32, L/16, or H/14. Note that this normalisation leaves the improvement for H/14 the same. We tried to normalize with respect to other choices of ViT family, different from H/14. Our conclusions are robust, in the sense that both the ordering and the monotonic behavior with respect to scale are preserved. Using the ratio for normalisation also has the advantage that the normalisation is less sensitive to the particular parameterisation of $\varphi$.

Table 10: Percentage improvements in NLL for $\text{E}^3$ with $(K, M) = (1, 2)$ and V-MoE with $K = 1$ vs. ViT for families of increasing size. The top two rows show normalised improvements, see (4), which take into consideration the increased difficulty of improving NLL for larger ViT families whose performance is beginning to saturate. The bottom two rows are the original percentage improvements without normalisation.

|  |  | S/32 | B/32 | L/32 | L/16 | H/14 |
|---|---|---|---|---|---|---|
| Normalised | $\text{E}^3$ vs. ViT | 0.02% | 0.09% | 0.24% | 2.35% | 4.27% |
|  | V-MoE vs. ViT | 0.01% | 0.06% | -0.04% | 0.89% | 0.02% |
| Not normalised | $\text{E}^3$ vs. ViT | 9.82% | 9.53% | 3.76% | 5.38% | 4.27% |
|  | V-MoE vs. ViT | 7.98% | 6.62% | -0.60% | 2.05% | 0.02% |

Table 10 shows both the difficulty-normalised and original improvements (without normalisation). Looking first at the original improvements, we can see that while both $\text{E}^3$ and V-MoE have smaller improvements over ViT for larger families, $\text{E}^3$'s improvements decrease more slowly. Furthermore, by comparing the normalised improvements we see that $\text{E}^3$'s improvements actually *grow* monotonically when taking difficulty into account. This is not the case for V-MoE.

## E    From Batch Ensembles to Sparse MoEs

Wen et al. (2019) have shown that, given a batch of $B$ inputs $\boldsymbol{X} \in \mathbb{R}^{B \times P}$, a single forward pass can efficiently compute the predictions of all the ensemble members $\{f(\boldsymbol{X}; \boldsymbol{\theta}_m)\}_{m=1}^{M}$. By appropriately tiling the inputs of the network $\boldsymbol{X}_{\text{tiled}} \in \mathbb{R}^{(M \cdot B) \times P}$ by a factor $M$, each internal operation per ensemble member can then be vectorized.

We take the previous example of a dense layer with parameters $\boldsymbol{U} \in \mathbb{R}^{D \times L}$ and we assume the layer receives the tiled inputs $\{\boldsymbol{H}_m\}_{m=1}^{M}$ where $\boldsymbol{H}_m \in \mathbb{R}^{B \times D}$. We need to compute for each ensemble member $\boldsymbol{H}_m \boldsymbol{U}_m = \boldsymbol{H}_m[\boldsymbol{U} \circ (\boldsymbol{r}_m \boldsymbol{s}_m^\top)]$. Denoting by $\boldsymbol{h}_{i,m} \in \mathbb{R}^D$ the $i$-th input in $\boldsymbol{H}_m$, we have

$$\boldsymbol{h}_{i,m}^\top \boldsymbol{U}_m = \sum_{e=1}^{E} g_e(\boldsymbol{h}_{i,m}) \cdot \texttt{expert}_e(\boldsymbol{h}_{i,m}) \quad \text{with } M = E, \quad \begin{cases} g_e(\boldsymbol{h}_{i,m}) = 1 \text{ if } e = m, \\ g_e(\boldsymbol{h}_{i,m}) = 0 \text{ otherwise} \end{cases} \tag{5}$$

and $\texttt{expert}_e(\boldsymbol{z}) = \boldsymbol{z}^\top[\boldsymbol{U} \circ (\boldsymbol{r}_e \boldsymbol{s}_e^\top)]$. Although (5) may appear as a convoluted way of writing the operations in batch ensembles, it unveils a connection with (1). Indeed, operations in batch ensembles can be seen as a specific sparse MoE, e.g., with binary routing weights depending only on the position in the tiled inputs. While Wen et al. (2019) primarily tiled the inputs for the sake of efficiency, it also induces some form of conditional computation, an insight that we exploit in $\textsc{e}^3$.

## F    Batch Ensembles versus Efficient Ensemble of Experts

Table 11: Summary of the differences between BE and $\textsc{e}^3$.

|  | SHARED PARAMETERS | UNIQUE PARAMETERS | TILING | TRAINING EPOCHS |
|---|---|---|---|---|
| BE | Kernels in each linear layer | "Fast-weights" and biases | Start of the model | Typically $\approx$50% more |
| $\textsc{e}^3$ | All parameters outside MoE layer | Experts in subset $\mathcal{E}_m$ | Before 1st MoE layer | Unchanged |

While $\textsc{e}^3$ is inspired in several ways by BE, it has also been specialised to sparse MoEs. This leads to a number of significant differences between the two algorithms. These differences are summarised in Table 11, and we elaborate upon them here:

- **Shared parameters:** In BE, the shared parameters are the kernels in each linear (e.g. dense or convolution) layer. This makes up the majority of the parameters in BE. In $\textsc{e}^3$, the shared parameters are all of those not found in the expert layers.

- **Ensemble member specific parameters:** BE uses pairs of "fast-weights" $\{\boldsymbol{r}_m, \boldsymbol{s}_m\}$ and bias terms for each linear layer. $\textsc{e}^3$ uses the parameters of all the $E/M$ experts in the $m$-th subset $\mathcal{E}_m$ and the router for those expert, in each MoE layer. This results in a much larger set of ensemble member specific parameters, and thus a higher level of predictive diversity (see Section 4.2.3).

- **Tiling:** In BE, tiling is performed at the very beginning of the model, regardless of the model's internal structure. Tiling in $\textsc{e}^3$ occurs before the first MoE layer, thus saving redundant computation for "Last-n" V-MoE.

- **Number of training epochs:** BE usually requires 50% more training epochs than a vanilla model. $\textsc{e}^3$ requires the same number of training epochs as V-MoE.

## G    Efficient Ensemble Comparisons

In this section, we compare efficient ensemble of experts ($\textsc{e}^3$) to several popular efficient ensemble approaches, namely MIMO (Havasi et al., 2020), batch ensemble (BE) (Wen et al., 2019), and MC Dropout (Gal & Ghahramani, 2016).

Table 12: ImageNet performance of different efficient ensemble approaches. The table reports the means ± standard errors over 8 replications. All models have a ViT-B/32 architecture. $K$ stands for the sparsity in V-MoEs, $M$ denotes the ensemble size while "BR" corresponds to the batch repetition in MIMO (Havasi et al., 2020).

| | K | M | NLL ↓ | Error ↓ | ECE ↓ | KL ↑ | GFLOPs ↓ |
|---|---|---|---|---|---|---|---|
| ViT | – | – | $0.688_{\pm 0.003}$ | $18.65_{\pm 0.08}$ | $0.022_{\pm 0.000}$ | – | 78.0 |
| BE ViT | – | 2 | $0.682_{\pm 0.003}$ | $18.47_{\pm 0.05}$ | $0.021_{\pm 0.000}$ | $0.040_{\pm 0.001}$ | 97.1 |
| | – | 4 | $0.675_{\pm 0.003}$ | $18.40_{\pm 0.09}$ | $0.017_{\pm 0.000}$ | $0.035_{\pm 0.001}$ | 135.4 |
| V-MoE | 1 | – | $0.642_{\pm 0.002}$ | $16.90_{\pm 0.05}$ | $0.029_{\pm 0.001}$ | – | 82.4 |
| | 2 | – | $0.638_{\pm 0.001}$ | $16.76_{\pm 0.05}$ | $0.033_{\pm 0.001}$ | – | 94.9 |
| | 4 | – | $0.636_{\pm 0.001}$ | $16.70_{\pm 0.04}$ | $0.034_{\pm 0.001}$ | – | 120.1 |
| | 8 | – | $0.635_{\pm 0.002}$ | $16.72_{\pm 0.06}$ | $0.028_{\pm 0.001}$ | – | 170.4 |
| MC Dropout V-MoE | 1 | 2 | $0.648_{\pm 0.002}$ | $17.10_{\pm 0.05}$ | $0.019_{\pm 0.001}$ | $0.046_{\pm 0.000}$ | 97.2 |
| | 1 | 4 | $0.641_{\pm 0.002}$ | $16.96_{\pm 0.05}$ | $0.017_{\pm 0.001}$ | $0.046_{\pm 0.001}$ | 135.6 |
| | 2 | 2 | $0.642_{\pm 0.002}$ | $16.94_{\pm 0.04}$ | $0.021_{\pm 0.001}$ | $0.046_{\pm 0.001}$ | 113.7 |
| | 2 | 4 | $0.634_{\pm 0.001}$ | $16.80_{\pm 0.03}$ | $0.020_{\pm 0.000}$ | $0.046_{\pm 0.001}$ | 168.6 |
| | 4 | 2 | $0.639_{\pm 0.002}$ | $16.91_{\pm 0.06}$ | $0.022_{\pm 0.001}$ | $0.045_{\pm 0.001}$ | 146.7 |
| MIMO V-MoE (BR=1) | 2 | 2 | $0.636_{\pm 0.002}$ | $16.97_{\pm 0.04}$ | $0.028_{\pm 0.001}$ | $0.000_{\pm 0.000}$ | 96.3 |
| | 2 | 4 | $0.672_{\pm 0.001}$ | $17.72_{\pm 0.04}$ | $0.037_{\pm 0.000}$ | $0.001_{\pm 0.000}$ | 99.0 |
| MIMO V-MoE (BR=2) | 2 | 2 | $0.638_{\pm 0.001}$ | $17.14_{\pm 0.03}$ | $0.031_{\pm 0.000}$ | $0.001_{\pm 0.000}$ | 192.6 |
| | 2 | 4 | $0.665_{\pm 0.002}$ | $17.38_{\pm 0.04}$ | $0.038_{\pm 0.000}$ | $0.000_{\pm 0.000}$ | 198.1 |
| E³ | 1 | 2 | $0.622_{\pm 0.001}$ | $16.70_{\pm 0.03}$ | $0.018_{\pm 0.000}$ | $0.217_{\pm 0.003}$ | 105.9 |
| | 1 | 4 | $0.624_{\pm 0.001}$ | $16.99_{\pm 0.03}$ | $0.013_{\pm 0.000}$ | $0.164_{\pm 0.001}$ | 153.0 |
| | 2 | 2 | $0.612_{\pm 0.001}$ | $16.49_{\pm 0.02}$ | $0.013_{\pm 0.000}$ | $0.198_{\pm 0.003}$ | 131.1 |
| | 2 | 4 | $0.620_{\pm 0.001}$ | $16.86_{\pm 0.02}$ | $0.015_{\pm 0.000}$ | $0.170_{\pm 0.001}$ | 203.3 |
| | 4 | 2 | $0.611_{\pm 0.001}$ | $16.45_{\pm 0.03}$ | $0.014_{\pm 0.000}$ | $0.193_{\pm 0.003}$ | 181.4 |

Table 12 reports the ImageNet performance of those different techniques, when all models are based on a ViT-B/32 architecture. We start by highlighting the most salient conclusions of the experiment and defer to the next subsections the descriptions of the different competing techniques.

We make the following observations:

- BE built upon ViT improves the performance of ViT in terms of NLL, classification error and ECE. However, the resulting increase in FLOPs makes BE a less viable option compared to E³.

- MC Dropout V-MoE is on par with standard V-MoE in terms of NLL and classification error, while it improves the ECE. For all values of $K$, we observe that the performance tends to improve as the number of samples, i.e., $M$, increases. However, already for $M$ in $\{2, 4\}$, the resulting increase in FLOPs makes MC Dropout V-MoE a less favorable option compared to E³.

- Perhaps surprisingly (see detailed investigations in Appendix G.3), MIMO V-MoE does not lead to improvements compared with V-MoE. In fact, for higher ensembles sizes, MIMO V-MoE results in worse performance than standard V-MoE. Moreover, increasing the batch repetition parameter of MIMO ("BR" in Table 12) further worsens the results. Interestingly, we can see that MIMO does not manage to produce diverse predictions, as illustrated by the small values of KL.

- E³ offers the best performance vs. FLOPs trade-offs, e.g., when looking at $(K, M) = (1, 2)$ and $(K, M) = (2, 2)$. We notably observe that the diversity of the predictions in E³ is orders of magnitude larger than that of the other ensemble approaches.

We briefly recall the optimization explained in Section 4.1 to save redundant computations: In the "last-$n$" setting of Riquelme et al. (2021), it is sufficient to tile the representations only when entering the first MoE layer/dropout layer/batch-ensemble layer for respectively $E^3$/MC Dropout V-MoE/BE. We apply this optimization to all the efficient ensemble methods.

### G.1 Batch Ensembles

Following the design of V-MoEs, we place the batch-ensemble layers in the MLP layers of the Transformer, following the "last-$n$" setting of Riquelme et al. (2021); see Section 2.1.

The vectors of the rank-1 parametrization introduced at fine-tuning time (see Appendix B) need to be initialized and optimized. Following the recommendation from Wen et al. (2019), we consider the following hyperparameters in addition to the common sweep described in Table 7:

- **Initialization:** Either a random sign vector with entries in $\{-1, 1\}$ independently drawn with probability $\frac{1}{2}$ or a random Gaussian vector with entries independently drawn from $\mathcal{N}(1, 0.5)$.

- **Learning-rate scale factor:** The vectors of the rank-1 parametrization are updated with a learning rate scaled by a factor in $\{0.5, 1, 2\}$.

### G.2 MC Dropout V-MoEs

For MC Dropout V-MoE, we take the available fine-tuned V-MoEs and enable dropout at prediction time. Indeed, as described in Table 7, all V-MoE models already have a 0.1 dropout rate in the experts.

### G.3 MIMO V-MoEs

Following Havasi et al. (2020) we consider two MIMO-specific hyperparameters, in addition to the hyperparameters listed in Table 7:

- **Input replication probability:** $\{0.5, 0.625, 0.75\}$

- **Batch repetitions:** $\{1, 2\}$

Our preliminary investigations also considered lower input repetition probabilities and higher batch repetitions. However, lower input repetition probabilities tended to result in poorer performance. While higher batch repetitions did help to some extent, the additional computational cost made it impractical.

Given the surprising result that an ensemble size of $M = 2$ provides no performance improvement over the standard V-MoE and that increasing $M$ further provides worse performance, there seems to be some incompatibility between MIMO and V-MoE. In fact, our investigations revealed that ViT is the source of the problems since applying MIMO to vanilla ViT without experts resulted in the same trends as for V-MoE. Thus we hypothesise that the differences between ViT and ResNet—the architecture to which MIMO was originally applied by Havasi et al. (2020)—are responsible for MIMO's poor performance when applied to ViT.

**Difference 1: Class token.** One of the differences between ViT and ResNet is that ViT makes use of a special learnable class token to classify an image (see Dosovitskiy et al. (2021) for details). ResNet on the other hand makes use of the representation from an entire image for classification. We tried two strategies to mitigate this difference:

1. We applied the global average pooling (GAP) and multi-head attention pooling (MAP) classification strategies introduced in Dosovitskiy et al. (2021) and Zhai et al. (2021), respectively. In short, both of these methods make use of all the tokens from an image for classification. However, neither of these strategies made a significant difference to the relative performance of MIMO and ViT. In fact, the choice of classification method was the least impactful hyperparameter in our sweep.

2. Rather than learning a single class token, we learnt $M$ class tokens. This strategy resulted in MIMO with $M = 2$ outperforming ViT. However, for $M > 2$ the improvement was small enough that ViT still outperformed MIMO.

**Difference 2: Attention.** The other major difference between ViT and ResNet is the building block for each model. While ResNets are primarily composed of convolution operations, ViT makes heavy used of attention. We hypothesised that attention is less suited to separating the information for $M$ input images stored in the channel dimension of a single image. We tried two strategies to mitigate this potential issue:

1. We applied the hybrid architecture, described in Dosovitskiy et al. (2021), in which the input sequence to ViT is formed by CNN feature maps. We used ResNet14 and ResNet50. In both cases, we found that strategy boosted the performance of ViT and MIMO equally.

2. Rather than concatenating images in the channel dimension, we concatenated them in the width dimension, resulting in 3 times as many patches for ViT to process. This strategy was successful in the sense that the MIMO performance for $M > 2$ improved significantly. However, the significant additional computational cost made it an infeasible solution.

Our findings suggest that MIMO and ViT are indeed somewhat incompatible. Unfortunately, none of our proposed solutions to this problem provided high enough predictive performance increases (or indeed low enough computational cost increases in some cases) to warrant immediate further investigation.

## H Sensitivity Analyses

### H.1 Load Balancing

Figure 10 explores the effect of the load balancing loss on diversity and predictive performance. We see that for both V-MoE and $\text{E}^3$ the predictive performance, as measured by NLL, Error, and ECE are largely insensitive to the strength of the load balancing loss. Similarly, the diversity of the predictions of $\text{E}^3$—as measured by the KL—mildly depends on that regularization. Only when the load balancing loss strength becomes excessively large (4 orders of magnitude larger than standard values) do all the performance metrics plummet. In other words, this hyperparameter does not allow us to *increase* the diversity and thereby the predictive performance of our models.

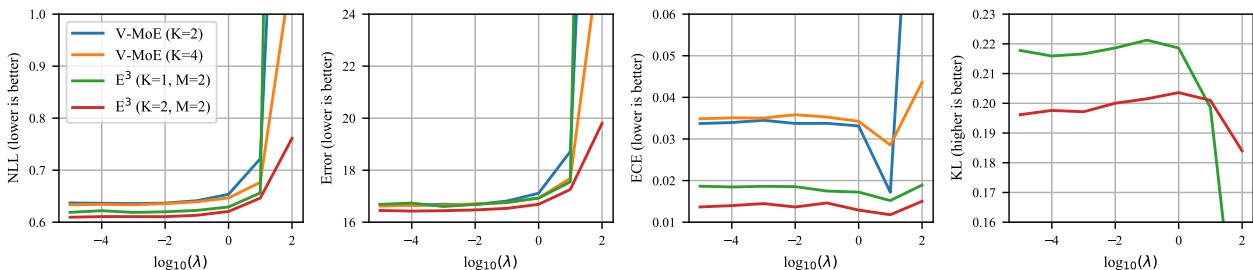

Figure 10: The effect of $\lambda$—the strength of the load balancing, see appendix A of Riquelme et al. (2021)—on NLL, Error, ECE and diversity, for $\text{E}^3$ and V-MoE. Results are averaged over three random seeds. All models have a ViT-B/32 architecture.

### H.2 Expert Initialization

Figure 11 explores the influence of the initialization of the experts. In particular, we add Gaussian noise with varying standard deviations to the initial weights of the expert MLPs. We notably show that more diverse initializations (larger standard deviations) do not translate to any clear performance gain. Note that this experiment takes place *upstream*, since downstream, the experts are already initialized (we fine-tune them from the upstream checkpoint).

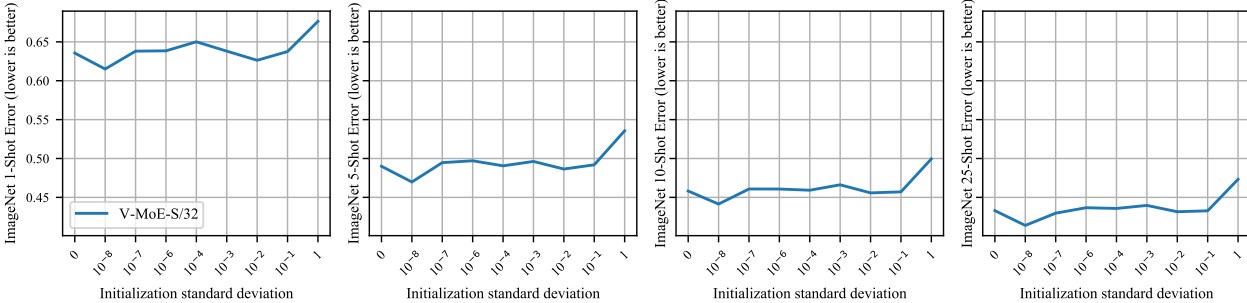

Figure 11: The influence of noise in the expert MLPs' initial random weights. The models are trained on JFT-300M and we measure the ImageNet few-shot Error as in Riquelme et al. (2021). Results are for a single random seed.

### H.3 Expert Group Permutation

Table 13 investigates the performance of $\text{E}^3$ where the ordering of the upstream V-MoE experts has been randomly permuted before fine-tuning[1]. We see that the permutation of the experts has little-to-no significant impact on performance (as measured by NLL and error). The small discrepancies between error for $K = 1$ and NLL for $K = 2$ can be explained by noting that $\text{E}^3$ aggregates results with different upstream checkpoints while $\text{E}^3$ (permuted) only takes one of these checkpoints and varies the permutations; this was done to isolate the effect of the permutations only. Note also that the standard errors for $\text{E}^3$ and $\text{E}^3$ (permuted) are very close to each other, which indicates that permutation of experts has a similar impact to random initialization.

Table 13: Comparison of $\text{E}^3$-B/32 with $E = 32$ total experts and $M = 2$ partitions, and $\text{E}^3$ where the order of the experts has been permuted. For $\text{E}^3$ results are aggregated over 8 random seeds whereas for $\text{E}^3$ (permuted) results are aggregated over 5 permutations.

|  | K | NLL ↓ | ERROR ↓ |
|---|---|---|---|
| $\text{E}^3$ | 1 | **0.622** $_{\pm\,0.001}$ | **16.70** $_{\pm\,0.03}$ |
| $\text{E}^3$ (permuted) | 1 | **0.620** $_{\pm\,0.001}$ | 16.80 $_{\pm\,0.04}$ |
| $\text{E}^3$ | 2 | 0.612 $_{\pm\,0.001}$ | **16.49** $_{\pm\,0.02}$ |
| $\text{E}^3$ (permuted) | 2 | **0.608** $_{\pm\,0.000}$ | **16.47** $_{\pm\,0.03}$ |

## I The Roles of Ensemble Diversity and Individual Model Performance

Table 14 shows the individual member performance for each of our efficient ensemble variants as well as upstream and downstream deep ensembles, for sizes $M = 2$ and $M = 4$. For each method and ensemble size, the first row shows the combined ensemble performance, and the following rows show the performance of the individual ensemble members.

For $\text{E}^3$, the gap between single members and their ensemble is considerable. This is reminiscent of deep ensembles (both variants) and in stark contrast with what we observe for the other efficient ensembles: BE ViT and MIMO V-MoE. The diversity of $\text{E}^3$ is comparable to that of upstream deep ensembles and much larger than downstream deep ensembles. For example, if we compare MIMO V-MoE (M=2) and $\text{E}^3$ (M=2), the single members of $\text{E}^3$ are all worse in NLL, ACC, and ECE than the single members of MIMO. However, the performance gap between the individual members and the ensemble is much larger for $\text{E}^3$ than MIMO (where there is almost no difference). The diversity of the individual models in $\text{E}^3$ is key to its strong performance. Thus, $\text{E}^3$ outperforms MIMO.

---

[1]We also permute the columns of the routing matrix $\boldsymbol{W}_m$ accordingly to keep the routing consistent.

In short, this suggests that $\text{E}^3$ approximates a deep ensemble of smaller V-MoE models. That is, with $\text{E}^3$, we are able to take advantage of the overparameterization of the experts to create a rich set of ensemble members within a single model. Each ensemble member has a large number of non-shared parameters, thus high induced diversity. In comparison, BE only has a few vectors specific to each member.

Table 14: Comparison of the individual ensemble member performance and combined ensemble performance for BE ViT, MIMO V-MoE ($K = 2$, BR=1), $\mathrm{E}^3$ ($K = 1$), as well as upstream and downstream V-MoE ($K = 1$) ensembles.

| | | | NLL ↓ | ERROR ↓ | ECE ↓ | KL ↑ |
|---|---|---|---|---|---|---|
| **BE ViT** | M=2 | ensemble | $0.682_{\pm 0.003}$ | $18.47_{\pm 0.05}$ | $0.021_{\pm 0.000}$ | $0.040_{\pm 0.001}$ |
| | | member 0 | $0.693_{\pm 0.003}$ | $18.68_{\pm 0.05}$ | $0.025_{\pm 0.000}$ | — |
| | | member 1 | $0.693_{\pm 0.003}$ | $18.68_{\pm 0.04}$ | $0.025_{\pm 0.001}$ | — |
| | M=4 | ensemble | $0.675_{\pm 0.003}$ | $18.40_{\pm 0.09}$ | $0.017_{\pm 0.000}$ | $0.035_{\pm 0.001}$ |
| | | member 0 | $0.690_{\pm 0.003}$ | $18.70_{\pm 0.09}$ | $0.022_{\pm 0.000}$ | — |
| | | member 1 | $0.690_{\pm 0.003}$ | $18.70_{\pm 0.08}$ | $0.023_{\pm 0.000}$ | — |
| | | member 2 | $0.690_{\pm 0.003}$ | $18.70_{\pm 0.07}$ | $0.023_{\pm 0.000}$ | — |
| | | member 3 | $0.691_{\pm 0.003}$ | $18.70_{\pm 0.07}$ | $0.023_{\pm 0.000}$ | — |
| **MIMO V-MoE** | M=2 | ensemble | $0.636_{\pm 0.002}$ | $16.97_{\pm 0.04}$ | $0.028_{\pm 0.001}$ | $0.001_{\pm 0.000}$ |
| | | member 0 | $0.636_{\pm 0.002}$ | $16.97_{\pm 0.04}$ | $0.028_{\pm 0.001}$ | — |
| | | member 1 | $0.636_{\pm 0.002}$ | $16.97_{\pm 0.04}$ | $0.028_{\pm 0.000}$ | — |
| | M=4 | ensemble | $0.672_{\pm 0.001}$ | $17.72_{\pm 0.04}$ | $0.037_{\pm 0.000}$ | $0.001_{\pm 0.000}$ |
| | | member 0 | $0.672_{\pm 0.001}$ | $17.74_{\pm 0.05}$ | $0.037_{\pm 0.000}$ | — |
| | | member 1 | $0.672_{\pm 0.001}$ | $17.71_{\pm 0.05}$ | $0.037_{\pm 0.000}$ | — |
| | | member 2 | $0.672_{\pm 0.001}$ | $17.72_{\pm 0.05}$ | $0.037_{\pm 0.000}$ | — |
| | | member 3 | $0.673_{\pm 0.001}$ | $17.73_{\pm 0.05}$ | $0.037_{\pm 0.000}$ | — |
| **$\mathrm{E}^3$** | M=2 | ensemble | $0.622_{\pm 0.001}$ | $16.70_{\pm 0.03}$ | $0.018_{\pm 0.000}$ | $0.217_{\pm 0.003}$ |
| | | member 0 | $0.671_{\pm 0.003}$ | $17.74_{\pm 0.06}$ | $0.038_{\pm 0.001}$ | — |
| | | member 1 | $0.683_{\pm 0.002}$ | $17.94_{\pm 0.005}$ | $0.038_{\pm 0.001}$ | — |
| | M=4 | ensemble | $0.624_{\pm 0.001}$ | $16.99_{\pm 0.03}$ | $0.013_{\pm 0.000}$ | $0.164_{\pm 0.001}$ |
| | | member 0 | $0.677_{\pm 0.002}$ | $18.04_{\pm 0.05}$ | $0.034_{\pm 0.001}$ | — |
| | | member 1 | $0.685_{\pm 0.001}$ | $18.23_{\pm 0.05}$ | $0.034_{\pm 0.001}$ | — |
| | | member 2 | $0.691_{\pm 0.002}$ | $18.35_{\pm 0.07}$ | $0.035_{\pm 0.001}$ | — |
| | | member 3 | $0.697_{\pm 0.002}$ | $18.47_{\pm 0.08}$ | $0.035_{\pm 0.001}$ | — |
| **Up-DE** | M=2 | ensemble | $0.588_{\pm 0.001}$ | $15.74_{\pm 0.05}$ | $0.017_{\pm 0.001}$ | $0.214_{\pm 0.001}$ |
| | | member 0 | $0.640_{\pm 0.001}$ | $16.82_{\pm 0.04}$ | $0.030_{\pm 0.000}$ | — |
| | | member 1 | $0.645_{\pm 0.002}$ | $16.97_{\pm 0.06}$ | $0.029_{\pm 0.002}$ | — |
| | M=4 | ensemble | $0.561_{\pm 0.001}$ | $15.10_{\pm 0.03}$ | $0.020_{\pm 0.000}$ | $0.214_{\pm 0.001}$ |
| | | member 0 | $0.639_{\pm 0.001}$ | $16.80_{\pm 0.03}$ | $0.030_{\pm 0.000}$ | — |
| | | member 1 | $0.641_{\pm 0.001}$ | $16.84_{\pm 0.05}$ | $0.030_{\pm 0.000}$ | — |
| | | member 2 | $0.642_{\pm 0.001}$ | $16.90_{\pm 0.03}$ | $0.031_{\pm 0.000}$ | — |
| | | member 3 | $0.647_{\pm 0.002}$ | $17.05_{\pm 0.06}$ | $0.028_{\pm 0.002}$ | — |
| **Down-DE** | M=2 | ensemble | $0.620_{\pm 0.001}$ | $16.44_{\pm 0.04}$ | $0.023_{\pm 0.000}$ | $0.073_{\pm 0.001}$ |
| | | member 0 | $0.642_{\pm 0.001}$ | $16.83_{\pm 0.03}$ | $0.030_{\pm 0.000}$ | — |
| | | member 1 | $0.643_{\pm 0.001}$ | $16.84_{\pm 0.03}$ | $0.030_{\pm 0.000}$ | — |
| | M=4 | ensemble | $0.607_{\pm 0.000}$ | $16.17_{\pm 0.02}$ | $0.021_{\pm 0.001}$ | $0.073_{\pm 0.000}$ |
| | | member 0 | $0.641_{\pm 0.001}$ | $16.82_{\pm 0.03}$ | $0.030_{\pm 0.000}$ | — |
| | | member 1 | $0.642_{\pm 0.001}$ | $16.85_{\pm 0.03}$ | $0.030_{\pm 0.000}$ | — |
| | | member 2 | $0.643_{\pm 0.001}$ | $16.89_{\pm 0.04}$ | $0.031_{\pm 0.000}$ | — |
| | | member 3 | $0.643_{\pm 0.001}$ | $16.93_{\pm 0.07}$ | $0.031_{\pm 0.000}$ | — |

## J    Upstream & Downstream versus Downstream-only Ensembles

In Section 5, and Appendix L include *downstream* deep ensembles (down-DE) of V-MoE, and in some cases ViT, as a baseline. This choice was motivated by the fact that like ViT, V-MoE, and $\text{E}^3$, down-DE requires a only a single upstream checkpoint, which all of the methods more comparable. However, it is clear that using different upstream checkpoints and then further fine-tuning each of these with different random seeds to construct an *upstream* deep ensemble (up-DE) would result in more varied ensemble members and as a result, a better performing ensemble. This idea has recently been explored by Mustafa et al. (2020).

Thus, for completeness, we also investigate the effects of upstream ensembling on V-MoE. Table 15 compares the performance of upstream and downstream V-MoE ($K = 1$) ensembles of sizes $M = 2$ and $M = 4$. Across the range of metrics, for both ImageNet and ImageNet-C, for all ViT families, and for both values of $M$, we see that up-DE outperforms down-DE. In fact, up-DE with $M = 2$ is very often better than or equal to down-DE with $M = 4$. This is especially true for the diversity metrics, which indicates that diversity is indeed the driver for improved performance in up-DE. Not shown in the table is the very large computational cost associated with training upstream ensembles.

Table 15: Comparison of upstream and downstream ensembles of V-MoE with ($K = 1$).

| | | M | ImageNet | | | | | | ImageNet-C | | | | | |
| | | | NLL ↓ | Error ↓ | ECE ↓ | KL ↑ | Cos. Sim. ↓ | Norm. Dis. ↑ | NLL ↓ | Error ↓ | ECE ↓ | KL ↑ | Cos. Sim. ↓ | Norm. Dis. ↑ |
|---|---|---|---|---|---|---|---|---|---|---|---|---|---|---|
| H/14 | down-DE | 2 | 0.403 ±0.000 | 11.35 ±0.05 | 0.018 ±0.001 | 0.079 ±0.003 | 0.974 ±0.001 | 0.488 ±0.006 | 0.871 ±0.012 | 21.37 ±0.20 | **0.021** ±0.001 | 0.218 ±0.002 | 0.925 ±0.001 | 0.628 ±0.003 |
| | up-DE | 2 | 0.391 ±0.000 | 11.12 ±0.10 | 0.016 ±0.001 | **0.126** ±0.008 | **0.963** ±0.002 | 0.625 ±0.012 | 0.839 ±0.011 | 20.66 ±0.22 | 0.022 ±0.000 | **0.355** ±0.007 | **0.892** ±0.002 | 0.809 ±0.006 |
| | down-DE | 4 | 0.392 ±0.000 | 11.20 ±0.000 | 0.014 ±0.000 | 0.083 ±0.000 | 0.973 ±0.000 | 0.509 ±0.000 | 0.851 ±0.000 | 20.97 ±0.000 | **0.021** ±0.000 | 0.221 ±0.000 | 0.923 ±0.000 | 0.650 ±0.000 |
| | up-DE | 4 | **0.375** ±0.000 | **10.66** ±0.000 | **0.013** ±0.000 | **0.129** ±0.000 | **0.963** ±0.000 | **0.652** ±0.000 | **0.792** ±0.000 | **19.61** ±0.000 | 0.032 ±0.000 | **0.361** ±0.000 | **0.892** ±0.000 | **0.850** ±0.000 |
| L/16 | down-DE | 2 | 0.450 ±0.002 | 12.62 ±0.04 | 0.016 ±0.000 | 0.061 ±0.001 | 0.979 ±0.000 | 0.419 ±0.002 | 1.010 ±0.006 | 24.43 ±0.12 | **0.021** ±0.000 | 0.168 ±0.002 | 0.936 ±0.001 | 0.539 ±0.003 |
| | up-DE | 2 | 0.434 ±0.000 | 12.23 ±0.04 | **0.014** ±0.000 | 0.118 ±0.001 | 0.964 ±0.000 | 0.584 ±0.001 | 0.961 ±0.001 | 23.46 ±0.03 | 0.023 ±0.000 | **0.342** ±0.001 | **0.890** ±0.000 | 0.766 ±0.001 |
| | down-DE | 4 | 0.440 ±0.002 | 12.39 ±0.06 | 0.015 ±0.000 | 0.061 ±0.001 | 0.979 ±0.000 | 0.425 ±0.002 | 0.983 ±0.006 | 23.95 ±0.12 | **0.020** ±0.000 | 0.166 ±0.001 | 0.937 ±0.001 | 0.547 ±0.002 |
| | up-DE | 4 | **0.418** ±0.000 | **11.86** ±0.01 | **0.013** ±0.000 | **0.118** ±0.000 | **0.964** ±0.000 | **0.603** ±0.001 | **0.916** ±0.001 | **22.45** ±0.02 | 0.034 ±0.000 | **0.341** ±0.000 | **0.890** ±0.000 | **0.800** ±0.001 |
| L/32 | down-DE | 2 | 0.533 ±0.002 | 14.55 ±0.04 | 0.025 ±0.001 | 0.092 ±0.001 | 0.969 ±0.000 | 0.479 ±0.004 | 1.184 ±0.003 | 27.98 ±0.04 | 0.029 ±0.000 | 0.199 ±0.002 | 0.925 ±0.001 | 0.556 ±0.002 |
| | up-DE | 2 | 0.511 ±0.001 | 14.07 ±0.02 | 0.019 ±0.000 | **0.191** ±0.001 | **0.945** ±0.000 | 0.694 ±0.005 | 1.133 ±0.002 | 26.97 ±0.04 | **0.022** ±0.000 | **0.449** ±0.005 | **0.861** ±0.001 | 0.820 ±0.003 |
| | down-DE | 4 | 0.518 ±0.002 | 14.29 ±0.03 | 0.022 ±0.000 | 0.092 ±0.001 | 0.969 ±0.000 | 0.487 ±0.003 | 1.154 ±0.004 | 27.47 ±0.05 | 0.023 ±0.000 | 0.199 ±0.002 | 0.925 ±0.001 | 0.567 ±0.002 |
| | up-DE | 4 | **0.486** ±0.000 | **13.52** ±0.02 | **0.016** ±0.000 | **0.190** ±0.001 | **0.946** ±0.000 | **0.722** ±0.001 | **1.073** ±0.001 | **25.74** ±0.02 | 0.030 ±0.000 | **0.446** ±0.001 | **0.862** ±0.000 | **0.857** ±0.000 |
| B/16 | down-DE | 2 | 0.519 ±0.002 | 14.09 ±0.02 | 0.021 ±0.001 | 0.048 ±0.000 | 0.982 ±0.000 | 0.351 ±0.002 | 1.316 ±0.008 | 30.02 ±0.18 | 0.030 ±0.000 | 0.132 ±0.001 | 0.943 ±0.000 | 0.448 ±0.002 |
| | up-DE | 2 | 0.489 ±0.001 | 13.40 ±0.03 | **0.015** ±0.000 | 0.169 ±0.002 | 0.951 ±0.000 | 0.668 ±0.004 | 1.231 ±0.004 | 28.41 ±0.09 | **0.023** ±0.000 | **0.481** ±0.006 | **0.845** ±0.001 | 0.838 ±0.003 |
| | down-DE | 4 | 0.511 ±0.002 | 13.95 ±0.01 | 0.019 ±0.001 | 0.048 ±0.000 | 0.982 ±0.000 | 0.354 ±0.002 | 1.293 ±0.008 | 29.67 ±0.18 | 0.026 ±0.000 | 0.132 ±0.001 | 0.943 ±0.000 | 0.453 ±0.002 |
| | up-DE | 4 | **0.468** ±0.000 | **12.89** ±0.03 | 0.016 ±0.000 | **0.168** ±0.000 | 0.951 ±0.000 | **0.690** ±0.001 | **1.166** ±0.002 | **27.08** ±0.05 | 0.037 ±0.000 | 0.479 ±0.001 | **0.846** ±0.000 | **0.879** ±0.001 |
| B/32 | down-DE | 2 | 0.620 ±0.001 | 16.44 ±0.04 | 0.023 ±0.000 | 0.073 ±0.001 | 0.973 ±0.000 | 0.414 ±0.002 | 1.510 ±0.005 | 33.79 ±0.08 | 0.032 ±0.000 | 0.175 ±0.001 | 0.925 ±0.000 | 0.498 ±0.002 |
| | up-DE | 2 | 0.588 ±0.001 | 15.74 ±0.05 | **0.017** ±0.001 | **0.214** ±0.001 | **0.937** ±0.000 | 0.709 ±0.001 | 1.430 ±0.003 | 32.37 ±0.05 | **0.022** ±0.000 | **0.537** ±0.002 | **0.824** ±0.001 | 0.844 ±0.002 |
| | down-DE | 4 | 0.607 ±0.000 | 16.17 ±0.02 | **0.021** ±0.001 | 0.073 ±0.000 | 0.973 ±0.000 | 0.418 ±0.005 | 1.483 ±0.008 | 33.36 ±0.13 | 0.027 ±0.000 | 0.174 ±0.001 | 0.926 ±0.001 | 0.504 ±0.002 |
| | up-DE | 4 | **0.561** ±0.001 | **15.10** ±0.03 | 0.020 ±0.000 | **0.214** ±0.001 | **0.937** ±0.000 | **0.739** ±0.001 | **1.357** ±0.002 | **30.92** ±0.03 | 0.036 ±0.000 | **0.537** ±0.001 | **0.824** ±0.000 | **0.884** ±0.001 |
| S/32 | down-DE | 2 | 0.807 ±0.003 | 20.90 ±0.10 | **0.018** ±0.001 | 0.102 ±0.001 | 0.962 ±0.000 | 0.458 ±0.003 | 2.106 ±0.010 | 44.52 ±0.18 | 0.038 ±0.001 | 0.223 ±0.003 | 0.900 ±0.001 | 0.521 ±0.002 |
| | up-DE | 2 | 0.763 ±0.001 | 19.85 ±0.04 | **0.016** ±0.000 | 0.305 ±0.002 | **0.911** ±0.000 | 0.773 ±0.002 | 2.004 ±0.004 | 42.92 ±0.08 | **0.025** ±0.000 | **0.683** ±0.003 | **0.767** ±0.001 | 0.856 ±0.002 |
| | down-DE | 4 | 0.795 ±0.003 | 20.66 ±0.13 | **0.015** ±0.001 | 0.102 ±0.002 | 0.962 ±0.001 | 0.462 ±0.004 | 2.076 ±0.012 | 44.16 ±0.21 | 0.031 ±0.000 | 0.222 ±0.003 | 0.900 ±0.001 | 0.526 ±0.003 |
| | up-DE | 4 | **0.728** ±0.001 | **19.06** ±0.04 | 0.025 ±0.000 | **0.304** ±0.001 | **0.911** ±0.000 | **0.808** ±0.002 | **1.914** ±0.003 | **41.38** ±0.05 | 0.034 ±0.000 | **0.682** ±0.003 | **0.767** ±0.001 | **0.891** ±0.002 |

Table 16: ImageNet performance (means $\pm$ standard errors over 3 seeds) of V-MoE and $\text{E}^3$ for a S/32 backbone, without pre-training.

|  | K | NLL $\downarrow$ | ERROR $\downarrow$ |
|---|---|---|---|
| $\text{E}^3$ $(M = 2)$ | 1 | **1.420** $_{\pm\ 0.007}$ | **23.78** $_{\pm\ 0.22}$ |
| V-MoE | 2 | 1.478 $_{\pm\ 0.003}$ | 24.45 $_{\pm\ 0.08}$ |

## K  Preliminary ImageNet Results without Pre-training

Training large-scale sparse MoEs on datasets such as ImageNet and in absence of any pre-training is a difficult task. Indeed, the massive number of parameters causes models to severely overfit in that regime. In practice, we need to combine various regularization techniques to control this overfitting.

To obtain preliminary results of $\text{E}^3$ trained from scratch on ImageNet, we adapt the recipe proposed in the code released by Riquelme et al. (2021). The recipe is itself based on the regularisation protocol of Steiner et al. (2021) ("AugReg"), which trained performant dense ViT models from scratch on ImageNet, without pre-training. Overall, compared to the default training schema used for this paper, we add:

- Mixup (Zhang et al., 2018),

- Weight decay,

- Dropout (Srivastava et al., 2014) on expert MLPs.

The baseline configuration for V-MoE was tuned according to the hyperparameter search space defined by Steiner et al. (2021), with an extra sweep over expert MLP dropout in $\{0.1, 0.2\}$. Selecting according to validation accuracy, the optimal V-MoE setting was `medium2`, i.e., mixup ratio 0.5 and RandAugment (Cubuk et al., 2020) parameters 2 and 15 (2 augmentations applied of magnitude 15), alongside expert dropout 0.2. For $\text{E}^3$, these regularization-related hyperparameters were kept fixed and were not further tuned. More precisely, we keep `medium2` and tune only the learning rate in $\{0.001, 0.003\}$; we thus suspect we could improve our current results for $\text{E}^3$ by considering a broader sweep of hyperparameters (e.g., re-tuning all the regularization-related hyperparameters).

In Table 16, we report the performance for both V-MoE and $\text{E}^3$ with a S/32 backbone. In terms of both NLL and classification error, $\text{E}^3$ outperforms V-MoE. The model checkpoint, along with our code, can be found at https://github.com/google-research/vmoe.

Code can be found at https://github.com/google-research/vmoe.

## L  Additional Experimental Results

In this section we expand on various experiments presented in Sections 3 to 5. In experiments considering multiple ViT families we also include B/16 which was excluded from the main text for clarity.

### L.1  Static versus Adaptive Combination

Here we continue the investigation into static versus adaptive combination from Section 3.

**Individual gains with respect to $E, K$ and $M$.**  Figure 12 shows the effect of increasing the the various 'ensemble size' parameters for a deep ensemble of V-MoEs. In particular, we investigate the static combination axis $M$ (the number of ensemble members), as well as the two adaptive axes—$K$ (the number of experts chosen per patch) and $E$ (the total number of experts).

When investigating the effect of $K$, we fix $E = 32$ and average over $M \in \{1, .., 8\}$. Similarly, when investigating $M$, we fix $E = 32$ and average over $K \in \{1, .., 8\}$. When investigating the effect of $E$ we fix $K = 2$

and average over $M \in \{1, .., 8\}$. As a result of this procedure the exact values of the curves are not directly comparable. However, we can still examine the relative trends.

Specifically, we note that while the variation in $K$ and $M$ curves is roughly of the same size, the variation in the $E$ curve is smaller. We also note that there is very little variation beyond $E = 8$ (note the difference in the scales of the axes for the curves). These observations motivate the design of $\text{E}^3$, where we split the sub-models along the $E$ axis, in order to better take advantage of the experts.

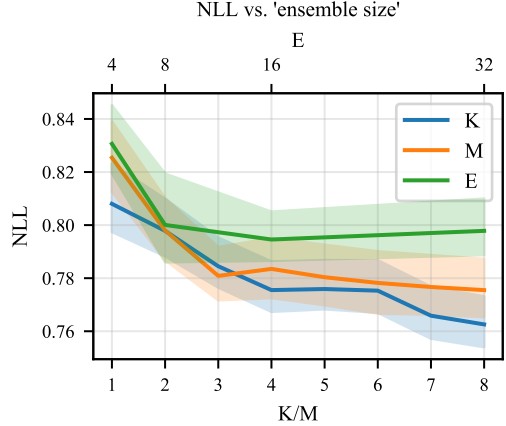

Figure 12: Comparison for the impact on ImageNet NLL of variations in $K$, $E$ and $M$. The underlying model is ViT-S/32.

**Extended Results for the Cumulative Effects of Static and Adaptive Combination.** In Figure 13 we extend the ImageNet NLL results, presented in Figure 3, to a range of other datasets and performance metrics. We see that in most cases, the trends are the same as before. That is, (static) ensembling tends to improve the performance of ViT and V-MoE equally. The two exceptions are ECE and OOD detection for ViT-S/32 where we see that larger ensemble sizes can result in decreased performance. These results indicate that the for small ViT models, larger ensembles can have slightly lower quality uncertainty estimates. The trend for ImageNet-C performance is also not as consistent with ensembling sometimes helping ViT or V-MoE less (as indicated by the changes in ordering on the y-axis).

## L.2 An Additional Motivating Experiment – Deep Ensembles of V-MoEs with Fewer Experts

As an additional motivation for combining sparse MoEs and ensembles, Table 17 compares the performance of a V-MoE with $E = 32$ total experts and ensembles of V-MoEs with ($M = 2$, $E = 16$) and ($M = 4$, $E = 8$), for both $K = 1$ and $K = 2$. We see that, in terms of NLL, ($M = 4$, $E = 8$) is better than ($M = 2$, $E = 16$) which is in turn better than ($M = 1$, $E = 32$). We see similar results for Error and ECE.

We note that this result is especially remarkable since the *individual* upstream (and later, downstream) models are such that $E = 8$ is worse than $E = 16$ which in turn performs worse than $E = 32$; ensembling thus manages to counterbalance the poorer individual model performance. This suggests that the efficient ensembling—i.e., the combination of multiple models within a single model—of sparse MoEs could lead to strong performance while reducing computational costs.

## L.3 Extended Results for the Tiling with Increasing Parameter Sharing Ablation

In Table 18, we extend the results for our parameter sharing ablation in Section 4.2.3, from $K = 2$ to $K = 1$. We see that the results remain the same in this case.

Table 17: Comparison of upstream deep ensembles of V-MoE-B/32 models with fewer experts.

|  | K | M | E | NLL ↓ | ERROR ↓ | ECE ↓ | KL ↑ |
|---|---|---|---|---|---|---|---|
| | | 1 | 32 | 0.642 ± 0.002 | 16.90 ± 0.05 | 0.029 ± 0.001 | − |
| | 1 | 2 | 16 | 0.588 | 15.97 | 0.015 | 0.211 |
| V-MoE | | 4 | 8 | 0.577 | 15.82 | 0.017 | 0.228 |
| | | 1 | 32 | 0.638 ± 0.001 | 16.76 ± 0.05 | 0.033 ± 0.001 | − |
| | 2 | 2 | 16 | 0.583 | 15.75 | 0.016 | 0.211 |
| | | 4 | 8 | 0.580 | 15.94 | 0.015 | 0.184 |

Table 18: Extension of Table 3, showing the impact of parameter sharing in $\text{E}^3$, for $K = 1$.

| OVERLAP | NLL ↓ | ERROR ↓ | ECE ↓ | KL ↑ |
|---|---|---|---|---|
| 0 $(=\text{E}^3)$ | **0.622** ± 0.001 | **16.70** ± 0.03 | **0.018** ± 0.000 | **0.217** ± 0.003 |
| 2 | 0.627 ± 0.003 | 16.83 ± 0.07 | 0.022 ± 0.001 | 0.194 ± 0.005 |
| 4 | 0.634 ± 0.002 | 16.92 ± 0.07 | 0.024 ± 0.001 | 0.178 ± 0.004 |
| 8 | 0.642 ± 0.001 | 17.04 ± 0.10 | 0.028 ± 0.001 | 0.151 ± 0.009 |
| 16 | 0.659 ± 0.004 | 17.28 ± 0.12 | 0.036 ± 0.001 | 0.103 ± 0.009 |

## L.4 Extended Results for Few-shot Learning

In Figure 14, we extend the few-shot learning results of Figure 4 to also include 1, 5, and 25-shot. Additionally, we show results for the weighted aggregation strategy mentioned in Appendix A.5.

We confirm the result that few-shot performance for $\text{E}^3$ gets better, relative to the other baselines, with larger ViT families. Additionally, we see that $\text{E}^3$ performance seems to get better, again relative to the other baselines, with more shots. This phenomenon can most easily be noticed by comparing the results for S/32 across the different numbers of shots. Finally, we see that the trends with and without the weighted mean are the same.

## L.5 Extended Results for OOD Detection

Here we extended the OOD results of Figure 6. Specifically, we add CIFAR100 as an in-distribution dataset and Describable Textures Dataset (DTD) (Cimpoi et al., 2014) as an OOD dataset. We also add area under the receiver operating characteristic (AUC (ROC)) and area under the precision-recall curve (AUC (PR)) as metrics. Figures 15 and 16 contain the results with CIFAR10 and CIFAR100 as the in-distribution datasets, respectively.

As in Figure 6, we see that $\text{E}^3$ performs better (relative to the other baselines) for larger ViT families. Furthermore, $\text{E}^3$ seems to perform better in near OOD detection (i.e. CIFAR10 versus CIFAR100, and vice versa) than far OOD detection. Finally, we see that these trends are consistent across OOD metrics.

## L.6 Extended Results for ImageNet

In this section we extend the results for ImageNet and the corrupted variants presented in Figures 4, 5 and 7. In addition to NLL, classification error, ECE (for standard ImageNet), and Brier score, Figure 17 provides classification error and ECE for all ImageNet variants.

Most of the trends observed in Section 5 remain true:

- $\text{E}^3$ tends to be Pareto efficient in the presence of distribution shift.

- For smaller ViT families, V-MoE outperforms ViT in the presence of distribution shift.

- E³ improves ECE over ViT and V-MoE.

- E³ improves classification performance.

- ViT consistently provides better ECE than V-MoE.

However, there are some exceptions:

- **ImageNet-A classification error.** All models (including E³) under-perform relative to ViT-S/32 and ViT-H/14.

- **ECE for ImageNet-C, ImageNet-A, and ImageNet-V2.** Interestingly for the non-standard ImageNet variants, and in particular for ImageNet-A, there is a strong correlation between lower ECE and larger ViT families.

We also find that the results for classification error and Brier score follow those for NLL closely.

## L.7   Additional CIFAR10, CIFAR100, Flowers, and Pets Results

Here we extend the results for ImageNet and the corrupted variants presented in Figures 4, 5 and 7 to four additional datasets. Figures 18, 19, 20 and 21 provide results for CIFAR10, CIFAR100, Oxford Flowers 102, and Oxford IIIT Pet, respectively. As in Appendix L.6, we find that the results are similar to those in Section 5.

Compared to ImageNet, for CIFAR10, CIFAR10-C, and CIFAR100, E³ seems to perform even better relative to the other baselines. Note, for example, that E³ is Pareto efficient (even for S/32) in the cases of CIFAR10-C and CIFAR100 NLL. As in Appendix L.6, we see that the ECE has a stronger downward trend with respect to increased ViT family size for shifted test data.

For Flowers and Pets, where we only have results for smaller ViT families, E³ seems to under perform. However, the performance for L/32 is better than for S/32 and B/32 which suggests that the results for these datasets are consistent with the other datasets presented in this work and, therefore, that we should expect E³'s predictive performance to keep improving with larger models.

## L.8   Efficient ensemble of experts and V-MoE with larger values of $K$ and $M$

Figure 22 and Figure 23 show the effect of varying $K$ on E³ and V-MoE, and the effect of varying $M$ on E³, respectively. We make the following observations:

- In almost all cases, increasing $K$ or $M$ does not result in Pareto efficient models.

- For V-MoE, increasing $K$ seems to help in most cases, except for ECE performance where it usually hurts.

- For E³, going from $K = 1$ to $K = 2$ seems to help in most cases but going from $K = 2$ to $K = 4$ usually hurts. Going from $K = 1$ to $K = 4$ still helps but to a lesser extent than from $K = 1$ to $K = 2$.

- For E³, increasing $M$ either doesn't make a consistent and significant difference or hurts (e.g. in OOD detection).

These conclusions should, however, be considered with caution. Recall that the upstream checkpoints used for fine-tuning all V-MoE and E³ models in this work are V-MoE models with $K = 2$. Thus, the results in this experiment are confounded by upstream and downstream checkpoint mismatch for all E³ models and all V-MoE models with $K \neq 2$. This phenomenon was observed in Riquelme et al. (2021) for V-MoE models. I.e., performance of downstream V-MoE models with mismatched values of $K$ were relatively worse than those with matched values of $K$; see their Appendix E.4 (Figures 33-35). We also hypothesise that it is more

difficult to train downstream $\text{E}^3$ models with larger values of $M$ from upstream V-MoE models because in each subset of the experts some common expert specialisations will need to be duplicated. Our ensemble members are fine-tuned from an upstream V-MoE with a predefined total number of experts ($E = 32$) meaning that increasing $M$ decreases the number of experts available to each ensemble member (with $E/M$ experts per member). This could also impact the performance of $\text{E}^3$ with larger sizes of $M$.

### L.8.1 Upstream vs. Downstream Mismatch

Table 19: ImageNet performance of $\text{E}^3$ models fine-tuned from V-MoE-B/32 checkpoints with $K = 2$ or $K = 4$, and $E = 32$. $K_{\text{upstream}} = 2$ results are averaged over 8 random seeds, while $K_{\text{upstream}} = 4$ results are averaged over 3 seeds.

|  | K | M | $K_{\text{upstream}}$ | NLL ↓ | Error ↓ |
|---|---|---|---|---|---|
| $\text{E}^3$ | 1 | 2 | 2 | **0.622** ± 0.001 | **16.70** ± 0.03 |
|  |  |  | 4 | **0.622** ± 0.001 | 16.81 ± 0.05 |
|  | 1 | 4 | 2 | 0.624 ± 0.001 | 16.99 ± 0.03 |
|  |  |  | 4 | **0.622** ± 0.000 | **16.93** ± 0.01 |

Here we investigate whether upstream versus downstream mismatch partially explains the counter-intuitive result that increasing $M$ can result in worse performance for $\text{E}^3$. We train downstream $\text{E}^3$ models with ($K = 1$, $M = 2$) and ($K = 1$, $M = 4$) from upstream V-MoE checkpoints with $K = 2$ and $K = 4$. Table 19 shows the results. We see that ($K = 1$, $M = 2$) does not seem to benefit from increasing $K_{\text{upstream}}$ from 2 to 4, despite the fact that the upstream $K = 4$ model is better than the upstream $K = 2$ model (NLL upstream is 8.18 for $K = 4$ and 8.27 for $K = 2$). On the other hand, ($K = 1$, $M = 4$) does benefit from having K=4 upstream.

This confirms that the mismatch between upstream and downstream models is one of the factors explaining the results of $\text{E}^3$ for growing $M$. However, we also see that the best performing model is ($K_{\text{upstream}} = 2$, $K = 1$, $M = 2$), which suggests that there are other confounding factors, such as those mentioned above.

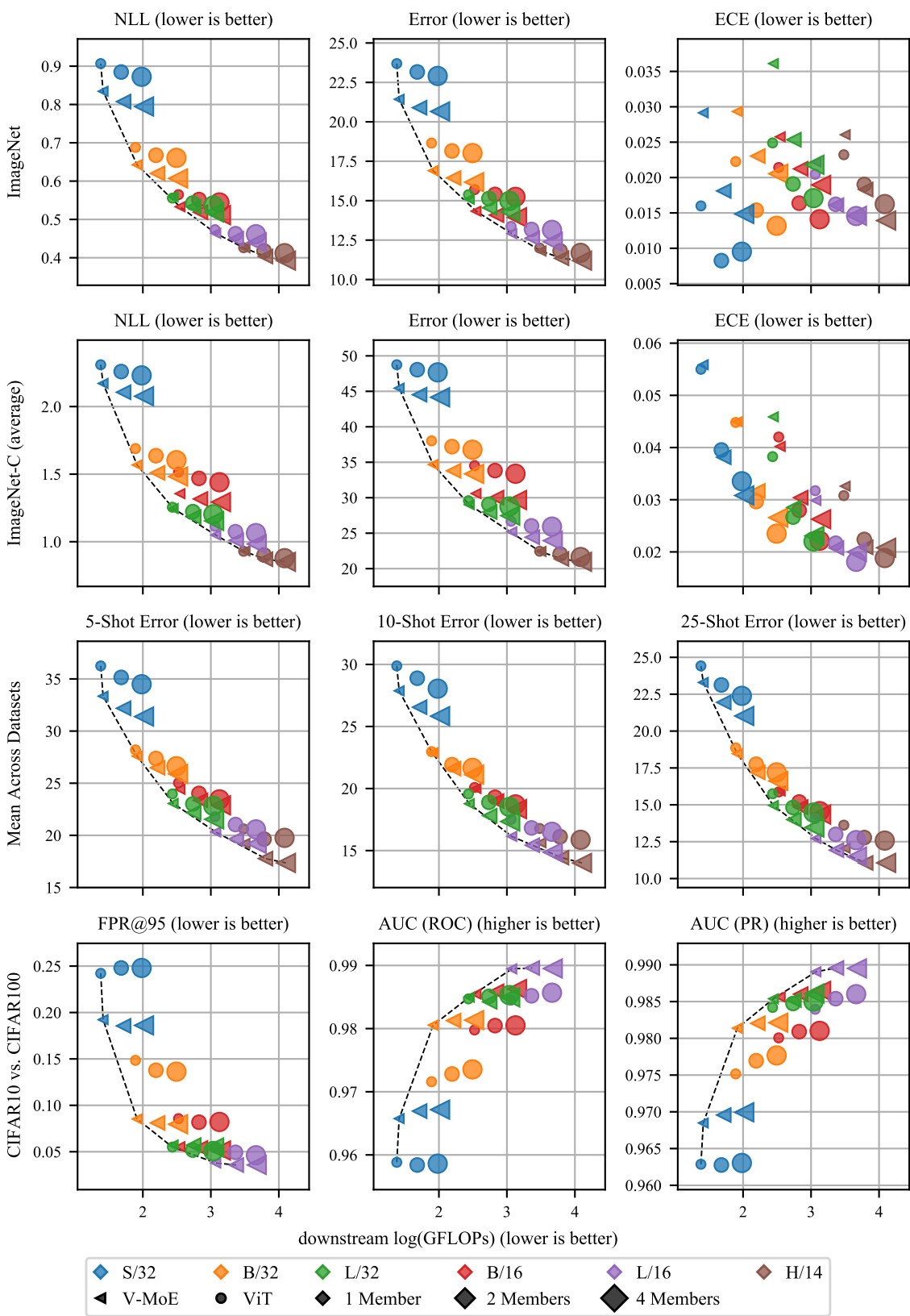

Figure 13: Extended results for Figure 3 to a selection of other tasks and metrics. We see that in most cases, ensembles tend to help ViT and V-MoE ($K = 1$) equally.

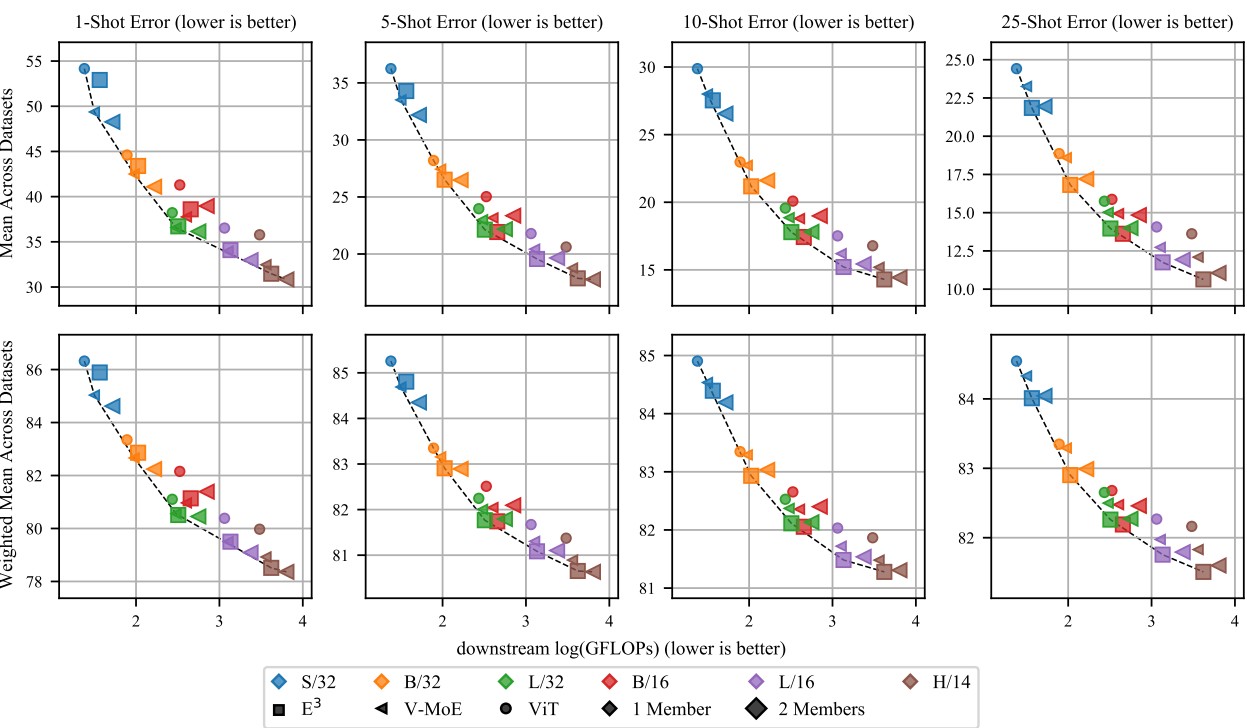

Figure 14: Extended few-shot results from Figure 4 with an additional aggregation method and numbers of shots.

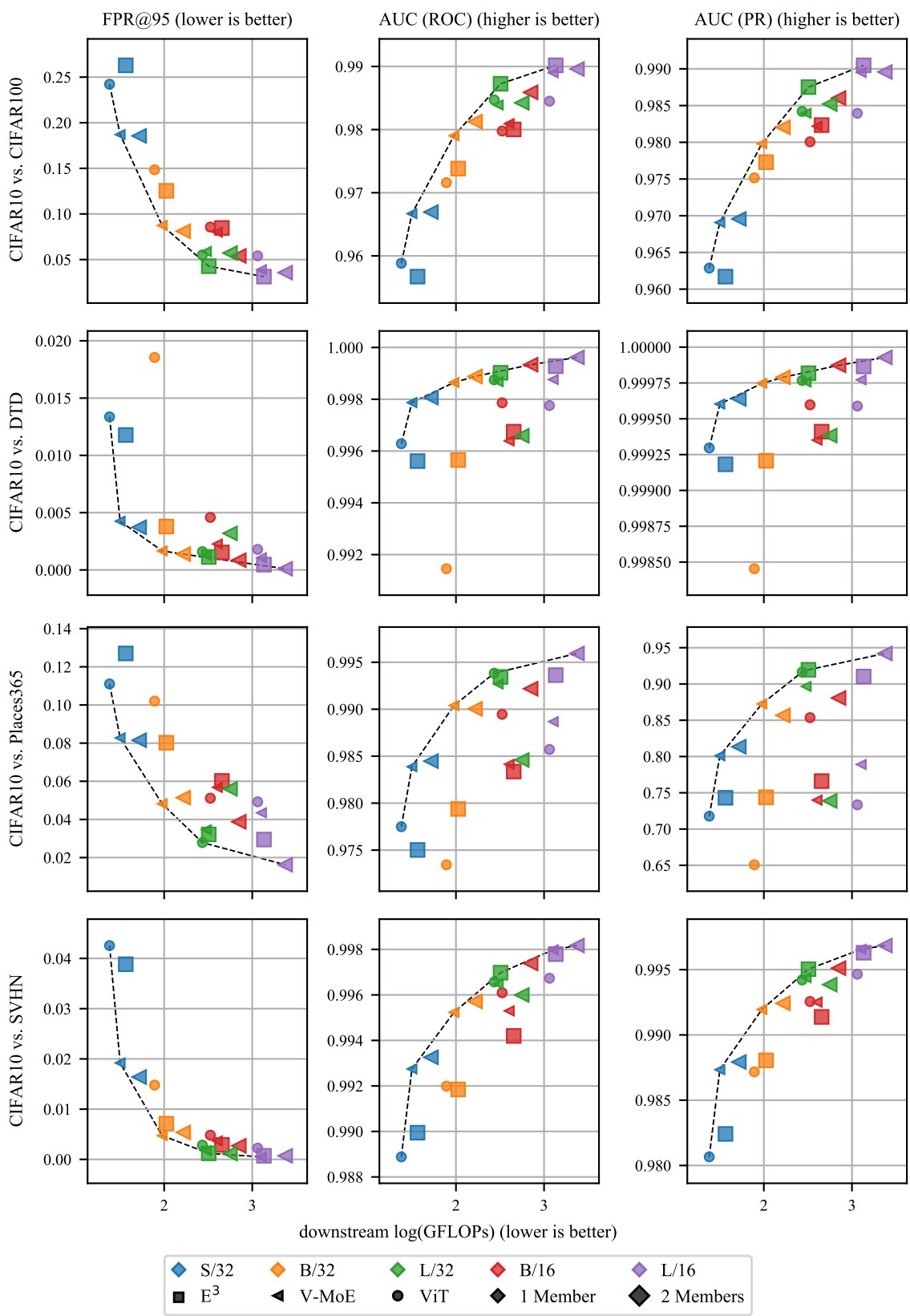

Figure 15: Extended OOD detection the results from Figure 6 with an additional OOD dataset and more metrics.

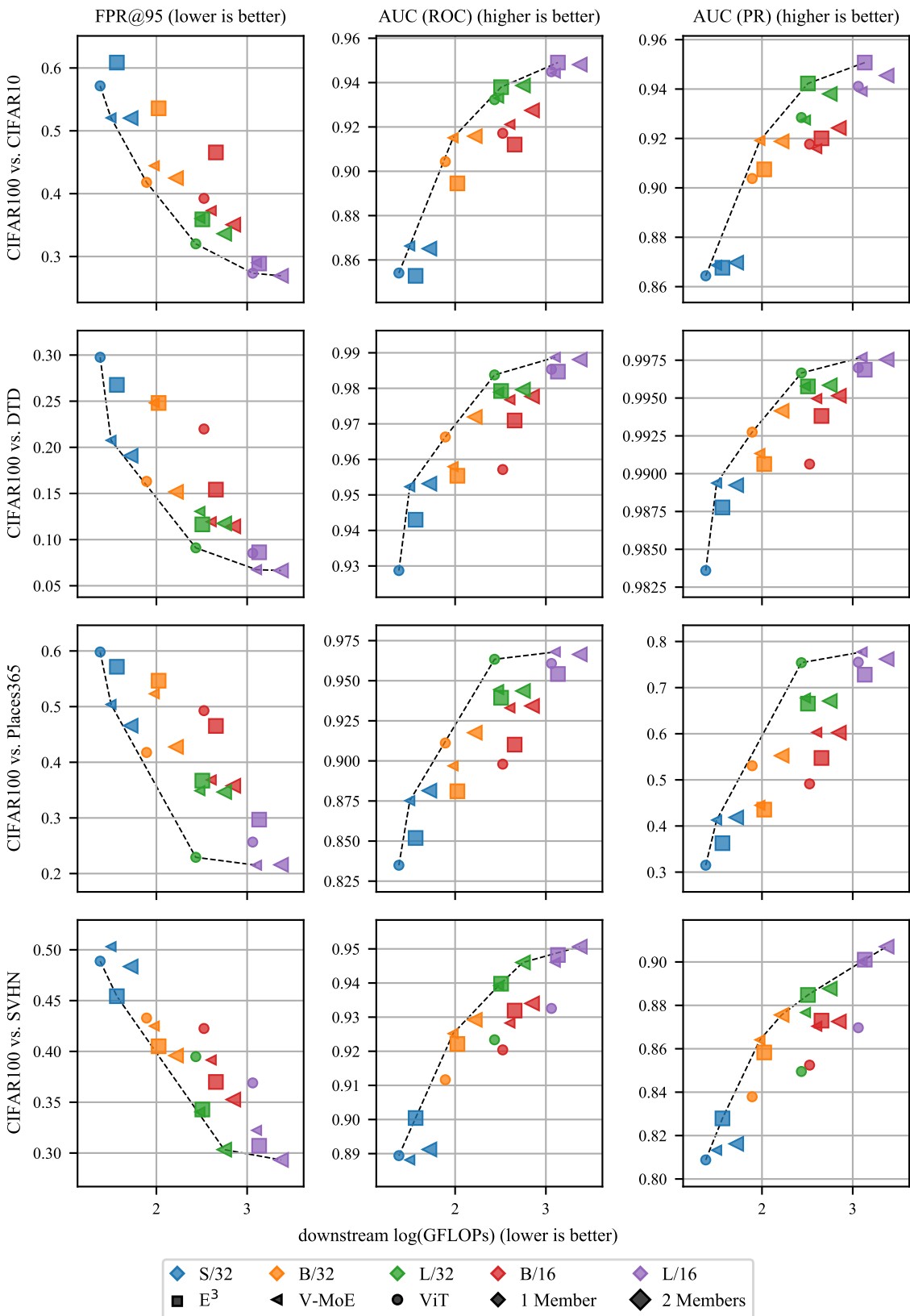

Figure 16: Extended OOD detection the results from Figure 6 with CIFAR100 as the in-distribution dataset, an additional OOD dataset, and more metrics.

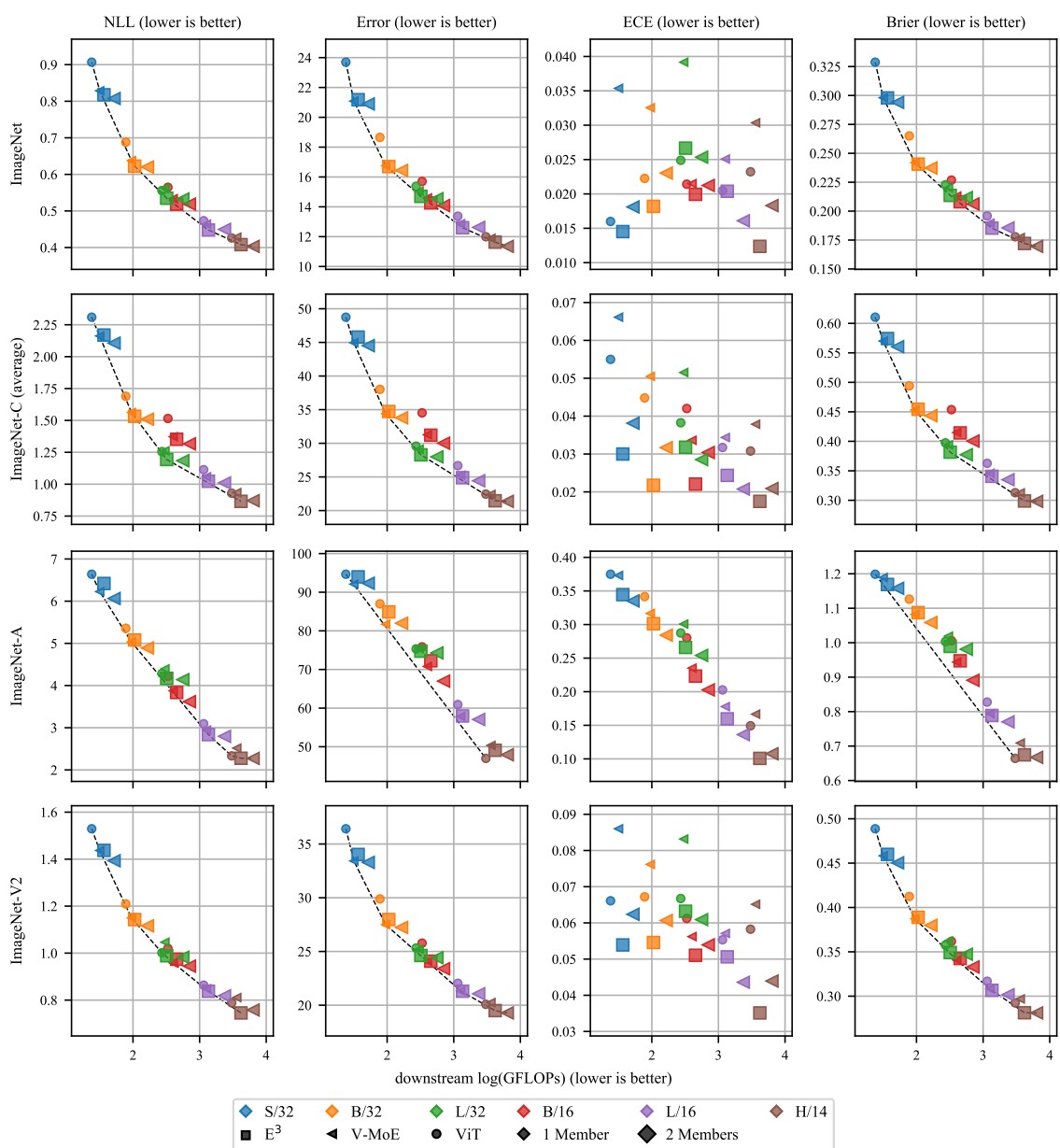

Figure 17: Extended results from Figures 4, 5 and 7 with additional metrics.

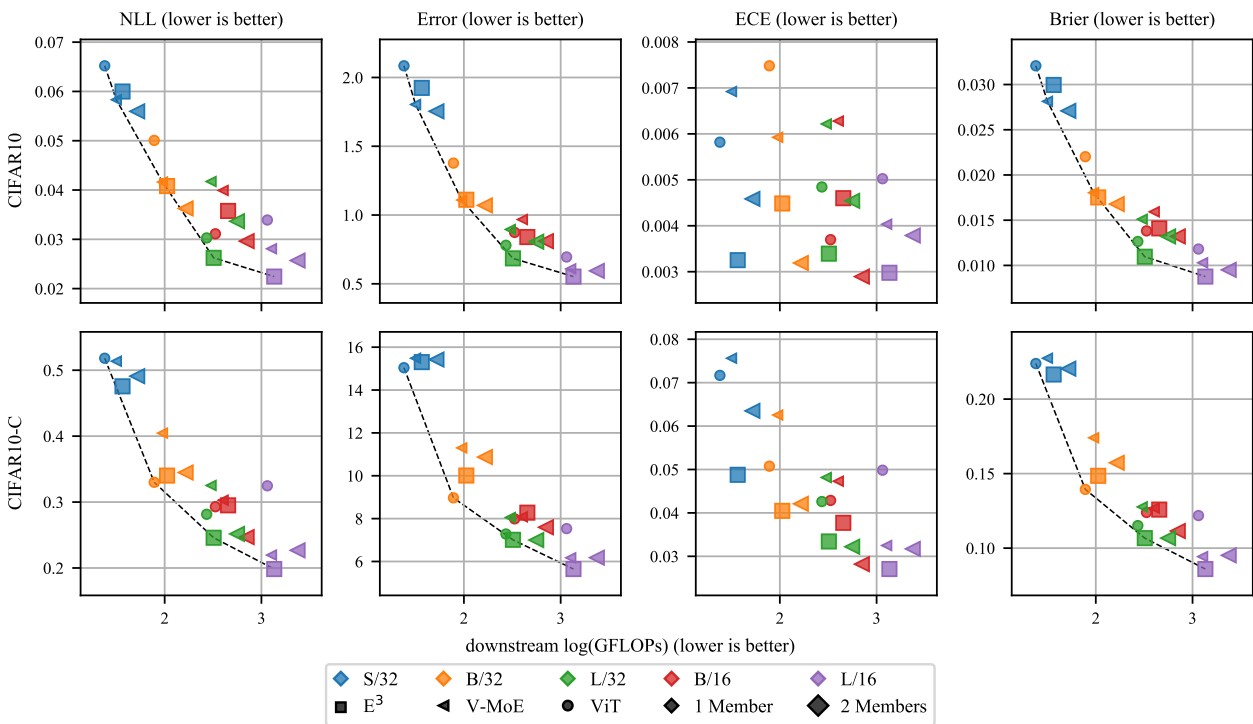

Figure 18: Results for CIFAR10 and CIFAR10-C.

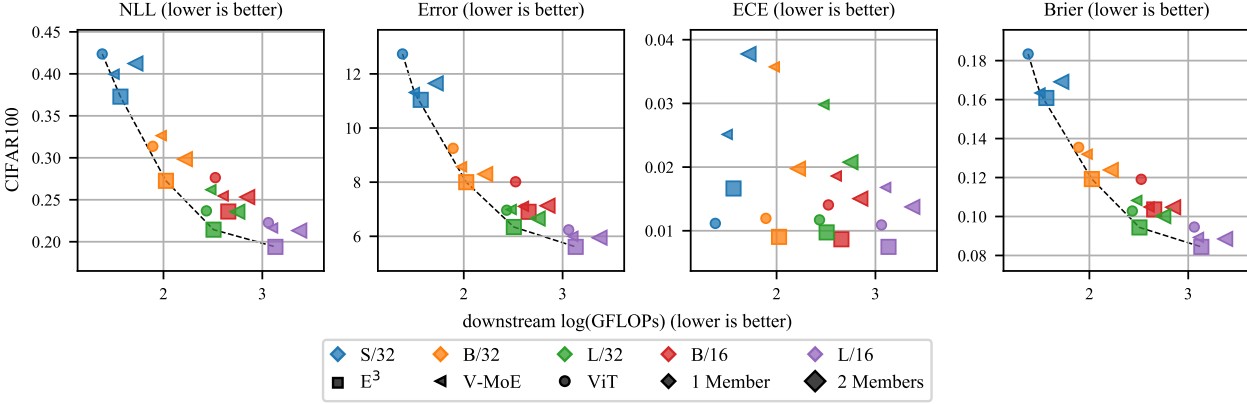

Figure 19: Results for CIFAR100.

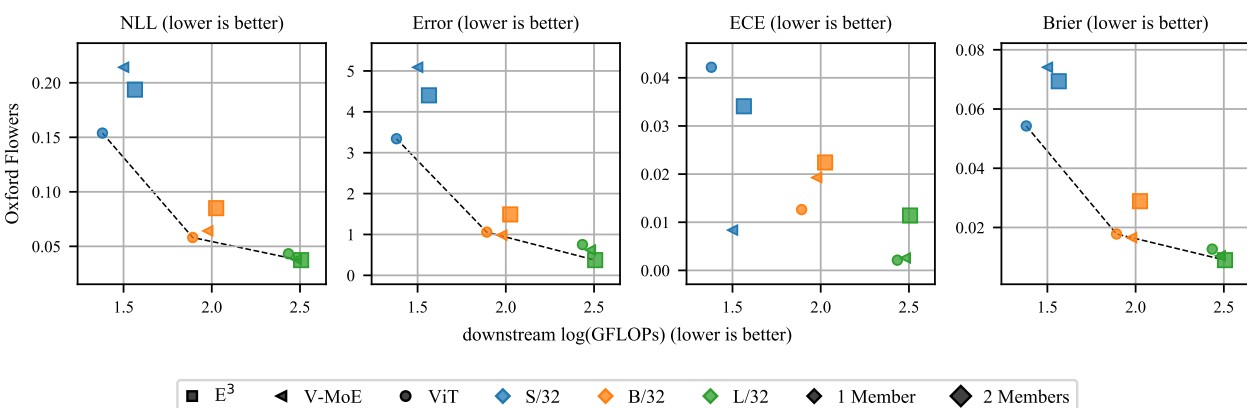

Figure 20: Results for Oxford Flowers 102.

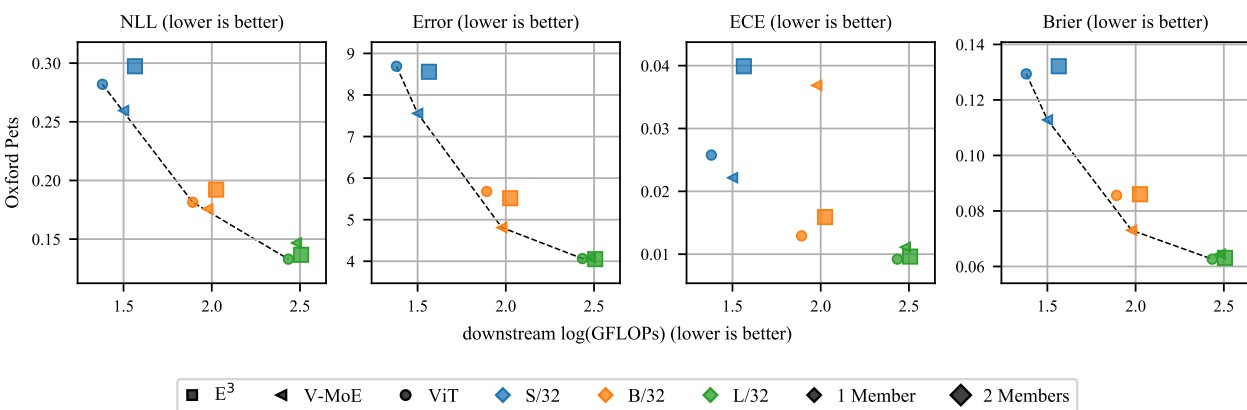

Figure 21: Results for Oxford IIIT Pet.

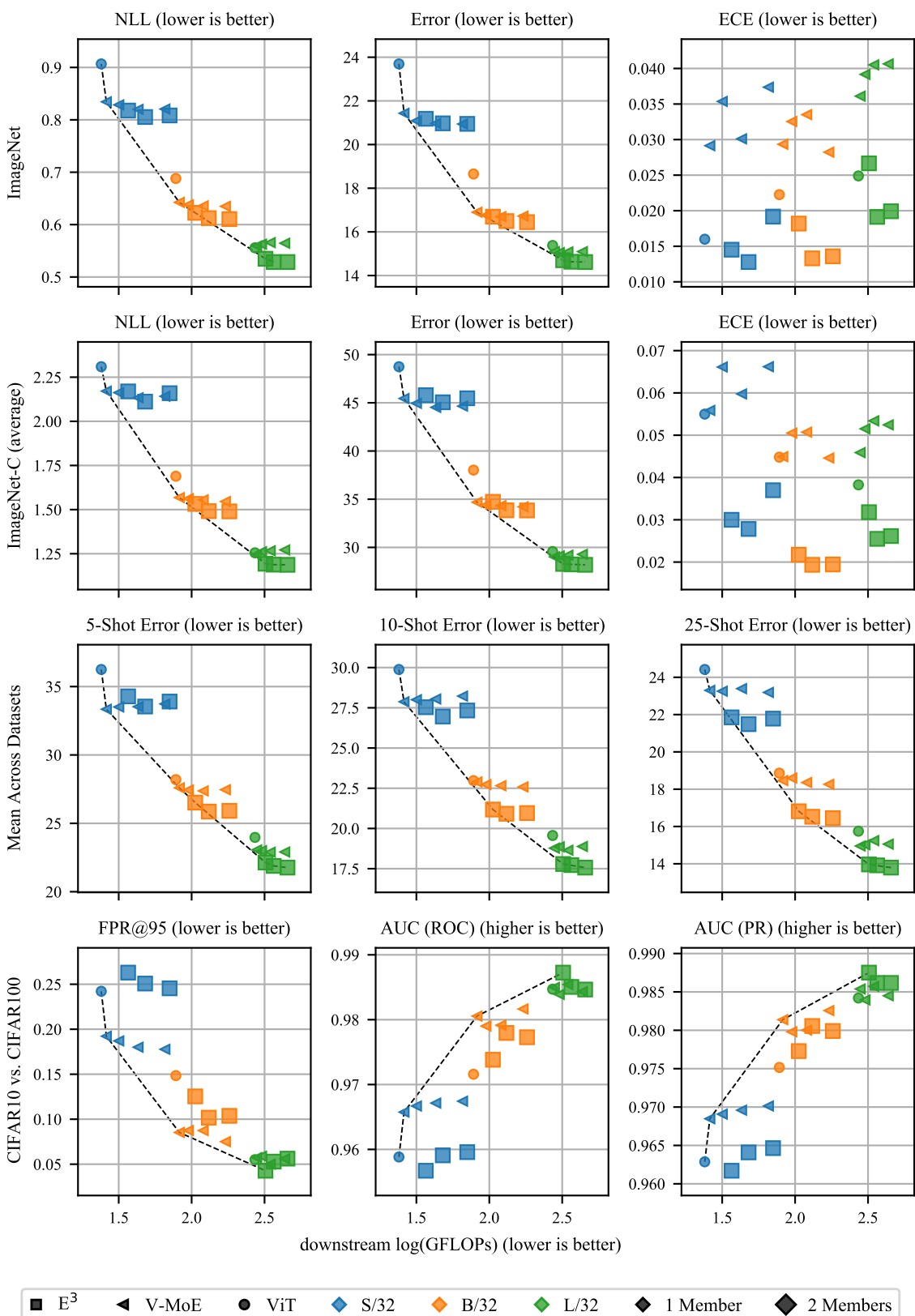

Figure 22: Results for V-MoE with $K \in \{1, 2, 4, 8\}$ and $\text{E}^3$ with $K \in \{1, 2, 4\}$. Models with larger values of $K$ have larger FLOPs.

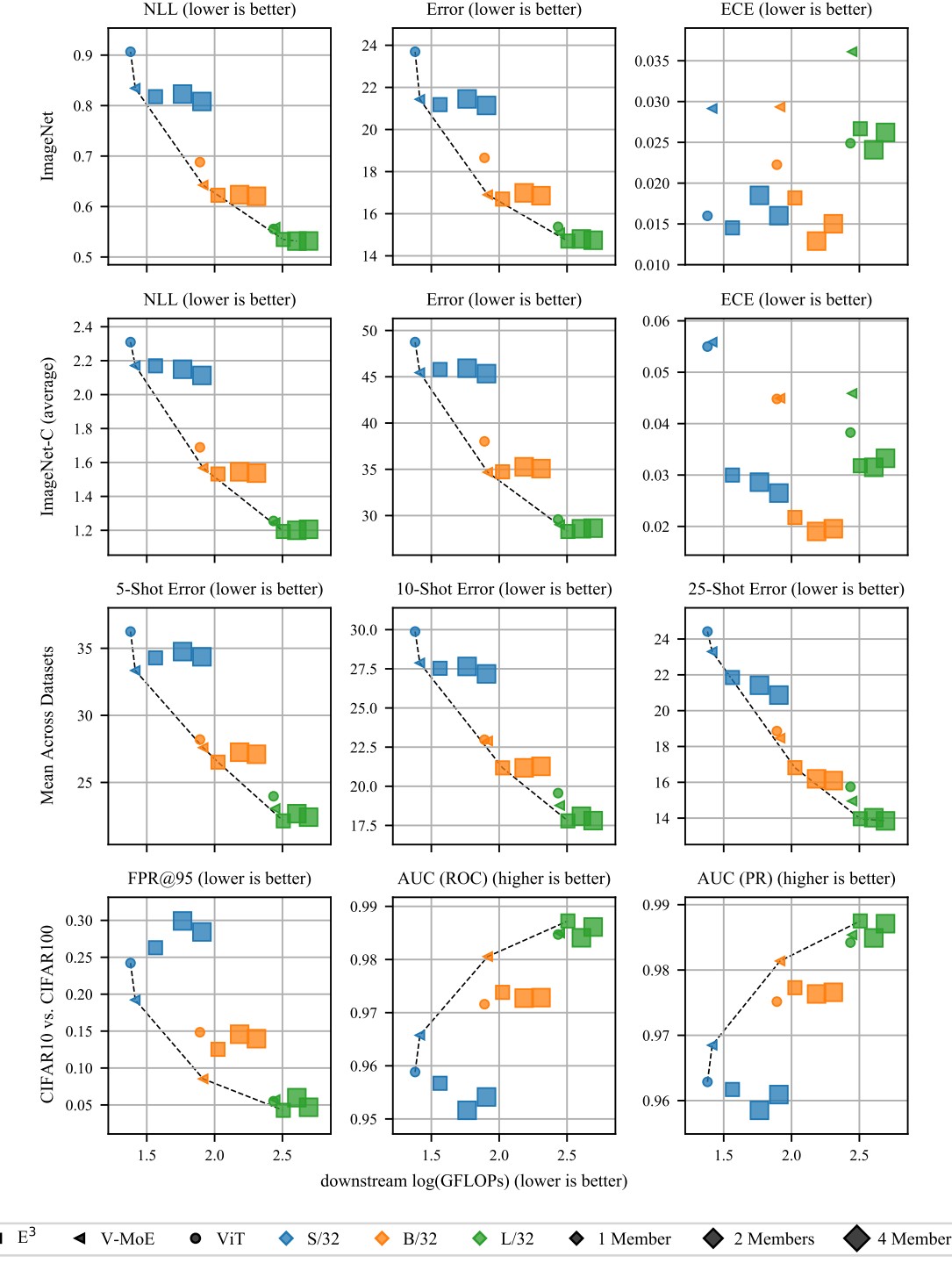

Figure 23: Results for $\mathrm{E}^3$ with $M = 4$ and $K \in \{1, 2\}$.

## M    Results Tables

### M.1    FLOPs Numbers

Table 20 provides the downstream training FLOPs for various $E^3$, V-MoE, and ViT configurations. These numbers correspond to the x-values of the points in the figures presented in Section 5 and Appendix L. Table 21 provides the percentage difference in FLOPs between the $E^3$, V-MoE and down-DE models most commonly used in this work. Note that the percentage differences for H/14 do not follow the trend of the other sizes, e.g. that the percentage difference between $E^3$ and V-MoE gets smaller for larger sizes, due to the fact that for H/15 we use a last-5 configuration rather than the last-2 configuration used for the other ViT families. Table 22 provides the downstream training FLOPs for the various ablation study models presented in Section 4.2.[2]

Table 20: Downstream training GFLOPs for the various $E^3$, V-MoE, and ViT baselines used in this work.

|  | K | M | S/32 | B/32 | B/16 | L/32 | L/16 | H/14 |
|---|---|---|---|---|---|---|---|---|
| | 1 | 2 | 36.69 | 105.89 | 452.62 | 320.92 | 1356.46 | 4183.28 |
| | 1 | 4 | 58.03 | 152.98 | — | 403.77 | — | — |
| $E^3$ | 2 | 2 | 47.98 | 131.06 | 552.65 | 365.42 | 1533.74 | — |
| | 2 | 4 | 80.66 | 203.31 | — | 492.77 | — | — |
| | 4 | 2 | 70.61 | 181.40 | — | 454.43 | — | — |
| | - | 1 | 24.01 | 77.97 | 334.48 | 271.83 | 1151.71 | 3033.60 |
| ViT | - | 2 | 48.02 | 155.93 | 668.95 | 543.66 | 2303.43 | 6067.21 |
| | - | 4 | 96.03 | 311.87 | 1337.90 | 1087.33 | 4606.85 | 12134.41 |
| | 1 | 1 | 26.02 | 82.35 | 351.91 | 279.55 | 1182.08 | 3179.90 |
| | 1 | 2 | 52.04 | 164.70 | 703.81 | 559.10 | 2364.16 | 6359.81 |
| V-MoE | 1 | 4 | 104.07 | 329.41 | 1407.63 | 1118.21 | 4728.32 | 12719.61 |
| | 2 | 1 | 31.66 | 94.93 | 401.98 | 301.75 | 1270.78 | 3617.94 |
| | 4 | 1 | 42.95 | 120.11 | 501.99 | 346.25 | 1448.06 | — |
| | 8 | 1 | 65.59 | 170.44 | — | 435.26 | — | — |

Table 21: Percentage difference in downstream training FLOPs for $E^3$ with $(K, M) = (1, 2)$ compared with V-MoE with $K = 1$ and an ensemble of two such V-MoE members.

|  | S/32 | B/32 | B/16 | L/32 | L/16 | H/14 |
|---|---|---|---|---|---|---|
| $E^3$ vs. V-MoE | 41.01 | 28.58 | 28.62 | 14.80 | 14.75 | 31.55 |
| $E^3$ vs. down-DE | -29.49 | -35.71 | -35.69 | -42.60 | -42.62 | -34.22 |

### M.2    Parameter Counts

Table 23 compares the parameter counts for ViT and V-MoE/$E^3$ models in each ViT family.

### M.3    Summary for NLL under Distribution Shift

Table 24 shows the percentage improvement for $E^3$ versus V-MoE in NLL (i.e., $\frac{\text{NLL}_{\text{V-MoE}} - \text{NLL}_{E^3}}{\text{NLL}_{\text{V-MoE}}} \times 100$, with positive values thus indicating that $E^3$ improves upon V-MoE) averaged over ImageNet-C, ImageNet-A, and ImageNet-V2, for a given ViT family and $(K, M)$ in $\{(1, 2), (2, 2)\}$ (compared to V-MoE with $K = 2$ and

---

[2]Note that the $E^3$ and V-MoE results here are different to those in Table 20 due to a difference in implementation. We used a simplified, but less computationally efficient expert-routing implementation for all of the ablation studies. As a result the V-MoE and $E^3$ FLOPs in Table 20 are lower and cannot be fairly compared with the ablation models presented here. We thus re-benchmarked $E^3$ and V-MoE to obtain comparable results.

Table 22: Downstream training GFLOPs comparison for the ablation study models in Section 4.2.

|  | K | M | GFLOPs |
|---|---|---|---|
| V-MoE | 2 | — | 96.895 |
| V-MoE | 4 | — | 123.644 |
| V-MoE | 8 | — | 178.133 |
| $\textsc{e}^3$ | 1 | 2 | 109.781 |
| $\textsc{e}^3$ | 2 | 2 | 138.457 |
| $\textsc{e}^3$ | 4 | 2 | 196.870 |
| Tiling | 2 | 2 | 138.460 |
| Partitioning | 2 | — | 97.885 |
| Overlap = 2 | 2 | 2 | 138.457 |
| Overlap = 4 | 2 | 2 | 138.458 |
| Overlap = 8 | 2 | 2 | 138.459 |
| Overlap = 6 | 2 | 2 | 138.460 |
| Multi-pred | 2 | — | 96.330 |
| Multi-pred | 4 | — | 122.526 |
| Multi-pred | 8 | — | 175.889 |

Table 23: Parameter counts for ViT vs V-MoE and $\textsc{e}^3$.

|  | S/32 | B/32 | B/16 | L/32 | L16 | H/14 |
|---|---|---|---|---|---|---|
| ViT | 36.5M | 102.1M | 100.5M | 325.3M | 323.1M | 655.8M |
| V-MoE/$\textsc{e}^3$ | 166.7M | 395.0M | 393.3M | 845.8M | 843.6M | 2688.6M |

$K = 4$, respectively). We see that for all ViT families except S/32, $\textsc{e}^3$ outperforms V-MoE. This result is not unexpected, since ensembles tend to provide improved performance under distribution shift relative to single models (Lakshminarayanan et al., 2017; Ovadia et al., 2019; Havasi et al., 2020).

Table 24: Percentage improvement for $\textsc{e}^3$ vs V-MoE in NLL under distribution shift, averaged over ImageNet-C, ImageNet-A, and ImageNet-V2.

|  | S/32 | B/32 | B/16 | L/32 | L/16 | H/14 |
|---|---|---|---|---|---|---|
| $\textsc{e}^3$ $(K = 1, M = 2)$ vs. V-MoE $(K = 2)$ | -1.15 | 0.52 | 0.31 | 5.34 | 2.78 | 8.33 |
| $\textsc{e}^3$ $(K = 2, M = 2)$ vs. V-MoE $(K = 4)$ | -0.21 | 3.91 | 1.33 | 7.37 | 2.27 | — |

### M.4 Auxiliary Tables with Standard Errors.

The tables in this section provide the mean values and corresponding standard errors for many of the results depicted in figures throughout Section 5 and Appendix L.

Table 25: ImageNet comparison of V-MoE, downstream ensembles there-of, and E³ with 2 experts per input in each case.

| | | K | M | IMAGENET NLL ↓ | ERROR ↓ | ECE ↓ | IMAGENET-C (AVERAGE) NLL ↓ | ERROR ↓ | ECE ↓ | IMAGENET-A NLL ↓ | ERROR ↓ | ECE ↓ | IMAGENET-V2 NLL ↓ | ERROR ↓ | ECE ↓ | DOWNSTREAM ΔFLOPs (%) ↓ |
|---|---|---|---|---|---|---|---|---|---|---|---|---|---|---|---|---|
| | E³ | 1 | 2 | 0.408 ±0.001 | 11.63 ±0.05 | **0.012** ±0.000 | **0.865** ±0.010 | **21.46** ±0.16 | **0.018** ±0.000 | **2.276** ±0.042 | 49.09 ±0.78 | **0.101** ±0.003 | **0.745** ±0.001 | 19.50 ±0.08 | **0.035** ±0.001 | **15.63** |
| H/14 | down-DE | 1 | 2 | **0.403** ±0.000 | **11.35** ±0.05 | 0.018 ±0.001 | 0.871 ±0.012 | 21.37 ±0.20 | 0.021 ±0.001 | 2.273 ±0.049 | **47.93** ±0.93 | 0.108 ±0.001 | 0.758 ±0.003 | **19.28** ±0.17 | 0.044 ±0.001 | 75.79 |
| | V-MoE | 2 | 1 | 0.428 ±0.003 | 11.89 ±0.13 | 0.030 ±0.001 | 0.934 ±0.013 | 22.41 ±0.20 | 0.038 ±0.001 | 2.517 ±0.063 | 50.34 ±1.12 | 0.167 ±0.003 | 0.811 ±0.005 | 20.25 ±0.13 | 0.065 ±0.001 | — |
| | E³ | 1 | 2 | 0.448 ±0.001 | 12.60 ±0.03 | 0.020 ±0.000 | 1.023 ±0.005 | 24.89 ±0.11 | 0.024 ±0.000 | 2.836 ±0.015 | 57.97 ±0.21 | 0.160 ±0.001 | 0.838 ±0.002 | 21.30 ±0.07 | 0.051 ±0.001 | **6.74** |
| L/16 | down-DE | 1 | 2 | 0.450 ±0.002 | 12.62 ±0.04 | **0.016** ±0.000 | **1.010** ±0.006 | 24.43 ±0.12 | **0.021** ±0.000 | **2.796** ±0.016 | **57.06** ±0.32 | **0.136** ±0.000 | **0.818** ±0.003 | **21.06** ±0.11 | **0.044** ±0.001 | 86.04 |
| | V-MoE | 2 | 1 | 0.464 ±0.001 | 12.88 ±0.04 | 0.025 ±0.000 | 1.058 ±0.004 | 25.27 ±0.08 | 0.034 ±0.000 | 2.945 ±0.016 | 57.85 ±0.28 | 0.178 ±0.001 | 0.848 ±0.002 | 21.33 ±0.03 | 0.057 ±0.001 | — |
| | E³ | 1 | 2 | **0.535** ±0.001 | 14.70 ±0.03 | **0.027** ±0.000 | 1.193 ±0.003 | 28.28 ±0.05 | 0.032 ±0.000 | 4.170 ±0.010 | 74.73 ±0.09 | 0.266 ±0.002 | 0.989 ±0.002 | 24.62 ±0.08 | **0.063** ±0.001 | **6.35** |
| L/32 | down-DE | 1 | 2 | **0.533** ±0.002 | **14.55** ±0.04 | 0.025 ±0.001 | **1.184** ±0.003 | **27.98** ±0.04 | 0.029 ±0.000 | **4.139** ±0.017 | 74.29 ±0.21 | **0.254** ±0.003 | **0.982** ±0.002 | 24.42 ±0.08 | **0.061** ±0.002 | 85.29 |
| | V-MoE | 2 | 1 | 0.563 ±0.001 | 15.05 ±0.03 | 0.039 ±0.000 | 1.261 ±0.003 | 29.12 ±0.07 | 0.052 ±0.000 | 4.394 ±0.014 | 75.39 ±0.17 | 0.301 ±0.002 | 1.046 ±0.002 | 25.22 ±0.06 | 0.083 ±0.001 | — |
| | E³ | 1 | 2 | 0.519 ±0.001 | 14.26 ±0.03 | **0.020** ±0.000 | 1.352 ±0.005 | 31.20 ±0.09 | **0.022** ±0.000 | 3.835 ±0.019 | 72.25 ±0.22 | 0.223 ±0.002 | 0.974 ±0.002 | 24.10 ±0.10 | 0.051 ±0.001 | **12.60** |
| B/16 | down-DE | 1 | 2 | 0.519 ±0.002 | **14.09** ±0.02 | 0.021 ±0.001 | **1.316** ±0.008 | **30.02** ±0.18 | 0.030 ±0.000 | **3.618** ±0.013 | **66.99** ±0.28 | **0.203** ±0.002 | **0.945** ±0.003 | **23.39** ±0.09 | 0.054 ±0.001 | 75.09 |
| | V-MoE | 2 | 1 | 0.533 ±0.001 | 14.60 ±0.04 | 0.022 ±0.000 | 1.372 ±0.005 | 31.27 ±0.11 | 0.034 ±0.000 | 3.875 ±0.016 | 70.79 ±0.30 | 0.235 ±0.002 | 0.959 ±0.003 | 24.09 ±0.11 | 0.056 ±0.001 | — |
| | E³ | 1 | 2 | 0.622 ±0.001 | 16.70 ±0.03 | **0.018** ±0.000 | 1.532 ±0.005 | 34.73 ±0.08 | **0.022** ±0.000 | 5.080 ±0.013 | 84.91 ±0.09 | 0.301 ±0.001 | 1.143 ±0.002 | 27.97 ±0.10 | **0.055** ±0.001 | 11.54 |
| B/32 | down-DE | 1 | 2 | **0.620** ±0.001 | **16.44** ±0.04 | 0.023 ±0.000 | **1.510** ±0.005 | **33.79** ±0.08 | 0.032 ±0.000 | **4.891** ±0.014 | 81.98 ±0.18 | **0.284** ±0.002 | **1.116** ±0.002 | 27.24 ±0.08 | 0.061 ±0.000 | 73.49 |
| | V-MoE | 2 | 1 | 0.638 ±0.001 | 16.76 ±0.05 | 0.033 ±0.001 | 1.562 ±0.004 | 34.40 ±0.06 | 0.050 ±0.001 | 5.032 ±0.014 | **81.71** ±0.11 | 0.317 ±0.002 | 1.150 ±0.002 | 27.50 ±0.08 | 0.076 ±0.001 | — |
| | E³ | 1 | 2 | 0.818 ±0.002 | 21.18 ±0.06 | **0.015** ±0.000 | 2.169 ±0.008 | 45.79 ±0.10 | **0.030** ±0.000 | 6.419 ±0.011 | 93.98 ±0.09 | 0.345 ±0.001 | 1.437 ±0.002 | 34.03 ±0.05 | **0.054** ±0.001 | **15.87** |
| S/32 | down-DE | 1 | 2 | **0.807** ±0.003 | **20.90** ±0.10 | 0.018 ±0.001 | **2.106** ±0.010 | **44.52** ±0.18 | 0.038 ±0.001 | **6.063** ±0.007 | 92.33 ±0.07 | **0.335** ±0.001 | **1.393** ±0.003 | **33.29** ±0.10 | 0.062 ±0.002 | 64.34 |
| | V-MoE | 2 | 1 | 0.829 ±0.002 | 21.09 ±0.08 | 0.035 ±0.001 | 2.162 ±0.007 | 44.95 ±0.12 | 0.066 ±0.001 | 6.227 ±0.015 | **92.09** ±0.12 | 0.373 ±0.001 | 1.437 ±0.002 | 33.42 ±0.08 | 0.086 ±0.001 | — |

Table 26: ImageNet comparison of V-MoE, downstream ensembles there-of, and E³ with 4 experts per input in each case.

| | | K | M | IMAGENET NLL ↓ | ERROR ↓ | ECE ↓ | IMAGENET-C (AVERAGE) NLL ↓ | ERROR ↓ | ECE ↓ | IMAGENET-A NLL ↓ | ERROR ↓ | ECE ↓ | IMAGENET-V2 NLL ↓ | ERROR ↓ | ECE ↓ | DOWNSTREAM ΔFLOPs (%) ↓ |
|---|---|---|---|---|---|---|---|---|---|---|---|---|---|---|---|---|
| | E³ | 2 | 2 | 0.451 ±0.001 | 12.61 ±0.04 | 0.023 ±0.000 | 1.028 ±0.003 | 24.90 ±0.07 | 0.028 ±0.000 | 2.886 ±0.016 | 58.26 ±0.24 | 0.171 ±0.002 | 0.849 ±0.002 | 21.40 ±0.11 | 0.055 ±0.001 | **5.92** |
| L/16 | down-DE | 1 | 4 | **0.440** ±0.002 | **12.39** ±0.06 | **0.015** ±0.000 | **0.983** ±0.006 | **23.95** ±0.12 | **0.020** ±0.000 | **2.712** ±0.015 | **56.36** ±0.36 | **0.120** ±0.002 | **0.803** ±0.002 | **20.83** ±0.10 | **0.039** ±0.001 | 226.53 |
| | V-MoE | 4 | 1 | 0.465 ±0.001 | 12.86 ±0.05 | 0.026 ±0.000 | 1.060 ±0.005 | 25.28 ±0.10 | 0.035 ±0.000 | 2.976 ±0.022 | 58.26 ±0.29 | 0.186 ±0.002 | 0.856 ±0.002 | 21.53 ±0.07 | 0.060 ±0.001 | — |
| | E³ | 2 | 2 | 0.529 ±0.001 | 14.63 ±0.02 | **0.019** ±0.000 | 1.188 ±0.004 | 28.23 ±0.07 | 0.025 ±0.000 | 4.095 ±0.015 | 74.38 ±0.14 | 0.253 ±0.002 | **0.962** ±0.001 | 24.41 ±0.04 | 0.055 ±0.001 | **5.54** |
| L/32 | down-DE | 1 | 4 | **0.518** ±0.002 | **14.29** ±0.03 | 0.022 ±0.000 | **1.154** ±0.004 | **27.47** ±0.05 | **0.023** ±0.000 | **4.033** ±0.016 | **73.79** ±0.13 | **0.237** ±0.002 | 0.959 ±0.002 | **24.03** ±0.04 | **0.054** ±0.001 | 222.95 |
| | V-MoE | 4 | 1 | 0.566 ±0.002 | 15.08 ±0.04 | 0.041 ±0.000 | 1.267 ±0.003 | 29.19 ±0.05 | 0.053 ±0.000 | 4.428 ±0.017 | 75.52 ±0.06 | 0.306 ±0.001 | 1.050 ±0.003 | 25.21 ±0.05 | 0.084 ±0.000 | — |
| | E³ | 2 | 2 | **0.510** ±0.001 | 14.13 ±0.02 | **0.016** ±0.000 | 1.328 ±0.006 | 30.81 ±0.11 | **0.020** ±0.000 | 3.743 ±0.011 | 71.12 ±0.16 | 0.211 ±0.002 | 0.945 ±0.002 | 23.75 ±0.03 | **0.044** ±0.001 | **10.09** |
| B/16 | down-DE | 1 | 4 | 0.511 ±0.002 | **13.95** ±0.01 | 0.019 ±0.001 | **1.293** ±0.008 | **29.67** ±0.18 | 0.026 ±0.000 | **3.544** ±0.015 | **66.49** ±0.27 | **0.190** ±0.002 | **0.930** ±0.003 | **23.17** ±0.06 | 0.050 ±0.001 | 180.41 |
| | V-MoE | 4 | 1 | 0.532 ±0.001 | 14.21 ±0.04 | 0.029 ±0.001 | 1.350 ±0.005 | 30.44 ±0.11 | 0.046 ±0.001 | 3.726 ±0.013 | 66.88 ±0.10 | 0.238 ±0.002 | 0.973 ±0.003 | 23.69 ±0.10 | 0.069 ±0.001 | — |
| | E³ | 2 | 2 | 0.612 ±0.001 | 16.49 ±0.02 | **0.013** ±0.000 | 1.491 ±0.003 | 33.85 ±0.05 | **0.019** ±0.000 | 4.872 ±0.007 | 82.80 ±0.08 | 0.275 ±0.001 | 1.099 ±0.003 | 27.36 ±0.10 | **0.045** ±0.001 | **9.12** |
| B/32 | down-DE | 1 | 4 | **0.607** ±0.000 | **16.17** ±0.02 | 0.021 ±0.001 | **1.483** ±0.008 | **33.36** ±0.13 | 0.027 ±0.000 | **4.787** ±0.011 | 82.11 ±0.07 | 0.276 ±0.000 | 1.099 ±0.003 | 26.98 ±0.01 | 0.055 ±0.002 | 174.26 |
| | V-MoE | 4 | 1 | 0.636 ±0.001 | 16.70 ±0.04 | 0.034 ±0.001 | 1.555 ±0.003 | 34.33 ±0.05 | 0.051 ±0.001 | 5.031 ±0.013 | **81.62** ±0.13 | 0.322 ±0.002 | 1.150 ±0.003 | 27.49 ±0.10 | 0.079 ±0.002 | — |
| | E³ | 2 | 2 | 0.805 ±0.002 | 20.97 ±0.05 | **0.013** ±0.000 | 2.112 ±0.007 | 45.05 ±0.09 | **0.028** ±0.000 | 6.283 ±0.013 | 93.68 ±0.09 | 0.342 ±0.001 | 1.408 ±0.003 | 33.71 ±0.10 | **0.051** ±0.001 | **11.70** |
| S/32 | down-DE | 1 | 4 | **0.795** ±0.003 | **20.66** ±0.13 | 0.015 ±0.001 | **2.076** ±0.012 | **44.16** ±0.21 | 0.031 ±0.000 | **5.990** ±0.011 | 92.25 ±0.03 | **0.324** ±0.001 | **1.372** ±0.002 | **32.83** ±0.13 | 0.054 ±0.002 | 142.29 |
| | V-MoE | 4 | 1 | 0.820 ±0.002 | 20.97 ±0.07 | 0.030 ±0.001 | 2.133 ±0.006 | 44.53 ±0.12 | 0.060 ±0.001 | 6.142 ±0.014 | **91.62** ±0.12 | 0.365 ±0.002 | 1.417 ±0.003 | 33.30 ±0.12 | 0.081 ±0.001 | — |

Table 27: ImageNet comparison of V-MoE and ViT.

| | | K | M | IMAGENET NLL ↓ | IMAGENET ERROR ↓ | ECE ↓ | IMAGENET-C NLL ↓ | IMAGENET-C ERROR ↓ | ECE ↓ | IMAGENET-A NLL ↓ | IMAGENET-A ERROR ↓ | ECE ↓ | IMAGENET-V2 NLL ↓ | IMAGENET-V2 ERROR ↓ | ECE ↓ |
|---|---|---|---|---|---|---|---|---|---|---|---|---|---|---|---|
| H/14 | V-MoE | 1 | 1 | **0.426** ±0.002 | **11.81** ±0.08 | 0.026 ±0.000 | **0.932** ±0.009 | **22.43** ±0.16 | 0.033 ±0.000 | 2.494 ±0.057 | 50.22 ±1.02 | **0.153** ±0.004 | **0.806** ±0.005 | **20.08** ±0.13 | **0.059** ±0.001 |
| | ViT | - | 1 | **0.426** ±0.001 | 11.99 ±0.05 | **0.023** ±0.000 | **0.929** ±0.004 | 22.45 ±0.06 | **0.031** ±0.000 | **2.332** ±0.018 | **46.99** ±0.21 | **0.149** ±0.002 | **0.788** ±0.004 | **20.08** ±0.03 | **0.058** ±0.001 |
| L/16 | V-MoE | 1 | 1 | **0.464** ±0.001 | **12.91** ±0.04 | 0.022 ±0.000 | **1.050** ±0.004 | **25.19** ±0.08 | **0.030** ±0.000 | **2.914** ±0.016 | **58.02** ±0.33 | **0.167** ±0.001 | **0.844** ±0.003 | **21.42** ±0.06 | **0.053** ±0.001 |
| | ViT | - | 1 | 0.473 ±0.004 | 13.37 ±0.08 | **0.020** ±0.000 | 1.114 ±0.009 | 26.68 ±0.17 | 0.032 ±0.001 | 3.094 ±0.042 | 60.92 ±0.56 | 0.203 ±0.004 | 0.864 ±0.007 | 22.04 ±0.17 | **0.055** ±0.001 |
| L/32 | V-MoE | 1 | 1 | **0.559** ±0.001 | **15.10** ±0.03 | 0.036 ±0.001 | **1.246** ±0.004 | **29.02** ±0.07 | 0.046 ±0.000 | 4.320 ±0.013 | **75.25** ±0.11 | **0.288** ±0.002 | 1.032 ±0.003 | **25.13** ±0.09 | 0.077 ±0.001 |
| | ViT | - | 1 | **0.556** ±0.002 | 15.38 ±0.05 | **0.025** ±0.000 | 1.255 ±0.009 | 29.57 ±0.16 | **0.038** ±0.000 | **4.286** ±0.021 | 75.32 ±0.30 | **0.287** ±0.002 | **1.002** ±0.004 | 25.32 ±0.09 | **0.067** ±0.001 |
| B/16 | V-MoE | 1 | 1 | **0.533** ±0.001 | **14.33** ±0.04 | 0.026 ±0.000 | **1.355** ±0.005 | **30.59** ±0.10 | **0.040** ±0.000 | **3.727** ±0.014 | **67.72** ±0.16 | **0.222** ±0.001 | **0.965** ±0.003 | **23.63** ±0.11 | **0.060** ±0.001 |
| | ViT | - | 1 | 0.565 ±0.003 | 15.70 ±0.06 | **0.021** ±0.001 | 1.515 ±0.006 | 34.52 ±0.10 | 0.042 ±0.001 | 4.219 ±0.032 | 75.85 ±0.29 | 0.280 ±0.003 | 1.020 ±0.006 | 25.77 ±0.12 | **0.061** ±0.001 |
| B/32 | V-MoE | 1 | 1 | **0.642** ±0.002 | **16.90** ±0.05 | 0.029 ±0.001 | **1.568** ±0.003 | **34.68** ±0.05 | 0.045 ±0.001 | **5.039** ±0.009 | **82.49** ±0.13 | **0.307** ±0.002 | **1.151** ±0.003 | **27.83** ±0.10 | 0.071 ±0.001 |
| | ViT | - | 1 | 0.688 ±0.003 | 18.65 ±0.08 | **0.022** ±0.000 | 1.689 ±0.005 | 38.02 ±0.09 | 0.045 ±0.000 | 5.358 ±0.014 | 87.00 ±0.12 | 0.342 ±0.001 | 1.209 ±0.005 | 29.89 ±0.10 | **0.067** ±0.001 |
| S/32 | V-MoE | 1 | 1 | **0.834** ±0.002 | **21.43** ±0.08 | 0.029 ±0.001 | **2.171** ±0.005 | **45.44** ±0.09 | **0.056** ±0.001 | **6.199** ±0.012 | **92.47** ±0.11 | **0.355** ±0.001 | **1.433** ±0.002 | **33.84** ±0.02 | 0.078 ±0.001 |
| | ViT | - | 1 | 0.907 ±0.003 | 23.69 ±0.05 | **0.016** ±0.000 | 2.309 ±0.010 | 48.75 ±0.13 | **0.055** ±0.001 | 6.639 ±0.014 | 94.66 ±0.07 | 0.375 ±0.001 | 1.529 ±0.003 | 36.42 ±0.08 | **0.066** ±0.001 |

Table 28: CIFAR10 comparison of V-MoE, downstream ensembles there-of, and $\textsc{e}^3$ with 2 experts per input in each case.

| | | K | M | NLL ↓ | Cifar10 Error ↓ | ECE ↓ | NLL ↓ | Cifar10-C Error ↓ | ECE ↓ | Downstream ΔFLOPs (%) ↓ |
|---|---|---|---|---|---|---|---|---|---|---|
| L/16 | $\textsc{e}^3$ | 1 | 2 | **0.022** ±0.000 | **0.55** ±0.02 | **0.003** ±0.000 | **0.198** ±0.012 | **5.65** ±0.20 | **0.027** ±0.002 | **6.74** |
| | down-DE | 1 | 2 | 0.026 ±0.001 | **0.59** ±0.04 | 0.004 ±0.000 | **0.227** ±0.011 | **6.18** ±0.21 | **0.032** ±0.003 | 86.04 |
| | V-MoE | 2 | 1 | 0.028 ±0.000 | **0.61** ±0.01 | 0.004 ±0.000 | **0.220** ±0.012 | **6.17** ±0.23 | **0.032** ±0.003 | — |
| L/32 | $\textsc{e}^3$ | 1 | 2 | **0.026** ±0.001 | **0.69** ±0.03 | **0.003** ±0.000 | **0.246** ±0.008 | **7.01** ±0.15 | **0.033** ±0.002 | **6.35** |
| | down-DE | 1 | 2 | 0.034 ±0.001 | 0.81 ±0.02 | 0.005 ±0.000 | **0.252** ±0.013 | **7.00** ±0.26 | **0.032** ±0.004 | 85.29 |
| | V-MoE | 2 | 1 | 0.042 ±0.001 | 0.90 ±0.03 | 0.006 ±0.000 | 0.325 ±0.011 | 8.05 ±0.17 | 0.048 ±0.002 | — |
| B/16 | $\textsc{e}^3$ | 1 | 2 | 0.036 ±0.001 | **0.84** ±0.02 | 0.005 ±0.000 | 0.295 ±0.008 | 8.28 ±0.15 | 0.038 ±0.002 | **12.60** |
| | down-DE | 1 | 2 | **0.030** ±0.001 | **0.81** ±0.04 | **0.003** ±0.000 | **0.247** ±0.005 | **7.60** ±0.16 | **0.028** ±0.001 | 75.09 |
| | V-MoE | 2 | 1 | 0.040 ±0.001 | 0.97 ±0.02 | 0.006 ±0.000 | 0.303 ±0.008 | 8.08 ±0.09 | 0.047 ±0.002 | — |
| B/32 | $\textsc{e}^3$ | 1 | 2 | 0.041 ±0.001 | **1.11** ±0.03 | 0.004 ±0.000 | **0.340** ±0.010 | **10.01** ±0.18 | **0.040** ±0.003 | **11.54** |
| | down-DE | 1 | 2 | **0.036** ±0.000 | **1.07** ±0.03 | **0.003** ±0.000 | **0.345** ±0.006 | 10.87 ±0.13 | **0.042** ±0.002 | 73.49 |
| | V-MoE | 2 | 1 | 0.042 ±0.000 | **1.11** ±0.03 | 0.006 ±0.000 | 0.405 ±0.015 | 11.30 ±0.24 | 0.063 ±0.004 | — |
| S/32 | $\textsc{e}^3$ | 1 | 2 | 0.060 ±0.001 | 1.92 ±0.03 | **0.003** ±0.000 | **0.476** ±0.007 | **15.30** ±0.17 | **0.049** ±0.002 | **15.86** |
| | down-DE | 1 | 2 | **0.056** ±0.001 | **1.75** ±0.03 | 0.005 ±0.000 | **0.491** ±0.004 | **15.42** ±0.15 | 0.064 ±0.001 | 64.34 |
| | V-MoE | 2 | 1 | **0.058** ±0.001 | **1.80** ±0.03 | 0.007 ±0.000 | 0.514 ±0.006 | **15.48** ±0.21 | 0.076 ±0.002 | — |

Table 29: CIFAR10 comparison of V-MoE, downstream ensembles there-of, and $\textsc{e}^3$ with 4 experts per input in each case.

| | | K | M | NLL ↓ | Cifar10 Error ↓ | ECE ↓ | NLL ↓ | Cifar10-C Error ↓ | ECE ↓ | Downstream ΔFLOPs (%) ↓ |
|---|---|---|---|---|---|---|---|---|---|---|
| L/16 | $\textsc{e}^3$ | 2 | 2 | **0.026** ±0.001 | **0.58** ±0.02 | **0.003** ±0.000 | **0.247** ±0.016 | **6.56** ±0.26 | **0.036** ±0.003 | **5.92** |
| | down-DE | 1 | 4 | **0.025** ±0.001 | **0.57** ±0.03 | **0.004** ±0.000 | **0.219** ±0.010 | **6.12** ±0.19 | **0.030** ±0.003 | 226.53 |
| | V-MoE | 4 | 1 | 0.034 ±0.001 | 0.75 ±0.02 | 0.005 ±0.000 | 0.278 ±0.010 | 7.71 ±0.19 | 0.043 ±0.002 | — |
| L/32 | $\textsc{e}^3$ | 2 | 2 | **0.033** ±0.001 | **0.80** ±0.02 | **0.004** ±0.000 | **0.242** ±0.006 | **6.65** ±0.12 | **0.031** ±0.002 | **5.54** |
| | down-DE | 1 | 4 | **0.030** ±0.001 | **0.78** ±0.02 | **0.004** ±0.000 | **0.232** ±0.012 | **6.74** ±0.27 | **0.028** ±0.003 | 222.94 |
| | V-MoE | 4 | 1 | 0.035 ±0.001 | **0.81** ±0.02 | 0.005 ±0.000 | 0.295 ±0.009 | 7.63 ±0.14 | 0.044 ±0.002 | — |
| B/16 | $\textsc{e}^3$ | 2 | 2 | 0.034 ±0.000 | **0.81** ±0.02 | 0.004 ±0.000 | 0.283 ±0.006 | 8.06 ±0.12 | 0.037 ±0.002 | **10.09** |
| | down-DE | 1 | 4 | **0.029** ±0.000 | **0.80** ±0.03 | **0.003** ±0.000 | **0.242** ±0.006 | **7.54** ±0.16 | **0.026** ±0.001 | 180.41 |
| | V-MoE | 4 | 1 | **0.032** ±0.001 | **0.79** ±0.02 | 0.005 ±0.000 | 0.266 ±0.004 | **7.55** ±0.10 | 0.040 ±0.001 | — |
| B/32 | $\textsc{e}^3$ | 2 | 2 | **0.034** ±0.001 | **1.00** ±0.02 | **0.003** ±0.000 | **0.306** ±0.009 | **9.44** ±0.23 | **0.037** ±0.002 | **9.12** |
| | down-DE | 1 | 4 | **0.035** ±0.001 | 1.06 ±0.04 | **0.003** ±0.000 | 0.338 ±0.005 | 10.77 ±0.12 | **0.040** ±0.002 | 174.26 |
| | V-MoE | 4 | 1 | 0.042 ±0.001 | 1.12 ±0.02 | 0.006 ±0.000 | 0.410 ±0.017 | 11.29 ±0.26 | 0.066 ±0.004 | — |
| S/32 | $\textsc{e}^3$ | 2 | 2 | 0.058 ±0.001 | **1.82** ±0.01 | **0.004** ±0.000 | **0.472** ±0.007 | **15.01** ±0.17 | **0.051** ±0.002 | **11.69** |
| | down-DE | 1 | 4 | **0.055** ±0.001 | **1.76** ±0.04 | **0.004** ±0.000 | **0.486** ±0.003 | **15.36** ±0.13 | 0.061 ±0.001 | 142.28 |
| | V-MoE | 4 | 1 | 0.059 ±0.001 | **1.80** ±0.05 | 0.007 ±0.000 | 0.536 ±0.012 | **15.64** ±0.27 | 0.084 ±0.004 | — |

Table 30: CIFAR10 comparison of V-MoE and ViT.

| | | K | M | NLL ↓ | Cifar10 Error ↓ | ECE ↓ | NLL ↓ | Cifar10-C Error ↓ | ECE ↓ |
|---|---|---|---|---|---|---|---|---|---|
| L/16 | V-MoE | 1 | 1 | **0.029** ±0.001 | **0.59** ±0.02 | **0.004** ±0.000 | **0.236** ±0.011 | **6.25** ±0.20 | **0.034** ±0.002 |
| | ViT | - | 1 | 0.034 ±0.001 | **0.69** ±0.03 | **0.005** ±0.000 | 0.325 ±0.020 | 7.53 ±0.40 | 0.050 ±0.004 |
| L/32 | V-MoE | 1 | 1 | 0.040 ±0.001 | **0.86** ±0.02 | 0.006 ±0.000 | **0.290** ±0.013 | **7.35** ±0.22 | **0.043** ±0.003 |
| | ViT | - | 1 | **0.030** ±0.001 | **0.78** ±0.02 | **0.005** ±0.000 | **0.281** ±0.010 | **7.28** ±0.13 | **0.043** ±0.002 |
| B/16 | V-MoE | 1 | 1 | **0.030** ±0.001 | **0.85** ±0.02 | **0.003** ±0.000 | **0.249** ±0.004 | **7.53** ±0.12 | **0.030** ±0.001 |
| | ViT | - | 1 | **0.031** ±0.001 | **0.87** ±0.03 | **0.004** ±0.000 | 0.293 ±0.009 | **7.99** ±0.15 | 0.043 ±0.002 |
| B/32 | V-MoE | 1 | 1 | **0.038** ±0.000 | **1.14** ±0.03 | **0.004** ±0.000 | 0.359 ±0.008 | 11.09 ±0.17 | **0.046** ±0.002 |
| | ViT | - | 1 | 0.050 ±0.002 | 1.38 ±0.04 | 0.007 ±0.000 | **0.330** ±0.010 | **8.97** ±0.19 | **0.051** ±0.002 |
| S/32 | V-MoE | 1 | 1 | **0.059** ±0.001 | **1.84** ±0.02 | **0.006** ±0.000 | **0.497** ±0.007 | **15.53** ±0.20 | **0.066** ±0.002 |
| | ViT | - | 1 | 0.065 ±0.001 | 2.08 ±0.02 | **0.006** ±0.000 | 0.518 ±0.008 | **15.04** ±0.24 | **0.072** ±0.002 |

Table 31: CIFAR10 OOD comparison of V-MoE, downstream ensembles there-of, and $\textsc{e}^3$ with 2 experts per input in each case.

| | | K | M | CIFAR10 vs. CIFAR100 AUC (PR) ↑ | AUC (ROC) ↑ | FPR@95 ↓ | CIFAR10 vs. DTD AUC (PR) ↑ | AUC (ROC) ↑ | FPR@95 ↓ | CIFAR10 vs. Places365 AUC (PR) ↑ | AUC (ROC) ↑ | FPR@95 ↓ | CIFAR10 vs. SVHN AUC (PR) ↑ | AUC (ROC) ↑ | FPR@95 ↓ |
|---|---|---|---|---|---|---|---|---|---|---|---|---|---|---|---|
| L/16 | $\textsc{e}^3$ | 1 | 2 | **0.9905** $\pm$ 0.0003 | **0.9902** $\pm$ 0.0003 | **0.0313** $\pm$ 0.0010 | 0.9999 $\pm$ 0.0000 | 0.9993 $\pm$ 0.0000 | 0.0005 $\pm$ 0.0001 | 0.9103 $\pm$ 0.0068 | 0.9936 $\pm$ 0.0003 | 0.0294 $\pm$ 0.0013 | **0.9963** $\pm$ 0.0002 | **0.9978** $\pm$ 0.0001 | **0.0007** $\pm$ 0.0002 |
| | down-DE | 1 | 2 | **0.9896** $\pm$ 0.0005 | **0.9896** $\pm$ 0.0005 | 0.0357 $\pm$ 0.0029 | 0.9999 $\pm$ 0.0000 | 0.9996 $\pm$ 0.0000 | **0.0001** $\pm$ 0.0001 | **0.9421** $\pm$ 0.0063 | **0.9959** $\pm$ 0.0003 | **0.0162** $\pm$ 0.0010 | **0.9968** $\pm$ 0.0002 | **0.9982** $\pm$ 0.0002 | **0.0007** $\pm$ 0.0001 |
| | V-MoE | 2 | 1 | **0.9895** $\pm$ 0.0004 | **0.9890** $\pm$ 0.0003 | 0.0389 $\pm$ 0.0018 | 0.9998 $\pm$ 0.0000 | 0.9988 $\pm$ 0.0001 | 0.0011 $\pm$ 0.0001 | 0.7891 $\pm$ 0.0120 | 0.9887 $\pm$ 0.0004 | 0.0435 $\pm$ 0.0005 | **0.9966** $\pm$ 0.0001 | **0.9980** $\pm$ 0.0001 | **0.0005** $\pm$ 0.0001 |
| L/32 | $\textsc{e}^3$ | 1 | 2 | **0.9875** $\pm$ 0.0003 | **0.9873** $\pm$ 0.0002 | **0.0427** $\pm$ 0.0011 | **0.9998** $\pm$ 0.0000 | **0.9990** $\pm$ 0.0000 | **0.0011** $\pm$ 0.0002 | **0.9196** $\pm$ 0.0028 | **0.9934** $\pm$ 0.0002 | **0.0321** $\pm$ 0.0013 | **0.9950** $\pm$ 0.0002 | **0.9970** $\pm$ 0.0002 | **0.0012** $\pm$ 0.0001 |
| | down-DE | 1 | 2 | 0.9852 $\pm$ 0.0005 | 0.9842 $\pm$ 0.0005 | 0.0571 $\pm$ 0.0016 | 0.9994 $\pm$ 0.0001 | 0.9966 $\pm$ 0.0004 | 0.0032 $\pm$ 0.0004 | 0.7388 $\pm$ 0.0165 | 0.9846 $\pm$ 0.0007 | 0.0561 $\pm$ 0.0019 | **0.9939** $\pm$ 0.0004 | 0.9960 $\pm$ 0.0003 | **0.0012** $\pm$ 0.0002 |
| | V-MoE | 2 | 1 | 0.9839 $\pm$ 0.0007 | 0.9839 $\pm$ 0.0005 | 0.0589 $\pm$ 0.0020 | 0.9998 $\pm$ 0.0000 | 0.9987 $\pm$ 0.0001 | 0.0013 $\pm$ 0.0001 | 0.8967 $\pm$ 0.0056 | **0.9927** $\pm$ 0.0003 | 0.0346 $\pm$ 0.0012 | **0.9944** $\pm$ 0.0003 | **0.9965** $\pm$ 0.0002 | 0.0017 $\pm$ 0.0003 |
| B/16 | $\textsc{e}^3$ | 1 | 2 | 0.9823 $\pm$ 0.0003 | 0.9800 $\pm$ 0.0003 | 0.0846 $\pm$ 0.0026 | 0.9994 $\pm$ 0.0000 | 0.9967 $\pm$ 0.0002 | **0.0015** $\pm$ 0.0002 | 0.7661 $\pm$ 0.0121 | 0.9834 $\pm$ 0.0004 | 0.0602 $\pm$ 0.0010 | 0.9914 $\pm$ 0.0002 | 0.9942 $\pm$ 0.0002 | **0.0029** $\pm$ 0.0005 |
| | down-DE | 1 | 2 | **0.9860** $\pm$ 0.0005 | **0.9859** $\pm$ 0.0004 | 0.0540 $\pm$ 0.0026 | 0.9999 $\pm$ 0.0000 | 0.9993 $\pm$ 0.0001 | 0.0009 $\pm$ 0.0002 | 0.8807 $\pm$ 0.0148 | 0.9922 $\pm$ 0.0005 | 0.0388 $\pm$ 0.0013 | **0.9951** $\pm$ 0.0003 | **0.9974** $\pm$ 0.0002 | **0.0027** $\pm$ 0.0005 |
| | V-MoE | 2 | 1 | 0.9822 $\pm$ 0.0005 | 0.9809 $\pm$ 0.0005 | 0.0800 $\pm$ 0.0030 | 0.9994 $\pm$ 0.0000 | 0.9964 $\pm$ 0.0002 | 0.0023 $\pm$ 0.0003 | 0.7399 $\pm$ 0.0141 | 0.9841 $\pm$ 0.0006 | 0.0568 $\pm$ 0.0014 | 0.9925 $\pm$ 0.0004 | 0.9953 $\pm$ 0.0002 | 0.0037 $\pm$ 0.0009 |
| B/32 | $\textsc{e}^3$ | 1 | 2 | 0.9773 $\pm$ 0.0003 | 0.9738 $\pm$ 0.0004 | 0.1254 $\pm$ 0.0020 | 0.9992 $\pm$ 0.0000 | 0.9957 $\pm$ 0.0003 | 0.0038 $\pm$ 0.0006 | 0.7438 $\pm$ 0.0085 | 0.9794 $\pm$ 0.0006 | 0.0802 $\pm$ 0.0023 | 0.9881 $\pm$ 0.0001 | 0.9918 $\pm$ 0.0002 | 0.0071 $\pm$ 0.0004 |
| | down-DE | 1 | 2 | **0.9820** $\pm$ 0.0003 | **0.9813** $\pm$ 0.0002 | 0.0810 $\pm$ 0.0033 | 0.9998 $\pm$ 0.0000 | 0.9989 $\pm$ 0.0001 | 0.0014 $\pm$ 0.0004 | 0.8567 $\pm$ 0.0099 | 0.9900 $\pm$ 0.0005 | 0.0514 $\pm$ 0.0017 | 0.9924 $\pm$ 0.0002 | 0.9957 $\pm$ 0.0003 | 0.0054 $\pm$ 0.0004 |
| | V-MoE | 2 | 1 | 0.9798 $\pm$ 0.0004 | 0.9790 $\pm$ 0.0004 | 0.0875 $\pm$ 0.0020 | 0.9997 $\pm$ 0.0000 | 0.9986 $\pm$ 0.0001 | 0.0017 $\pm$ 0.0003 | 0.8725 $\pm$ 0.0074 | 0.9904 $\pm$ 0.0006 | 0.0482 $\pm$ 0.0019 | 0.9919 $\pm$ 0.0002 | 0.9952 $\pm$ 0.0003 | 0.0047 $\pm$ 0.0004 |
| S/32 | $\textsc{e}^3$ | 1 | 2 | 0.9617 $\pm$ 0.0003 | 0.9567 $\pm$ 0.0003 | 0.2629 $\pm$ 0.0040 | 0.9992 $\pm$ 0.0000 | 0.9956 $\pm$ 0.0002 | 0.0118 $\pm$ 0.0011 | 0.7431 $\pm$ 0.0070 | 0.9750 $\pm$ 0.0006 | 0.1270 $\pm$ 0.0036 | 0.9824 $\pm$ 0.0004 | 0.9899 $\pm$ 0.0002 | 0.0388 $\pm$ 0.0028 |
| | down-DE | 1 | 2 | **0.9696** $\pm$ 0.0003 | **0.9669** $\pm$ 0.0005 | 0.1856 $\pm$ 0.0046 | 0.9996 $\pm$ 0.0000 | 0.9981 $\pm$ 0.0001 | 0.0037 $\pm$ 0.0002 | 0.8135 $\pm$ 0.0146 | 0.9845 $\pm$ 0.0007 | 0.0815 $\pm$ 0.0036 | 0.9879 $\pm$ 0.0004 | 0.9933 $\pm$ 0.0002 | 0.0164 $\pm$ 0.0019 |
| | V-MoE | 2 | 1 | **0.9691** $\pm$ 0.0008 | **0.9667** $\pm$ 0.0009 | 0.1870 $\pm$ 0.0065 | 0.9996 $\pm$ 0.0000 | 0.9979 $\pm$ 0.0001 | 0.0043 $\pm$ 0.0006 | 0.8010 $\pm$ 0.0087 | 0.9839 $\pm$ 0.0007 | 0.0827 $\pm$ 0.0032 | 0.9873 $\pm$ 0.0005 | 0.9927 $\pm$ 0.0004 | 0.0192 $\pm$ 0.0016 |

Table 32: CIFAR10 OOD comparison of ViT and V-MoE

| | | K | M | CIFAR10 vs. CIFAR100 AUC (PR) ↑ | AUC (ROC) ↑ | FPR@95 ↓ | CIFAR10 vs. DTD AUC (PR) ↑ | AUC (ROC) ↑ | FPR@95 ↓ | CIFAR10 vs. Places365 AUC (PR) ↑ | AUC (ROC) ↑ | FPR@95 ↓ | CIFAR10 vs. SVHN AUC (PR) ↑ | AUC (ROC) ↑ | FPR@95 ↓ |
|---|---|---|---|---|---|---|---|---|---|---|---|---|---|---|---|
| L/16 | V-MoE | 1 | 1 | **0.9891** $\pm$ 0.0004 | **0.9895** $\pm$ 0.0004 | **0.0379** $\pm$ 0.0022 | 0.9999 $\pm$ 0.0000 | **0.9997** $\pm$ 0.0000 | **0.0001** $\pm$ 0.0001 | **0.9400** $\pm$ 0.0057 | **0.9960** $\pm$ 0.0002 | **0.0162** $\pm$ 0.0013 | **0.9972** $\pm$ 0.0001 | **0.9984** $\pm$ 0.0001 | **0.0007** $\pm$ 0.0001 |
| | ViT | - | 1 | 0.9839 $\pm$ 0.0008 | 0.9845 $\pm$ 0.0007 | 0.0541 $\pm$ 0.0026 | 0.9996 $\pm$ 0.0000 | 0.9978 $\pm$ 0.0002 | 0.0018 $\pm$ 0.0004 | 0.7334 $\pm$ 0.0227 | 0.9857 $\pm$ 0.0010 | 0.0492 $\pm$ 0.0024 | 0.9947 $\pm$ 0.0003 | 0.9967 $\pm$ 0.0002 | 0.0022 $\pm$ 0.0003 |
| L/32 | V-MoE | 1 | 1 | **0.9854** $\pm$ 0.0003 | **0.9850** $\pm$ 0.0003 | 0.0573 $\pm$ 0.0018 | 0.9996 $\pm$ 0.0000 | 0.9976 $\pm$ 0.0002 | 0.0030 $\pm$ 0.0003 | 0.7489 $\pm$ 0.0049 | 0.9862 $\pm$ 0.0003 | 0.0520 $\pm$ 0.0011 | 0.9946 $\pm$ 0.0003 | **0.9967** $\pm$ 0.0002 | **0.0015** $\pm$ 0.0003 |
| | ViT | - | 1 | **0.9842** $\pm$ 0.0005 | **0.9847** $\pm$ 0.0004 | 0.0551 $\pm$ 0.0018 | 0.9998 $\pm$ 0.0000 | 0.9987 $\pm$ 0.0001 | 0.0016 $\pm$ 0.0003 | 0.9164 $\pm$ 0.0046 | 0.9938 $\pm$ 0.0003 | 0.0279 $\pm$ 0.0015 | 0.9942 $\pm$ 0.0002 | 0.9966 $\pm$ 0.0001 | 0.0028 $\pm$ 0.0003 |
| B/16 | V-MoE | 1 | 1 | **0.9856** $\pm$ 0.0004 | **0.9855** $\pm$ 0.0004 | 0.0554 $\pm$ 0.0025 | 0.9998 $\pm$ 0.0000 | 0.9992 $\pm$ 0.0000 | **0.0015** $\pm$ 0.0005 | **0.8598** $\pm$ 0.0191 | 0.9912 $\pm$ 0.0008 | 0.0413 $\pm$ 0.0028 | 0.9949 $\pm$ 0.0003 | 0.9972 $\pm$ 0.0002 | **0.0027** $\pm$ 0.0002 |
| | ViT | - | 1 | 0.9801 $\pm$ 0.0005 | 0.9798 $\pm$ 0.0004 | 0.0857 $\pm$ 0.0023 | 0.9996 $\pm$ 0.0000 | 0.9979 $\pm$ 0.0001 | 0.0046 $\pm$ 0.0007 | **0.8536** $\pm$ 0.0057 | 0.9895 $\pm$ 0.0003 | 0.0511 $\pm$ 0.0012 | 0.9926 $\pm$ 0.0002 | 0.9961 $\pm$ 0.0002 | 0.0048 $\pm$ 0.0003 |
| B/32 | V-MoE | 1 | 1 | **0.9814** $\pm$ 0.0003 | **0.9806** $\pm$ 0.0003 | 0.0853 $\pm$ 0.0027 | 0.9998 $\pm$ 0.0000 | 0.9989 $\pm$ 0.0001 | 0.0017 $\pm$ 0.0005 | 0.8590 $\pm$ 0.0097 | 0.9902 $\pm$ 0.0005 | 0.0506 $\pm$ 0.0021 | 0.9923 $\pm$ 0.0003 | 0.9958 $\pm$ 0.0003 | 0.0061 $\pm$ 0.0005 |
| | ViT | - | 1 | 0.9752 $\pm$ 0.0005 | 0.9716 $\pm$ 0.0006 | 0.1485 $\pm$ 0.0040 | 0.9985 $\pm$ 0.0001 | 0.9915 $\pm$ 0.0004 | 0.0186 $\pm$ 0.0010 | 0.6507 $\pm$ 0.0056 | 0.9734 $\pm$ 0.0006 | 0.1021 $\pm$ 0.0037 | 0.9872 $\pm$ 0.0004 | 0.9920 $\pm$ 0.0003 | 0.0148 $\pm$ 0.0012 |
| S/32 | V-MoE | 1 | 1 | **0.9685** $\pm$ 0.0008 | **0.9658** $\pm$ 0.0010 | **0.1922** $\pm$ 0.0056 | 0.9996 $\pm$ 0.0000 | **0.9977** $\pm$ 0.0002 | **0.0045** $\pm$ 0.0007 | **0.8092** $\pm$ 0.0105 | **0.9841** $\pm$ 0.0008 | **0.0821** $\pm$ 0.0032 | **0.9874** $\pm$ 0.0006 | **0.9929** $\pm$ 0.0004 | **0.0185** $\pm$ 0.0019 |
| | ViT | - | 1 | 0.9629 $\pm$ 0.0004 | 0.9588 $\pm$ 0.0004 | 0.2422 $\pm$ 0.0027 | 0.9993 $\pm$ 0.0000 | 0.9963 $\pm$ 0.0002 | 0.0134 $\pm$ 0.0013 | 0.7177 $\pm$ 0.0048 | 0.9775 $\pm$ 0.0003 | 0.1110 $\pm$ 0.0015 | 0.9807 $\pm$ 0.0003 | 0.9889 $\pm$ 0.0002 | 0.0426 $\pm$ 0.0013 |

Table 33: CIFAR100 OOD comparison of V-MoE, downstream ensembles there-of, and E³ with 2 experts per input in each case.

| | | K | M | CIFAR100 vs. CIFAR10 AUC (PR) ↑ | AUC (ROC) ↑ | FPR@95 ↓ | CIFAR100 vs. DTD AUC (PR) ↑ | AUC (ROC) ↑ | FPR@95 ↓ | CIFAR100 vs. Places365 AUC (PR) ↑ | AUC (ROC) ↑ | FPR@95 ↓ | CIFAR100 vs. SVHN AUC (PR) ↑ | AUC (ROC) ↑ | FPR@95 ↓ |
|---|---|---|---|---|---|---|---|---|---|---|---|---|---|---|---|
| L/16 | E³ | 1 | 2 | **0.9507** ±0.0012 | **0.9490** ±0.0011 | **0.2889** ±0.0058 | 0.9969 ±0.0001 | 0.9847 ±0.0002 | 0.0862 ±0.0010 | 0.7281 ±0.0023 | 0.9542 ±0.0007 | 0.2969 ±0.0036 | **0.9011** ±0.0057 | **0.9482** ±0.0024 | **0.3071** ±0.0093 |
| | down-DE | 1 | 2 | 0.9455 ±0.0013 | **0.9481** ±0.0019 | 0.2692 ±0.0093 | **0.9975** ±0.0000 | **0.9881** ±0.0002 | **0.0665** ±0.0035 | 0.7619 ±0.0065 | 0.9664 ±0.0012 | 0.2158 ±0.0071 | **0.9071** ±0.0027 | **0.9507** ±0.0016 | **0.2932** ±0.0144 |
| | V-MoE | 2 | 1 | 0.9391 ±0.0014 | **0.9443** ±0.0015 | **0.2901** ±0.0076 | **0.9977** ±0.0000 | **0.9887** ±0.0002 | 0.0674 ±0.0017 | 0.7773 ±0.0054 | 0.9680 ±0.0009 | 0.2150 ±0.0041 | 0.9002 ±0.0045 | 0.9461 ±0.0024 | 0.3223 ±0.0124 |
| L/32 | E³ | 1 | 2 | **0.9423** ±0.0003 | **0.9379** ±0.0006 | 0.3591 ±0.0058 | **0.9958** ±0.0001 | 0.9792 ±0.0004 | 0.1165 ±0.0028 | 0.6653 ±0.0037 | 0.9394 ±0.0011 | 0.3670 ±0.0051 | 0.8848 ±0.0061 | 0.9398 ±0.0026 | 0.3428 ±0.0080 |
| | down-DE | 1 | 2 | **0.9380** ±0.0029 | **0.9387** ±0.0015 | 0.3362 ±0.0042 | **0.9958** ±0.0001 | 0.9796 ±0.0004 | 0.1176 ±0.0015 | 0.6710 ±0.0088 | 0.9436 ±0.0019 | 0.3465 ±0.0093 | 0.8877 ±0.0081 | 0.9460 ±0.0027 | 0.3033 ±0.0077 |
| | V-MoE | 2 | 1 | 0.9275 ±0.0041 | 0.9330 ±0.0020 | 0.3604 ±0.0052 | **0.9958** ±0.0001 | 0.9790 ±0.0005 | 0.1306 ±0.0029 | 0.6773 ±0.0075 | 0.9445 ±0.0017 | 0.3489 ±0.0089 | 0.8767 ±0.0075 | 0.9393 ±0.0020 | 0.3404 ±0.0069 |
| B/16 | E³ | 1 | 2 | **0.9200** ±0.0008 | 0.9121 ±0.0010 | 0.4656 ±0.0071 | 0.9938 ±0.0002 | 0.9710 ±0.0011 | 0.1542 ±0.0063 | 0.5475 ±0.0076 | 0.9102 ±0.0014 | 0.4653 ±0.0052 | 0.8730 ±0.0056 | 0.9319 ±0.0025 | 0.3699 ±0.0090 |
| | down-DE | 1 | 2 | **0.9242** ±0.0016 | **0.9275** ±0.0013 | 0.3506 ±0.0070 | **0.9952** ±0.0003 | **0.9777** ±0.0011 | **0.1144** ±0.0052 | 0.6023 ±0.0101 | 0.9343 ±0.0015 | 0.3577 ±0.0049 | 0.8726 ±0.0049 | 0.9340 ±0.0027 | 0.3526 ±0.0124 |
| | V-MoE | 2 | 1 | 0.9159 ±0.0019 | 0.9211 ±0.0015 | **0.3729** ±0.0053 | 0.9950 ±0.0002 | 0.9768 ±0.0008 | 0.1193 ±0.0038 | 0.6029 ±0.0066 | 0.9331 ±0.0008 | 0.3682 ±0.0029 | 0.8704 ±0.0038 | 0.9283 ±0.0020 | 0.3915 ±0.0076 |
| B/32 | E³ | 1 | 2 | 0.9075 ±0.0012 | 0.8945 ±0.0015 | 0.5358 ±0.0063 | 0.9906 ±0.0004 | 0.9554 ±0.0019 | 0.2481 ±0.0101 | 0.4355 ±0.0071 | 0.8811 ±0.0014 | 0.5465 ±0.0043 | 0.8583 ±0.0040 | 0.9221 ±0.0024 | 0.4051 ±0.0100 |
| | down-DE | 1 | 2 | **0.9188** ±0.0021 | **0.9158** ±0.0014 | **0.4248** ±0.0077 | **0.9942** ±0.0003 | **0.9719** ±0.0014 | **0.1517** ±0.0090 | 0.5525 ±0.0096 | 0.9176 ±0.0022 | **0.4276** ±0.0085 | **0.8756** ±0.0083 | 0.9292 ±0.0033 | **0.3959** ±0.0083 |
| | V-MoE | 2 | 1 | **0.9192** ±0.0014 | 0.9151 ±0.0012 | **0.4444** ±0.0060 | 0.9913 ±0.0002 | 0.9580 ±0.0008 | 0.2481 ±0.0064 | 0.4449 ±0.0100 | 0.8969 ±0.0009 | 0.5230 ±0.0027 | 0.8641 ±0.0073 | 0.9252 ±0.0028 | 0.4249 ±0.0080 |
| S/32 | E³ | 1 | 2 | **0.8677** ±0.0016 | 0.8528 ±0.0017 | 0.6086 ±0.0049 | **0.9878** ±0.0007 | 0.9430 ±0.0026 | 0.2677 ±0.0084 | 0.3628 ±0.0072 | 0.8519 ±0.0026 | 0.5716 ±0.0074 | 0.8279 ±0.0037 | **0.9004** ±0.0024 | 0.4543 ±0.0098 |
| | down-DE | 1 | 2 | **0.8697** ±0.0019 | **0.8651** ±0.0024 | **0.5203** ±0.0095 | **0.9892** ±0.0005 | **0.9532** ±0.0023 | **0.1909** ±0.0086 | 0.4185 ±0.0063 | 0.8814 ±0.0020 | 0.4657 ±0.0078 | 0.8162 ±0.0045 | 0.8912 ±0.0026 | 0.4835 ±0.0066 |
| | V-MoE | 2 | 1 | **0.8687** ±0.0011 | **0.8663** ±0.0012 | **0.5206** ±0.0061 | **0.9894** ±0.0004 | 0.9523 ±0.0018 | 0.2076 ±0.0075 | 0.4129 ±0.0082 | 0.8752 ±0.0026 | 0.5039 ±0.0072 | 0.8133 ±0.0029 | 0.8882 ±0.0020 | 0.5031 ±0.0058 |

Table 34: CIFAR100 OOD comparison of V-MoE and ViT

| | | K | M | CIFAR100 vs. CIFAR10 AUC (PR) ↑ | AUC (ROC) ↑ | FPR@95 ↓ | CIFAR100 vs. DTD AUC (PR) ↑ | AUC (ROC) ↑ | FPR@95 ↓ | CIFAR100 vs. Places365 AUC (PR) ↑ | AUC (ROC) ↑ | FPR@95 ↓ | CIFAR100 vs. SVHN AUC (PR) ↑ | AUC (ROC) ↑ | FPR@95 ↓ |
|---|---|---|---|---|---|---|---|---|---|---|---|---|---|---|---|
| L/16 | V-MoE | 1 | 1 | **0.9454** ±0.0013 | **0.9481** ±0.0015 | **0.2682** ±0.0073 | **0.9976** ±0.0000 | **0.9882** ±0.0002 | **0.0658** ±0.0024 | **0.7631** ±0.0043 | **0.9667** ±0.0007 | **0.2141** ±0.0040 | **0.9115** ±0.0035 | **0.9533** ±0.0020 | **0.2794** ±0.0123 |
| | ViT | - | 1 | **0.9411** ±0.0019 | **0.9449** ±0.0012 | **0.2734** ±0.0062 | 0.9970 ±0.0001 | 0.9854 ±0.0006 | 0.0853 ±0.0039 | 0.7552 ±0.0118 | 0.9608 ±0.0017 | 0.2566 ±0.0066 | 0.8697 ±0.0033 | 0.9326 ±0.0016 | 0.3690 ±0.0058 |
| L/32 | V-MoE | 1 | 1 | **0.9358** ±0.0028 | **0.9368** ±0.0018 | 0.3431 ±0.0052 | 0.9961 ±0.0001 | 0.9806 ±0.0004 | 0.1148 ±0.0022 | 0.6757 ±0.0080 | 0.9461 ±0.0016 | 0.3308 ±0.0081 | **0.8863** ±0.0093 | **0.9459** ±0.0025 | **0.3063** ±0.0123 |
| | ViT | - | 1 | 0.9285 ±0.0020 | 0.9323 ±0.0016 | **0.3201** ±0.0068 | **0.9967** ±0.0001 | **0.9838** ±0.0006 | **0.0911** ±0.0036 | **0.7541** ±0.0066 | **0.9634** ±0.0014 | **0.2292** ±0.0068 | 0.8495 ±0.0068 | 0.9234 ±0.0035 | 0.3949 ±0.0137 |
| B/16 | V-MoE | 1 | 1 | **0.9206** ±0.0015 | **0.9241** ±0.0014 | **0.3572** ±0.0061 | **0.9951** ±0.0002 | **0.9776** ±0.0007 | **0.1094** ±0.0034 | **0.5924** ±0.0076 | **0.9334** ±0.0012 | **0.3532** ±0.0035 | **0.8720** ±0.0048 | **0.9309** ±0.0023 | **0.3705** ±0.0122 |
| | ViT | - | 1 | **0.9177** ±0.0013 | 0.9171 ±0.0014 | 0.3925 ±0.0069 | 0.9906 ±0.0003 | 0.9571 ±0.0012 | 0.2199 ±0.0059 | 0.4913 ±0.0088 | 0.8980 ±0.0024 | 0.4927 ±0.0079 | 0.8525 ±0.0045 | 0.9204 ±0.0022 | 0.4226 ±0.0065 |
| B/32 | V-MoE | 1 | 1 | **0.9166** ±0.0017 | **0.9145** ±0.0012 | 0.4211 ±0.0048 | **0.9940** ±0.0002 | **0.9714** ±0.0009 | **0.1562** ±0.0068 | **0.5433** ±0.0089 | **0.9178** ±0.0016 | 0.4258 ±0.0049 | **0.8730** ±0.0071 | **0.9276** ±0.0033 | **0.4026** ±0.0100 |
| | ViT | - | 1 | 0.9038 ±0.0022 | 0.9044 ±0.0018 | **0.4180** ±0.0061 | 0.9927 ±0.0002 | 0.9663 ±0.0008 | 0.1631 ±0.0055 | 0.5306 ±0.0046 | 0.9112 ±0.0015 | **0.4176** ±0.0062 | 0.8379 ±0.0029 | 0.9116 ±0.0014 | 0.4328 ±0.0062 |
| S/32 | V-MoE | 1 | 1 | **0.8678** ±0.0012 | **0.8631** ±0.0015 | **0.5281** ±0.0050 | **0.9893** ±0.0007 | **0.9524** ±0.0031 | **0.1978** ±0.0129 | **0.4024** ±0.0091 | **0.8778** ±0.0029 | **0.4763** ±0.0085 | **0.8224** ±0.0058 | **0.8945** ±0.0034 | **0.4729** ±0.0098 |
| | ViT | - | 1 | **0.8644** ±0.0018 | 0.8541 ±0.0020 | 0.5716 ±0.0061 | 0.9836 ±0.0007 | 0.9287 ±0.0026 | 0.2976 ±0.0082 | 0.3149 ±0.0031 | 0.8349 ±0.0024 | 0.5982 ±0.0061 | 0.8088 ±0.0051 | **0.8894** ±0.0029 | **0.4888** ±0.0084 |

Table 35: Few-shot comparison of E³, V-MoE and ensembles thereof.

| | | K | M | MEAN ACROSS DATASETS | | | | WEIGHTED MEAN ACROSS DATASETS | | | | DOWNSTREAM |
| | | | | 1-Shot Error ↓ | 5-Shot Error ↓ | 10-Shot Error ↓ | 25-Shot Error ↓ | 1-Shot Error ↓ | 5-Shot Error ↓ | 10-Shot Error ↓ | 25-Shot Error ↓ | ΔFLOPs (%) ↓ |
|---|---|---|---|---|---|---|---|---|---|---|---|---|
| H/14 | E³ | 1 | 2 | **31.47** ±0.72 | **17.87** ±0.32 | **14.29** ±0.22 | **10.64** ±0.25 | **78.51** ±0.24 | **80.65** ±0.08 | **81.28** ±0.05 | **81.51** ±0.06 | **15.63** |
| | down-DE | 1 | 2 | 30.84 ±0.42 | 17.77 ±0.51 | 14.43 ±0.03 | 11.06 ±0.40 | 78.36 ±0.20 | 80.63 ±0.14 | 81.31 ±0.02 | 81.60 ±0.09 | 75.79 |
| | V-MoE | 2 | 1 | 32.47 ±0.55 | 18.77 ±0.41 | 15.19 ±0.27 | 12.09 ±0.38 | 78.93 ±0.20 | 80.89 ±0.11 | 81.48 ±0.07 | 81.83 ±0.09 | — |
| L/16 | E³ | 1 | 2 | 34.08 ±0.21 | 19.57 ±0.16 | 15.21 ±0.14 | 11.75 ±0.11 | 79.50 ±0.07 | 81.09 ±0.04 | 81.48 ±0.03 | 81.76 ±0.02 | **6.74** |
| | down-DE | 1 | 2 | 32.98 ±0.42 | 19.65 ±0.16 | 15.44 ±0.16 | 11.92 ±0.14 | 79.09 ±0.13 | 81.10 ±0.04 | 81.54 ±0.04 | 81.79 ±0.03 | 86.04 |
| | V-MoE | 2 | 1 | 33.98 ±0.28 | 20.42 ±0.12 | 16.20 ±0.08 | 12.73 ±0.13 | 79.50 ±0.10 | 81.30 ±0.03 | 81.72 ±0.02 | 81.97 ±0.03 | — |
| L/32 | E³ | 1 | 2 | 36.71 ±0.20 | 22.14 ±0.16 | 17.79 ±0.05 | 13.97 ±0.09 | 80.51 ±0.07 | 81.77 ±0.04 | 82.11 ±0.01 | 82.26 ±0.02 | **6.35** |
| | down-DE | 1 | 2 | 36.15 ±0.28 | 22.18 ±0.15 | 17.81 ±0.14 | 14.00 ±0.12 | 80.45 ±0.13 | 81.79 ±0.04 | 82.13 ±0.03 | 82.27 ±0.03 | 85.29 |
| | V-MoE | 2 | 1 | 36.56 ±0.34 | 23.00 ±0.12 | 18.86 ±0.07 | 15.02 ±0.16 | 80.56 ±0.13 | 82.00 ±0.03 | 82.37 ±0.02 | 82.50 ±0.03 | — |
| B/16 | E³ | 1 | 2 | 38.60 ±0.38 | 21.93 ±0.17 | 17.43 ±0.13 | 13.62 ±0.08 | 81.14 ±0.11 | 81.74 ±0.04 | 82.05 ±0.03 | 82.19 ±0.02 | **12.60** |
| | down-DE | 1 | 2 | 38.96 ±0.36 | 23.36 ±0.23 | 18.99 ±0.12 | 14.84 ±0.11 | 81.39 ±0.10 | 82.09 ±0.06 | 82.40 ±0.03 | 82.46 ±0.02 | 75.09 |
| | V-MoE | 2 | 1 | **37.76** ±0.15 | 23.16 ±0.09 | 18.79 ±0.14 | 14.94 ±0.16 | 80.97 ±0.05 | 82.04 ±0.02 | 82.36 ±0.03 | 82.48 ±0.04 | — |
| B/32 | E³ | 1 | 2 | 43.39 ±0.33 | 26.51 ±0.11 | 21.18 ±0.17 | 16.82 ±0.09 | 82.86 ±0.11 | 82.91 ±0.03 | 82.93 ±0.04 | 82.90 ±0.02 | **11.54** |
| | down-DE | 1 | 2 | **41.09** ±0.31 | 26.48 ±0.24 | 21.61 ±0.22 | 17.20 ±0.13 | 82.24 ±0.13 | 82.89 ±0.06 | 83.03 ±0.05 | 82.99 ±0.03 | 73.49 |
| | V-MoE | 2 | 1 | 42.50 ±0.28 | 27.44 ±0.19 | 22.73 ±0.19 | 18.60 ±0.19 | 82.66 ±0.08 | 83.16 ±0.05 | 83.29 ±0.05 | 83.29 ±0.04 | — |
| S/32 | E³ | 1 | 2 | 52.91 ±0.27 | 34.28 ±0.19 | 27.54 ±0.18 | **21.85** ±0.11 | 85.89 ±0.08 | 84.81 ±0.04 | 84.39 ±0.04 | **84.01** ±0.02 | **15.87** |
| | down-DE | 1 | 2 | **48.27** ±0.21 | **32.19** ±0.22 | **26.55** ±0.10 | 21.95 ±0.20 | 84.62 ±0.06 | 84.35 ±0.05 | 84.19 ±0.03 | 84.04 ±0.05 | 64.34 |
| | V-MoE | 2 | 1 | 49.37 ±0.19 | 33.51 ±0.17 | 28.00 ±0.14 | 23.25 ±0.15 | 85.04 ±0.06 | 84.69 ±0.04 | 84.54 ±0.03 | 84.33 ±0.03 | — |

Table 36: Few-shot comparison of V-MoE and ViT.

| | | K | M | MEAN ACROSS DATASETS | | | | WEIGHTED MEAN ACROSS DATASETS | | | |
| | | | | 1-Shot Error ↓ | 5-Shot Error ↓ | 10-Shot Error ↓ | 25-Shot Error ↓ | 1-Shot Error ↓ | 5-Shot Error ↓ | 10-Shot Error ↓ | 25-Shot Error ↓ |
|---|---|---|---|---|---|---|---|---|---|---|---|
| H/14 | V-MoE | 1 | 1 | **33.04** ±0.32 | **19.23** ±0.35 | **15.63** ±0.25 | **12.06** ±0.36 | **79.05** ±0.12 | **81.00** ±0.09 | **81.59** ±0.06 | **81.82** ±0.08 |
| | ViT | - | 1 | 35.78 ±0.41 | 20.61 ±0.15 | 16.78 ±0.12 | 13.62 ±0.24 | 79.97 ±0.16 | 81.37 ±0.04 | 81.87 ±0.03 | 82.16 ±0.05 |
| L/16 | V-MoE | 1 | 1 | **34.15** ±0.27 | **20.32** ±0.12 | **16.14** ±0.12 | **12.71** ±0.15 | **79.54** ±0.08 | **81.27** ±0.03 | **81.71** ±0.03 | **81.97** ±0.03 |
| | ViT | - | 1 | 36.52 ±0.20 | 21.79 ±0.12 | 17.51 ±0.08 | 14.07 ±0.14 | 80.38 ±0.05 | 81.67 ±0.03 | 82.03 ±0.02 | 82.27 ±0.03 |
| L/32 | V-MoE | 1 | 1 | **36.83** ±0.27 | **23.05** ±0.08 | **18.78** ±0.11 | **14.95** ±0.07 | **80.67** ±0.10 | **82.01** ±0.02 | **82.35** ±0.02 | **82.48** ±0.02 |
| | ViT | - | 1 | 38.22 ±0.31 | 23.97 ±0.14 | 19.56 ±0.13 | 15.75 ±0.14 | 81.10 ±0.10 | 82.25 ±0.04 | 82.53 ±0.03 | 82.65 ±0.03 |
| B/16 | V-MoE | 1 | 1 | **39.22** ±0.27 | **24.42** ±0.13 | **20.01** ±0.11 | 15.94 ±0.18 | **81.47** ±0.08 | **82.36** ±0.04 | **82.64** ±0.03 | **82.70** ±0.04 |
| | ViT | - | 1 | 41.29 ±0.14 | 25.03 ±0.10 | 20.08 ±0.10 | **15.87** ±0.17 | 82.16 ±0.04 | 82.51 ±0.03 | 82.65 ±0.02 | 82.68 ±0.04 |
| B/32 | V-MoE | 1 | 1 | **42.37** ±0.31 | **27.60** ±0.20 | **22.89** ±0.19 | **18.46** ±0.10 | **82.64** ±0.11 | **83.19** ±0.05 | **83.33** ±0.05 | **83.26** ±0.02 |
| | ViT | - | 1 | 44.60 ±0.22 | 28.20 ±0.17 | 22.97 ±0.12 | 18.86 ±0.15 | 83.35 ±0.08 | 83.35 ±0.04 | 83.35 ±0.03 | 83.35 ±0.03 |
| S/32 | V-MoE | 1 | 1 | **49.60** ±0.28 | **33.34** ±0.13 | **27.88** ±0.11 | **23.30** ±0.18 | **85.09** ±0.08 | **84.64** ±0.03 | **84.50** ±0.03 | **84.33** ±0.04 |
| | ViT | - | 1 | 54.16 ±0.16 | 36.25 ±0.18 | 29.88 ±0.17 | 24.42 ±0.13 | 86.32 ±0.05 | 85.26 ±0.04 | 84.90 ±0.04 | 84.54 ±0.03 |

# N    Additional Algorithm Overview Diagrams

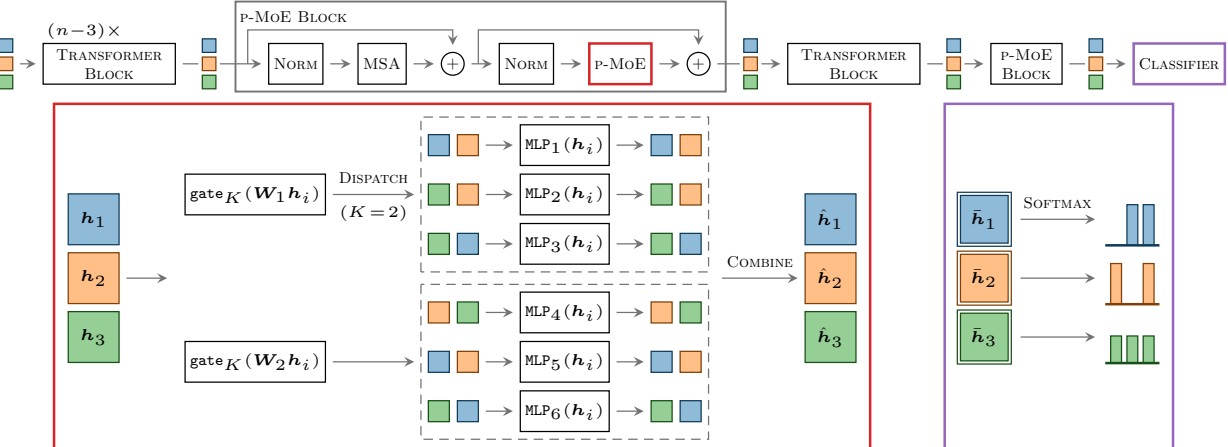

Figure 24: End-to-end overview of the *Partitioning* method, from Section 4.2.1, with $E=6$ experts, partitioned into $M=2$ groups, with sparsity of $K=2$, and a "last-2" configuration. **Top**: *Partitioning* contains a sequence of transformer blocks, followed by alternating transformer and p(artitioned)-MoE blocks. As in ViT, images are split into patches whose embeddings are processed by each block. Here, we show 1 embedding for each of three images (■, ■, ■). **Bottom left**: In a MoE block, we replace the transformer block's MLP with parallel partitioned expert MLPs. The effect of the routing weights is not depicted. **Bottom right**: The classifier makes predictions from the final representations (▣). Notice that without a tiling mechanism, there is only a single prediction per input.

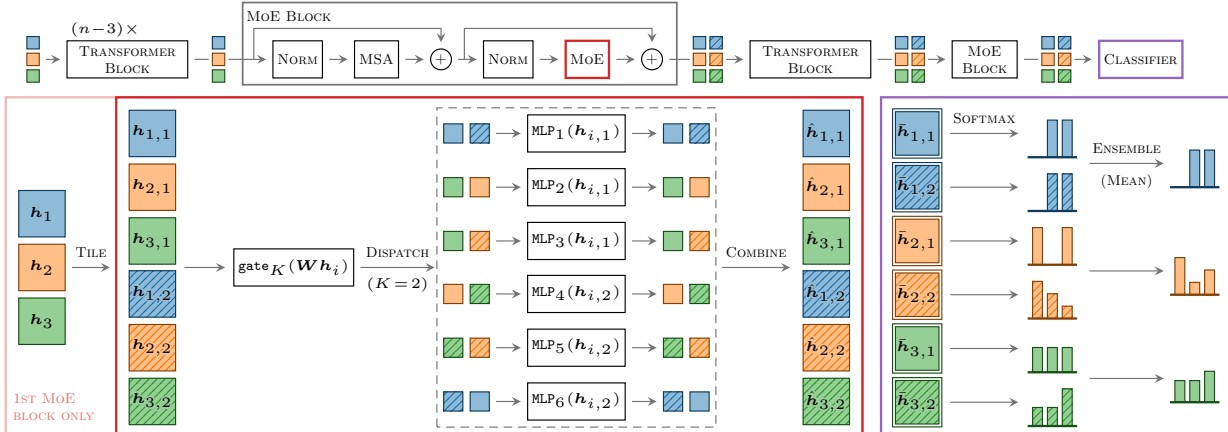

Figure 25: End-to-end overview of the *Tiling* method, from Section 4.2.2, with $E=6$ experts, a sparsity of $K=2$, and a "last-2" configuration. **Top**: *Tiling* contains a sequence of transformer blocks, followed by alternating transformer and MoE blocks. As in ViT, images are split into patches whose embeddings are processed by each block. Here, we show 1 embedding for each of three images (■, ■, ■). **Bottom left**: In a MoE block, we replace the transformer block's MLP with parallel partitioned expert MLPs. The effect of the routing weights is not depicted. Embeddings are tiled (▨) in the first p-MoE block only. **Bottom right**: The classifier averages predictions from the final tiled representations (▣). Notice that without partitioning, some patches and their corresponding tiled versions can be routed to the same experts, resulting in a reduced diversity in predictions.

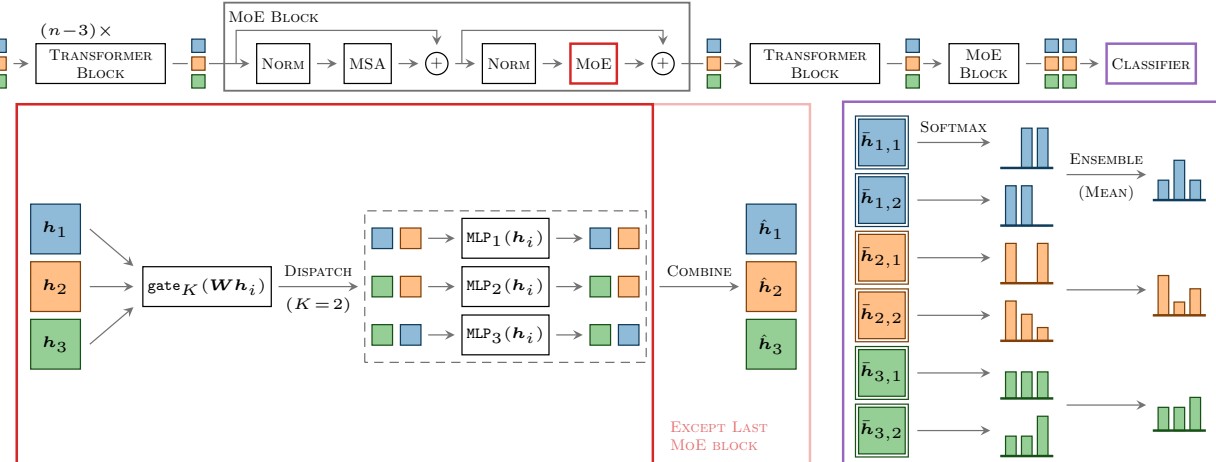

Figure 26: End-to-end overview of the simple *Multi-pred* MoE, from Section 4.2.4, with $E = 3$ experts, sparsity of $K = 2$, and a "last-2" configuration. **Top**: The multi-pred MoE contains a sequence of transformer blocks, followed by alternating transformer and MoE blocks. As in ViT, images are split into patches whose embeddings are processed by each block. Here, we show 1 embedding for each of three images (■, ■, ■). **Bottom left**: In all but the last MoE block, we recombine the predictions as usual. **Bottom right**: The classifier averages predictions from the final representations (▣).

