# OpenReview forum: "Sparse MoEs meet Efficient Ensembles"
_TMLR — Accepted by TMLR_

### Review · Reviewer_n8qH · 2022-06-08

**Summary Of Contributions:**

This paper attempts to combine the best of both ensembles and MoE (mixture of experts). This investigation is motivated by the observation that increasing the number of experts in MoE provides orthogonal improvements to training networks independently and averaging their predictions (i.e., ensembling). This motivates a proposed modification of tiling (i.e., making copies) of the input then running MoE but with disjoint groups of experts. Layers without experts remain shared.

**Requested Changes:**

- Adding a FLOPs comparison in Tables 2-5 and considering accuracy instead of NLL in Figures 3/4 (or, motivating why NLL is a good metric).
- Investigation or further discussion as to why M>2 seems to hurt performance despite increasing FLOPs, motivating why M=2 is a good choice.

**Strengths And Weaknesses:**

Strengths:
- The empirical investigation on ensembling and MoE is valuable to the community.
- While the proposed method introduces additional hyperparameters, it can be fine-tuned from existing MoE checkpoints.
- Consistent emperical improvements are observed in terms of NLL and ECE.
- The ablation studies in 4.2.* validate various design choices.
- MoE is presented very well for non-MoE-experts such as myself.

Weaknesses:
- Various tables and figures are missing metrics that could improve clarity of presentation and the ability to better contrast the method with other approaches. I would highly recommend adding FLOPs to Tables 2-5. Moreover, I would recommend a version of Figures 3/4 which looks at accuracy instead of NLL. It is not clear why NLL is a good metric.
- Many of the experiments in the main text are with M=2, which I would appreciate more discussion about. It seems from Appendix Table 12 that increasing M hurts performance (despite increasing FLOPs) which I don't understand and perhaps warrants further investigation.

---

> ### Author Response · Authors · 2022-06-30
> **Response to reviewer n8qH**
>
> Thank you for the thoughtful review. Below we respond to your points individually.
>
> > Consistent emperical improvements are observed in terms of NLL and ECE.
>
> We note that E$^3$ also provides consistent gains in terms of few-shot error.
>
> > Various tables and figures are missing metrics that could improve clarity of presentation and the ability to better contrast the method with other approaches. I would highly recommend adding FLOPs to Tables 2-5. Moreover, I would recommend a version of Figures 3/4 which looks at accuracy instead of NLL. It is not clear why NLL is a good metric.
>
> We have added FLOPs to Table 5, and we now provide FLOPs for all of the ablations of Tables 2-4 in Table 22 (Appendix M.1).
>
> We have chosen not to directly add the FLOPs to Tables 2-4 for the following reason. We have two implementations available to deal with the communication between the experts. One is more efficient than the other, but is more complex to implement. We had hence chosen to implement our ablations with the simpler approach. All the other results in the paper make use of the more efficient implementation. Thus, in order to avoid having conflicting FLOPs numbers for E$^3$ (simplified vs. efficient implementation) in the main text—where there is limited space to discuss these details—we have opted for a separate table of FLOPs in the appendix (where all of the ablations are fairly compared to E$^3$ and V-MoE which have been re-benchmarked with the simpler implementation, e.g., E$^3$-B/32 $K=M=2$ has 138.4 GLOPs compared to 131.1 GFLOPs with the more efficient implementation). Please let us know if presenting these FLOPs numbers in the appendix is satisfactory, given the above explanation.
>
> Regarding the use of NLL. We report accuracy results in Appendices L and M, e.g., Figure 17. We have used the NLL in the main text  because it measures both accuracy and uncertainty calibration. NLL is a common choice in the probabilistic and reliable deep learning communities ([Gal and Ghahramani; 2016](http://proceedings.mlr.press/v48/gal16.html), [Lakshminarayanan, et al.; 2016](https://arxiv.org/abs/1612.01474), [NeurIPS 2020 tutorial](https://slideslive.com/38935801/practical-uncertainty-estimation-outofdistribution-robustness-in-deep-learning?ref=recommended-presentation-38940952)). It is a proper scoring rule and as such, it is a principled metric to measure the performance of a probabilistic model ([Gneiting and Raftery; 2007](https://sites.stat.washington.edu/raftery/Research/PDF/Gneiting2007jasa.pdf)).

---

> > ### Author Response · Authors · 2022-06-30
> > **Response to reviewer n8qH – part 2**
> >
> > > Many of the experiments in the main text are with M=2, which I would appreciate more discussion about. It seems from Appendix Table 12 that increasing M hurts performance (despite increasing FLOPs) which I don't understand and perhaps warrants further investigation.
> >
> > In Appendix L.8 (Figure 23), we explore the impact of increasing $M$ on a wider range of benchmarks than Table 12. We see that increasing $M$ can sometimes improve performance, though there are no clear trends for when performance is improved versus degraded. In a standard deep ensemble, increasing $M$ consistently improves performance. In the case of E$^3$, the setting is more complex for the following reasons:
> >   * Firstly, the fact that our ensemble members are fine-tuned from an upstream V-MoE with a predefined total number of experts ($E=32$) meaning that increasing $M$ decreases the number of experts available to each ensemble member (with $E/M$ experts per member).
> >   * Secondly, there is a mismatch between the upstream and downstream models. The downstream models have all been fine tuned from V-MoEs with $K=2$ (as recommended by Riquelme et al. (2021)). We can see from Appendix E.4 (Figures 33-35) of Riquelme et al. (2021), that a mismatch between the upstream and downstream values of $K$ negatively impacts predictive performance of V-MoE. Thus, we would expect the same to be the case for the mismatch between the upstream V-MoE model’s K and the downstream E$^3$ models’ $K$*$M$.
> >
> > As suggested by the reviewer, we have further investigated this phenomenon.  We have run an additional experiment, comparing the performance of models with $K=2$ or $K=4$ upstream, and downstream ($K=1$, $M=2$) and ($K=1$, $M=4$). We have included these results in a new Appendix L.8.1 (Table 19). We see that ($K=1$, $M=2$) does not seem to benefit from increasing from $K_\text{up}=2$ to $K_\text{up}=4$, despite the fact that the upstream $K=4$ model is better than the upstream $K=2$ model (e.g., NLL upstream is 8.18 for $K=4$ and 8.27 for $K=2$). On the other hand, ($K=1$, $M=4$) does benefit from having $K=4$ upstream. This confirms that the mismatch between upstream and downstream models is one of the factors explaining the results of E$^3$ for growing $M$.
> >
> > More generally, we conjecture that there exists a non-trivial scaling law between ($K$, $M$, $E$) that affects the performance of fine-tuned models, by increasing the compatibility of the upstream and downstream models. We leave to future work an investigation of the relationship between the choice of $E$ and $M$.
> >
> > In this work, we focus instead on the most practical setup, by reusing the most popular V-MoE checkpoints ($K=2$) that are expensive to produce upstream. Moreover, in terms of increase in FLOPs, and as observed in Table 12, it is more efficient to consider ($K=2$, $M=2$) versus ($K=1$, $M=4$).

---

> > > ### Comment · Reviewer_n8qH · 2022-07-08
> > > **Thank you for addressing these comments.**
> > >
> > > I feel that for the most part my concerns are addressed. I believe it's okay to add FLOPs to Tables 2-4 even if it's the FLOPs attainable by the more efficient implementation.

---

> > > > ### Author Response · Authors · 2022-07-11
> > > > **Thanks!**
> > > >
> > > > Thanks for engaging with our response. We are glad that your concerns have mostly been addressed. Please let us know if there is any other point you would want to further discuss to finish addressing your concerns.

---

### Review · Reviewer_JkZs · 2022-06-15

**Summary Of Contributions:**

This paper studies the interplay of ensembles of neural networks and sparse mixture of experts (sparse MoEs). It first performs a comprehensive evaluation of sparse MoEs in uncertainty-related benchmarks and then proposes E^3 -- a scalable and simple ensemble of sparse MoEs. The extensive experiments are extensive, confirming some of the made claims.

**Requested Changes:**

I expect the authors to address my concerns listed above and will re-evaluate this submission after revision.

**Strengths And Weaknesses:**

## Strong aspects
- the writing/presentation is good and it is a pleasure to read this paper
- the empirical experiments are thorough


## Weak aspects
- The motivation for combining MoE and ensemble is not clear to me. Sec 3 is not surprising: like the naive deep ensemble, simple ensembling of sparse MoEs results in strong predictive performance while losing computational efficiency. What are the new insights? In my opinion, Sec 3 is not a qualified motivation for the proposal E^3. You should make clear the necessity of combining these two methods, for example, given an MoE with 16 experts and an ensemble of two MoEs with 8 experts respectively (assuming the two models have similar FLOPs), you show that the ensemble significantly surpasses the separate MoE. Then we are motivated to introduce ensemble into MoE, and will figure out how to realize this efficiently in the next section.

- The technical novelty is limited. The concept of using partitioning and tiling to implement an efficient ensemble has been explored by multiple related works like the (cited) batch ensemble, MIMO, etc. I would better regard it as a trick to implement ensemble instead of a considerable technical contribution.

- The improvements of E^3 upon V-MoE in aspects of Error and NLL in Table 4 and 5 are not significant. The OOD detection performance in  Figure 6 and the NLL under distribution shift in Figure 7 seem to prove that V-MoE is at least on par with E^3, with only marginally added GFLOPs.

- Though I know ViT architectures are powerful and widely used nowadays. However, I think conditioning your method on such a specific architecture is restrictive. It seems not pretty hard to apply this method to other architectures but adding relevant experiment results is definitely beneficial.

- An issue caused by tilling is the rapidly-increasing memory cost. It is better to clarify this point and report some details.

- A minor issue: why can replacing topK(softmax(Wh)) with topK (softmax(W h+ σε)) mitigate the non-differentiability of topK? I see a citation but I recommend the authors revise this part carefully.

---

> ### Author Response · Authors · 2022-06-30
> **Response to reviewer JkZs**
>
> Thank you for the detailed review! We respond to each of your points below.
>
> > The motivation for combining MoE and ensemble is not clear to me. Sec 3 is not surprising: like the naive deep ensemble, simple ensembling of sparse MoEs results in strong predictive performance while losing computational efficiency. What are the new insights?
>
> The main insight here is that traditional ensembling and (sparse) adaptive computation are complementary. It is not obvious that sparse MoE models (which average submodels at an activation level) would still benefit from an additional ensembling of models at a prediction level. It is, however, unsurprising that this is computationally inefficient.
>
> > In my opinion, Sec 3 is not a qualified motivation for the proposal E^3. You should make clear the necessity of combining these two methods, for example, given an MoE with 16 experts and an ensemble of two MoEs with 8 experts respectively (assuming the two models have similar FLOPs), you show that the ensemble significantly surpasses the separate MoE. Then we are motivated to introduce ensemble into MoE, and will figure out how to realize this efficiently in the next section.
>
> While we do believe our motivation is qualified, we also agree that the proposed investigation and motivation are also interesting and would complement our own. Thus, we have run the following new experiment: we have trained ensembles of V-MoE-B/32 models (with $K=1$ and $K=2$ selected experts), and ($M=1$, $E=32$), ($M=2$, $E=16$), ($M=4, E=8$). In short, we find that ($M=1$, $E=32$) < ($M=2$, $E=16$) < ($M=4$, $E=8$), for both $K=1$ and $K=2$, which further motivates E$^3$. We provide these new results in Appendix L.2, Table 17. We note that this result is especially remarkable since the individual upstream (and later, downstream) models are such that $E=8$ is worse than $E=16$ which in turn performs worse than $E=32$; ensembling thus manages to counterbalance the poorer *individual* model performance.
>
> We also note that the choice of $E$ has almost no impact on FLOPs (as rightly assumed by the reviewer), and thus the ensembles for $E=8/16$ are almost $M$ times more expensive than the $E=32$ model. We thank the reviewer for making our motivation stronger.
>
> > The technical novelty is limited. The concept of using partitioning and tiling to implement an efficient ensemble has been explored by multiple related works like the (cited) batch ensemble, MIMO, etc. I would better regard it as a trick to implement ensemble instead of a considerable technical contribution.
>
> Note that MIMO uses a somewhat different tiling mechanism and does not introduce any partitioning, and that batch ensemble does not make use of partitioning, but rather employs its own mechanism (‘fast weights’) to induce ensemble members. The choice of where to introduce ensemble-specific parameters–e.g., MoE MLPs vs. all layers–is equally important as the actual ensemble mechanism–e.g., partitioning of experts vs. fast weights. For more discussion of E$^3$ vs. batch ensemble, please see Appendix F. We would be grateful if the reviewer could provide pointers to existing works that apply ideas like partitioning and/or tiling to Sparse MoEs.
>
> > The improvements of E^3 upon V-MoE in aspects of Error and NLL in Table 4 and 5 are not significant. The OOD detection performance in Figure 6 and the NLL under distribution shift in Figure 7 seem to prove that V-MoE is at least on par with E^3, with only marginally added GFLOPs.
>
> Tables 4 and 5 provide results for B/32 family models and were meant more as ablation studies than demonstrations of E$^3$’s improvement over V-MoE. We have shown (c.f., Appendix D) that E$^3$ provides larger gains for larger ViT families, and our results in Section 5 confirm this (e.g., we see that E$^3$ often outperforms or is competitive with ensembles of 2 or even 4 V-MoEs).
>
> > The OOD detection performance in Figure 6 and the NLL under distribution shift in Figure 7 seem to prove that V-MoE is at least on par with E^3, with only marginally added GFLOPs.
>
> For both OOD detection and NLL under distribution shift we see that, as described above, E$^3$ performance relative to V-MoE improves with ViT family size. We regret that our phrasing has implied for the reviewer that E$^3$ outperforms V-MoE in OOD detection. We would appreciate any specific feedback to make our claim better match our evidence. We agree that our description of the NLL under distribution shift could have been clearer, and we have updated the text to better reflect the impact of ViT model size. Please let us know if the new text meets with your impression of the results.

---

> > ### Author Response · Authors · 2022-06-30
> > **Response to reviewer JkZs – part 2**
> >
> > > Though I know ViT architectures are powerful and widely used nowadays. However, I think conditioning your method on such a specific architecture is restrictive. It seems not pretty hard to apply this method to other architectures but adding relevant experiment results is definitely beneficial.
> >
> > Both in NLP (Lepikhin et al., 2021; Fedus et al., 2021) and vision (Riquelme et al., 2021; Yang et al., 2021), sparse MoEs have been predominantly built on top of Transformer-based models. The scope or applicability of this work is not limited since Transformers are the dominant architecture in many modalities. When we started this research, V-MoE (Riquelme et al., 2021) was the only available sparse MoEs architecture for vision, hence our design choice. To the best of our knowledge, we are not aware of any successful applications of sparse MoEs to ResNets (or other CNNs). We would welcome any pointers to such work for inclusion in our related work discussion. Only later was the Mixer-MLP architecture used with conditional computation (Lou et al., 2021), whose combination with E$^3$ is—given our already “thorough” experiments in the words of the reviewer—beyond the scope of our paper.
> >
> > > An issue caused by tilling is the rapidly-increasing memory cost. It is better to clarify this point and report some details.
> >
> > Thank you for the suggestion. We now mention this detail in the main text and provide an additional Appendix C.3 which provides more discussion. In short, the memory complexity of a standard V-MoE can be decomposed into two terms  O(memory_params + memory_activations), for the forward and backward passes respectively. For E$^3$, the complexity becomes O(memory_params + memory_activations * ((layer_before_tile + layers_after_tile * $M$) / total_layers)). Importantly, neither memory_params nor memory_activivations depend on $M$.
> >
> > Thanks to the “last-$n$” setting employed in the paper, we have layers_after_tile much smaller than layer_before_tile, and thus the increase in memory due to tiling remains mild.
> >
> > > A minor issue: why can replacing topK(softmax(Wh)) with topK (softmax(W h+ σε)) mitigate the non-differentiability of topK? I see a citation but I recommend the authors revise this part carefully.
> >
> > Briefly, the noise in the router allows for a smooth estimator of the load on each expert to be calculated and back-propagated through in an auxiliary loss term. We have updated the citation to specifically point to Appendix A in Shazeer et al., to aid the curious reader. Moreover, we added a reference to [Berthet et al., (2021)](https://arxiv.org/pdf/2002.08676.pdf)—and some references therein—to indicate that making non-differentiable operators smooth with noise injection is an active area of research.

---

> > > ### Comment · Reviewer_JkZs · 2022-07-09
> > > **Further comments**
> > >
> > > I thank the authors for their effort in addressing my concerns. I believe most of them have been addressed. However, I still feel negative about the technical novelty of the "tilling" concept in Bayesian deep learning/ensemble learning. Further, "E^3 provides larger gains for larger ViT families, and our results in Section 5 confirm this" is correct, but, as I have pointed out, it is also correct that E^3 cannot outperform V-MoE in OOD detection/NLL under distribution shift. So, the merits of E^3 over V-MoE in other aspects should be revealed.

---

> > > > ### Author Response · Authors · 2022-07-11
> > > > **Response to further comments**
> > > >
> > > > Thank you for the in-depth discussion.
> > > >
> > > > > the merits of E$^3$ over V-MoE in other aspects should be revealed.
> > > >
> > > > While the performance of E$^3$ and V-MoE in OOD detection depends on the family scale, E$^3$ outperforms V-MoE on many other dimensions. E$^3$ is superior to V-MoE in terms of few-shot error and in-distribution ImageNet NLL (Figure 4), in-distribution and OOD ECE (Figure 5), NLL for Cifar10/100/-C (Figure 8). With these considered, E$^3$ can offer a significantly more performant model overall to both V-MoE and ViT.
> > > >
> > > > > it is also correct that E$^3$ cannot outperform V-MoE in OOD detection/NLL under distribution shift
> > > >
> > > > We disagree that E$^3$ cannot outperform V-MoE in NLL under distribution shift. By examining Figure 7, we can see that E$^3$ reliably outperforms V-MoE for larger ViT families (L/32, L/16, and H/14). To make this conclusion clearer, we have computed the following tables that show the percentage improvement in NLL (i.e. $\frac{NLL_{V-MoE} - NLL_{E^3}}{NLL_{V-MoE}} \times 100$ with positive values thus indicating E$^3$ improves upon V-MoE) averaged over {INetC, INet-A, INet-V2}, for a given ViT family and (K, M) in {(1, 2), (2, 2)} setting (respectively compared to V-MoE K=2 and K=4). We see that for all ViT families except S/32, E$^3$ outperforms V-MoE. This result is not unexpected, since ensembles tend to provide improved performance under distribution shift relative to single models ([Lakshminarayanan et al., 2016](https://arxiv.org/abs/1612.01474); [Ovadia et al., 2019](https://arxiv.org/abs/1906.02530); [Havasi et al., 2020](https://arxiv.org/pdf/2010.06610.pdf)).
> > > >
> > > > **E$^3$ (K=1, M=2)  versus V-MoE (K=2)**
> > > > |                         |    S/32 |      B/32 |      B/16 |     L/32 |     L/16 |     H/14 |
> > > > |:------------------------|--------:|----------:|----------:|---------:|---------:|---------:|
> > > > | V-MoE --> E$^3$ improvement in NLL (%) | -1.14849 | 0.522627 | 0.310433 | 5.33673 | 2.77826 | 8.33255 |
> > > >
> > > > **E$^3$ (K=2, M=2) versus V-MoE (K=4)**
> > > > |                         |     S/32 |     B/32 |    B/16 |     L/32 |     L/16 |
> > > > |:------------------------|---------:|---------:|--------:|---------:|---------:|
> > > > | V-MoE --> E$^3$ improvement in NLL (%) | -0.208668 | 3.90788 | 1.3325 | 7.36884 | 2.26589 |
> > > >
> > > > For OOD detection, the picture is more nuanced. While V-MoE is better for smaller ViT families, E$^3$ can be better for larger ones (e.g., L/32 and L/16 for CIFAR10-vs-CIFAR100 and CIFAR10-vs-Places365; Figure 6).
> > > >
> > > > All in all, we evaluate E$^3$ with a very diverse set of tasks and metrics, and like all methods, it has strengths and weaknesses. We include all of the results–positive and negative–to paint the full picture. Nonetheless, we note that for most of the cases considered, E$^3$ tends to outperform V-MoE on aggregate.
> > > >
> > > > We hope that this clarifies our claims about the performance of E$^3$.

---

> > > > > ### Author Response · Authors · 2022-07-11
> > > > > **Response to further comments -- part 2**
> > > > >
> > > > > We note the main acceptance criteria for TMLR:
> > > > >
> > > > > > **Are the claims made in the submission supported by accurate, convincing and clear evidence?**
> > > > > This is the most important criteria. This implies assessing the technical soundness as well as the clarity of the narrative and arguments presented.
> > > > > Any gap between claims and evidence should be addressed by the authors. Often, this will lead reviewers to ask the authors to provide more evidence by running more experiments. However, this is not the only way to address such concerns. Another is simply for the authors to adjust (reduce) their claims.
> > > > >
> > > > > We believe that our revised manuscript has substantiated with evidence all claims made in the paper. We have reported both the cases where E$^3$ performs well relative to V-MoE (few-shot error and in-distribution ImageNet NLL, in-distribution and OOD ECE, NLL for Cifar10/100/-C), as well as the more nuanced cases (OOD detection, NLL under distribution shift), so as to transparently inform future research.
> > > > >
> > > > > We note that the only other acceptance criteria is:
> > > > >
> > > > > > **Would some individuals in TMLR's audience be interested in the findings of this paper?**
> > > > > This is arguably the most subjective criteria, and therefore needs to be treated carefully. Generally, a reviewer that is unsure as to whether a submission satisfies this criteria should assume that it does.
> > > > > Crucially, it should  not  be used as a reason to reject work that isn't considered “significant” or “impactful” because it isn't achieving a new state-of-the-art on some benchmark. Nor should it form the basis for rejecting work on a method considered not “novel enough”, as novelty of the studied method is not a necessary criteria for acceptance. We explicitly avoid these terms (“significant”, “impactful”, “novel”), and focus instead on the notion of “interest”. If the authors make it clear that there is something to be learned by some researchers in their area from their work, then the criteria of interest is considered satisfied.  TMLR instead relies on certifications (such as “Featured” and “Outstanding”) to provide annotations on submissions that pertain to (more speculative) assertions on significance or potential for impact.
> > > > >
> > > > > This criteria explicitly suggests that 'novelty' should not be used as a necessary criteria for acceptance. As already highlighted by Reviewer n8qH “The empirical investigation on ensembling and MoE is valuable to the community”, this method and study is clearly of interest to members of the TMLR audience.

---

> > > > > ### Comment · Reviewer_JkZs · 2022-07-12
> > > > > **Thanks for your reply**
> > > > >
> > > > > Thanks for addressing the further concerns. I suggest the authors integrate these results with the manuscript carefully, and I will increase my score.

---

> > > > > > ### Author Response · Authors · 2022-07-13
> > > > > > **Changes to the text**
> > > > > >
> > > > > > We have added these tables and discussion in a new appendix M.3. We also briefly summarise the results and link to appendix M.3 from the discussion in the bullet point "E$^3$ becomes Pareto efficient for larger ViT families in the presence of distribution shift". Please let us know if this integration is sufficient or if you would prefer something else? Thank you!

---

### Review · Reviewer_Qd4u · 2022-06-23

**Summary Of Contributions:**

This paper presents a system "E3" combining MoE and ensemble averaging, inside of a vision transformer.  Input tokens are replicated ("tiling"), and independently processed in groups ("partitioning") by transformers with MLPs replaced by MoEs.  This is similar to V-MoE, with smaller sets of MoEs applied independently in groups and averaged at the end.  V-MoE using top-K of E experts has similar computation in the MoE part of the system to E3 using M tilings each choosing top-K/M of E/M experts --- but the latter can average over more diverse predictions, as exhibited by the measurements of KL divergence between groups.  Extensive experiments show the method outperforms vanilla ViT and V-MoE in classification for imagenet variants and few-shot classification tasks, particularly in the largest computational budgets, and mixed results in OOD.

**Requested Changes:**

See above questions.  Addressing the missing points / inconsistencies in figures for Sec 5 is most important.  No others are critical, but would further strengthen and clarify several points.

**Strengths And Weaknesses:**

Overall, the system is explained well and the benefits clearly demonstrated on multiple tasks measuring error vs computation.  Ablations demonstrate the independent effects of tiling and partioning.  Measuring KL divergence between groups notably confirms the intuition that ensembling helps when predictions are diverse; the overlap study particularly drives this point home.

There were a few more minor points I didn't entirely understand.  First, how is the tiling of M representations handled in transformer attention?  Is attention applied within each of the M groups, or can they mix with one another?  Are transformer weights in non-final MLPs shared between the M groups or unshared?  And how is the computational cost of these parts outside the MoE affected by tiling replication?

Sec 5 figs 4, 5, 6, 7:  The zoomed-in (red outline) plot and full plot on the left include different points.  It seems some were included in one but not the other.  Maybe to keep the figures less busy, although sometimes there are points missing in the zoomed-in plot as well, e.g. brown triangle in fig4.  It's also not clear in the figure whether E3 uses M=2 or 4, I only see one box size in each plot.  I found this inconsistency confusing, it would be better to either include all points or explain why some are missing in the caption.



Additional questions:

Is it possible to measure KL-divergence between experts in addition to between groups?  This could extend the diversity measurements to V-MoE without partitioning to provide interesting comparisons (currently these are marked n/a in the tables).  The gating weights may pose an issue here, but perhaps these could be incorporated in an interpolation so that full weight on 1 expert results in 0 diversity measure between experts, while equal weights yields the full distance comparison.

Figure 2 is interesting, illustrating static and adaptive ensembling (M and K), but I didn't see a similar plot for the E3 system.  It could be nice to have another plot like this to show the effect of K vs M:  since K*M total experts are selected, there is a question of how to select K and M for a computation budget, perhaps using a smaller MLP where K or M might be larger.

Does the order of experts in initialization affect the system?  The paper states that sparse MoE checkpoints are used.  Does this mean the model is initialized with trained V-MoE weights for E experts, but applied using M partitioned groups?  What if the order of the E experts is permuted so that different experts are in each group?

I don't see the number of experts E mentioned for the experiments in tables 4 and 5, though the text around table 2 and many other places says E=32.  Was this the case here as well?  If so, it could make sense to say something along the lines of "we set the number of experts E=32 unless specified otherwise".

table 2:  I think it could also be good to include plain (non-tiled, non-partitioned) MoE as a baseline, e.g. selecting the same number K*M of all E experts with a single gating; though, V-MoE is compared in other tables.

---

> ### Author Response · Authors · 2022-06-30
> **Response to reviewer Qd4u**
>
> Thank you for the constructive review! We respond to each of your points below.
>
> > There were a few more minor points I didn't entirely understand. First, how is the tiling of M representations handled in transformer attention? Is attention applied within each of the M groups, or can they mix with one another? Are transformer weights in non-final MLPs shared between the M groups or unshared? And how is the computational cost of these parts outside the MoE affected by tiling replication?
>
> The M representations are seen as independent by the transformer attention. I.e., they are seen the same way as tokens from separate input images – they do not mix with one another. The only time that the $M$ representations interact with one another is when the predictions are averaged at the end. To be more concrete, for an input with shape (batch size, num tokens, representation dim) after tiling the shape would be (batch size * $M$, num tokens, representation dim). Attention operates in the second dimension, independently of the first dimension.
>
> All of the parameters, except for the experts, are shared between the $M$ groups.
> The computational cost for the parts outside of the MoE *after the tiling* is increased as if the batch size were $M$ times bigger. Though, due to the ‘last-$n$’ setting we employ in this paper, this cost remains mild. For example, for L/16 and $K = M = 2$, E$^3$ uses only 14.75% more FLOPs than V-MoE; see Table 21.
>
> Please let us know if we can further clarify the text to make the above clearer.
>
> > Sec 5 figs 4, 5, 6, 7: The zoomed-in (red outline) plot and full plot on the left include different points. It seems some were included in one but not the other. Maybe to keep the figures less busy, although sometimes there are points missing in the zoomed-in plot as well, e.g. brown triangle in fig4. It's also not clear in the figure whether E3 uses M=2 or 4, I only see one box size in each plot. I found this inconsistency confusing, it would be better to either include all points or explain why some are missing in the caption.
>
> Unfortunately, putting all of the points in the zoomed-out plots made for unreadable figures, which is why we decided to include zoomed-in versions with more details. We have clarified this in all of the captions. We have also included the missing points in the zoomed-in plots. Please let us know if this can be further clarified.
>
> We mention in the first paragraph of Section 5 that all E$^3$ results are for ($K$, $M$) = (1, 2). We had also hoped that comparing the size of the markers to those of the V-MoE/ViT ensembles would make this clear. Do you have any suggestions to improve this issue?
>
> > Is it possible to measure KL-divergence between experts in addition to between groups?
>
> While we agree that this would certainly be an interesting investigation, we were prevented from taking action due to two issues. Firstly, experts do not output discrete probability distributions, but produce real-valued activations, so we would need some other diversity metric beyond the KL divergence. Moreover, by design, experts are not applied to the same inputs, and it was not clear how this should be taken into account to measure pairwise diversity across all experts.
>
> > Figure 2 is interesting, illustrating static and adaptive ensembling (M and K), but I didn't see a similar plot for the E3 system. It could be nice to have another plot like this to show the effect of K vs M: since K*M total experts are selected, there is a question of how to select K and M for a computation budget, perhaps using a smaller MLP where K or M might be larger.
>
> We agree that this would be an interesting investigation. However, in order to do it properly, we would have to address the possible upstream-vs-downstream mismatch (c.f., our response to reviewer n8qH). This is unfortunately out of the scope of our current investigations that are essentially guided by practical considerations (e.g., reusing the most popular and available V-MoE checkpoints at the expense of possible downstream mismatch).
>
> Regarding the choice of $K$ and $M$ for a fixed computational budget, we do have some concrete advice: cost increases more quickly with $M$ than $K$ and thus it makes sense to increase $K$ before $M$. E.g., see Tables 12 and 20.
>
> We have not observed in any of our experiments the best E$^3$ of size $T$ (e.g., S/32) being better than the worst of E$^3$ of size $T+1$ (e.g., B/32), where the MLPs for $T$ are smaller than for $T+1$. In the light of Appendix D.2 from Dosovitskiy et al., 2021, we can see it is generally not effective to *independently* scale (up/down) the size of the MLPs in ViT models. Instead, it is advocated to scale all dimensions proportionally.

---

> > ### Author Response · Authors · 2022-06-30
> > **Response to reviewer Qd4u – part 2**
> >
> > > Does the order of experts in initialization affect the system? The paper states that sparse MoE checkpoints are used. Does this mean the model is initialized with trained V-MoE weights for E experts, but applied using M partitioned groups? What if the order of the E experts is permuted so that different experts are in each group?
> >
> > Indeed, we do initialize with a trained V-MoE with $E$ experts but applied using $M$ partitioned groups. As a result, if we permute the experts (together with the columns of the router weights, for consistency of the routing), we obtain $M$ initial groups composed of different experts (e.g., $M=2$, $E=4$, we could have $\mathcal{E}_1$={1, 2} and $\mathcal{E}_2$={3, 4} becoming with a permutation $\mathcal{E}’_1$={3, 1}, $\mathcal{E}’_2$={2, 4}). We confirm in Appendix H.3, with a new experiment, that permuting the experts has no impact on the performance of E$^3$.
> >
> > There is one situation where E$^3$ does not depend on pre-trained experts, when training on ImageNet from scratch (Appendix K). In that case, E$^3$ was not sensitive to the random initialization of the experts.
> >
> > > I don't see the number of experts E mentioned for the experiments in tables 4 and 5, though the text around table 2 and many other places says E=32. Was this the case here as well? If so, it could make sense to say something along the lines of "we set the number of experts E=32 unless specified otherwise".
> >
> > Thanks for the suggestion, we have clarified this in both Sections 3 and 5.
> >
> > > table 2: I think it could also be good to include plain (non-tiled, non-partitioned) MoE as a baseline, e.g. selecting the same number K*M of all E experts with a single gating; though, V-MoE is compared in other tables.
> >
> > Thanks again for the suggestion, we now include V-MoE with $K$*$M$ experts in Table 2.

---

> > > ### Comment · Reviewer_Qd4u · 2022-07-13
> > > **comments**
> > >
> > > Thanks for your responses.  These have addressed my concerns, in particular it's now clearer that the zoomed-in figure includes additional experiments only, and the zoomed-out figure is a subsample.  The additional permutation experiment is also good to see, as is the confirmation of how the groups are treated as separate batch items by the rest of the transformer.

---

> ### Author Response · Authors · 2022-07-13
> **Follow-up**
>
> Please let us know if our response has addressed your concerns, or if you would like any further clarifications.

---

### Author Response · Authors · 2022-06-30
**Thank you to the reviewers!**

We thank all of the reviewers for their time and constructive criticism which has led to numerous additions and clarifications in the paper. For your convenience, all of the changes have been highlighted by coloured and bolded text. The text colour has been matched to the reviewers who made the suggestions with n8qH being $\textcolor{OrangeRed}{\mathbf{OrangeRed}}$, JkZs being $\textcolor{ForestGreen}{\mathbf{ForestGreen}}$, and Qd4u being $\textcolor{Plum}{\mathbf{Plum}}$. We look forward to further discussions with the reviewers.

---

### Decision · Action_Editors · 2022-07-22

**Recommendation:** Accept as is

**Comment:**

This paper investigated MoE and ensemble averaging within the vision transformer. Though a similar idea has been explored before in V-MoE, the authors have given a thorough comparison between the proposed algorithm and V-MoE, and demonstrated the algorithm's advantages in terms of few-shot error and in-distribution ImageNet NLL,  in-distribution, and OOD ECE,  NLL for Cifar10/100/-C. Reviewers agree that the empirical findings in this paper are meaningful to the research community.  The authors have updated the manuscript accordingly by including the new discussions and experimental results.